# Interaction Asymmetry: A General Principle for Learning Composable Abstractions

**Jack Brady**[*1,2]    **Julius von Kügelgen**[3]    **Sébastien Lachapelle**[4]

**Simon Buchholz**[1,2]    **Thomas Kipf**[†5]    **Wieland Brendel**[†1,2,6]

[1] Max Planck Institute for Intelligent Systems, Tübingen    [2] Tübingen AI Center    [3] ETH Zürich
[4] Samsung - SAIT AI Lab, Montreal    [5] Google DeepMind    [6] ELLIS Institute, Tübingen

## Abstract

Learning disentangled representations of concepts and re-composing them in unseen ways is crucial for generalizing to out-of-domain situations. However, the underlying properties of concepts that enable such disentanglement and compositional generalization remain poorly understood. In this work, we propose the principle of *interaction asymmetry* which states: "Parts of the same concept have more complex interactions than parts of different concepts". We formalize this via block diagonality conditions on the $(n+1)^{\text{th}}$ order derivatives of the generator mapping concepts to observed data, where different orders of "complexity" correspond to different $n$. Using this formalism, we prove that interaction asymmetry enables *both* disentanglement and compositional generalization. Our results unify recent theoretical results for learning concepts of objects, which we show are recovered as special cases with $n = 0$ or $1$. We provide results for up to $n = 2$, thus extending these prior works to more flexible generator functions, and conjecture that the same proof strategies generalize to larger $n$. Practically, our theory suggests that, to disentangle concepts, an autoencoder should penalize its latent capacity and the interactions between concepts during decoding. We propose an implementation of these criteria using a flexible Transformer-based VAE, with a novel regularizer on the attention weights of the decoder. On synthetic image datasets consisting of objects, we provide evidence that this model can achieve comparable object disentanglement to existing models that use more explicit object-centric priors.

## 1 Introduction

A core feature of human cognition is the ability to use abstract conceptual knowledge to generalize far beyond direct experience (Behrens et al., 2018; Mitchell, 2021; Murphy, 2004; Tenenbaum et al., 2011). For example, by applying abstract knowledge of the concept "chair", we can easily infer how to use a "chair on a beach", even if we have not yet observed this combination of concepts. This feat is non-trivial and requires solving two key problems. Firstly, one must acquire an abstract, internal model of different concepts in the world. This implies learning a *separate* internal representation of each concept from sensory observations. Secondly, these representations must remain valid when observations consist of novel compositions of concepts, e.g., "chair" and "beach". In machine learning, these two problems are commonly referred to as learning *disentangled representations* (Bengio et al., 2013; Higgins et al., 2018; Schölkopf et al., 2021) and *compositional generalization* (Fodor and Pylyshyn, 1988; Goyal and Bengio, 2022; Greff et al., 2020; Lake et al., 2017).

Both problems are known to be challenging due to the issue of *non-identifiability* (Hyvärinen et al., 2023). Namely, many models can explain the same data equally well, but only some will learn representations of concepts which are disentangled and generalize compositionally. To guarantee *identifiability* with respect to (w.r.t.) these criteria, it is necessary to incorporate suitable inductive biases into a model (Hyvärinen and Pajunen, 1999; Lachapelle et al., 2023; Locatello et al., 2019). These inductive biases, in turn, must reflect some underlying properties of the concepts which give rise to observed data. This raises a fundamental question: What properties of concepts enable learning models which provably achieve disentanglement and compositional generalization?

---

[*]Correspondence to: `jack.brady@tue.mpg.de`. [†]Joint senior author.
    Code available at: github.com/JackBrady/interaction-asymmetry

Figure 1: **Illustration of Interaction Asymmetry.** *(Left)* Observations $\boldsymbol{x}$ result from a generator $\boldsymbol{f}$ applied to latent slots $\boldsymbol{z}_{B_k}$ that represent separate concepts. As indicated by the reflection of the cylinder upon the cube, slots can interact during generation. Our key assumption, interaction asymmetry, states that these interactions across slots must be less complex than interactions within the same slot. *(Right)* This is formalized by assuming block-diagonality *across* but not *within* slots for the $(n+1)^{\text{th}}$ order derivatives of the generator, i.e., $D^{n+1}\boldsymbol{f}$.

Many works aim to answer this question by studying properties enabling either disentanglement or compositional generalization *in isolation*. This is insufficient, however, as disentanglement alone does not imply compositional generalization (Montero et al., 2022a; 2021; Schott et al., 2022), while compositional generalization requires first disentangling the concepts to be composed. Only a few studies investigate properties enabling *both* disentanglement and compositional generalization (Brady et al., 2023; Lachapelle et al., 2023; Wiedemer et al., 2024a). Yet, the properties proposed in these works are rather restrictive and specific to objects in simple visual scenes. There is growing evidence, however, that the principles humans use to learn conceptual knowledge are not concept-specific, but shared across different concepts (objects, attributes, events, etc.) (Behrens et al., 2018; Constantinescu et al., 2016; Hawkins et al., 2018). This suggests there exist some *general* properties of concepts which enable both disentanglement and compositional generalization.

In this work, we seek to formulate such a general property for disentangling and composing concepts. We begin by aiming to deduce, from first principles, properties which are fundamental to concepts (§ 3). From this, we arrive at the guiding principle of *interaction asymmetry* (Principle 3.1) stating: "Parts of the same concept have more complex interactions than parts of different concepts". As illustrated in Fig. 1 (left), we define concepts as distinct groups, or *slots*, of latent variables which generate the observed data (§ 2). Interaction asymmetry is then formalized as a block-diagonality condition *across* but *not within* slots of $D^{n+1}\boldsymbol{f}$, the tensor of $(n+1)^{\text{th}}$ order partial derivatives of the generator function (Asm. 3.5), where $n$ determines the complexity of interactions, see Fig. 1 (right).

**Theory.** Using this formulation, we prove that interaction asymmetry dually enables *both* disentanglement (Thm. 4.3) *and* compositional generalization (Thm. 4.4). We also show that our formalism provides a unifying framework for prior results of Brady et al. (2023) and Lachapelle et al. (2023), by proving that the properties studied in these works for visual objects are special cases of our assumptions for $n=0$ and $1$, respectively. We provide results for up to $n=2$, thereby extending these prior works to more general function classes, and conjecture that our results generalize to arbitrary $n \geq 0$.

**Method.** Our theory suggests that to disentangle concepts, a model should (i) enforce invertibility, without using more latent dimensions than necessary, and (ii) penalize interactions across slots during decoding. To translate these insights into a practical method, we leverage a VAE loss (Kingma and Welling, 2014) for (i), and observe that the Transformer architecture (Vaswani et al., 2017) offers an approximate means to achieve (ii) since interactions are determined by the attention weights of the model. To this end, we introduce an inexpensive interaction regularizer for a cross-attention mechanism, which we incorporate, with the VAE loss, into a flexible Transformer-based model (§ 5).

**Empirical Results.** We test this model's ability to disentangle concepts of visual objects on a Sprites dataset (Watters et al., 2019a) and on CLEVR6 (Johnson et al., 2017). We find that the model reliably learns disentangled representations of objects, improving performance over an unregularized Transformer (§ 6). Furthermore, we provide preliminary evidence that our regularized Transformer can achieve comparable performance to models with more explicit object-centric priors such as Slot Attention (Locatello et al., 2020b) and Spatial Broadcast Decoders (Watters et al., 2019b).

**Notation.** We write scalars in lowercase ($z$), vectors in lowercase bold ($\boldsymbol{z}$), and matrices in capital bold ($\boldsymbol{M}$). $[K]$ stands for $\{1, 2, ..., K\}$. $D_i$ and $D_{i,j}^2$ stand for the first- and second-order partial derivatives with respect to (w.r.t.) $z_i$ and $(z_i, z_j)$, respectively. If $B \subseteq [n]$ and $\boldsymbol{z} \in \mathbb{R}^n$, $\boldsymbol{z}_B$ denotes the subvector $(z_i)_{i \in B}$ indexed by $B$. A function is $C^n$ if it is $n$-times continuously differentiable.

## 2 BACKGROUND

We begin with formalizing the core ideas of concepts, disentanglement, and compositional generalization, mostly following the setup of Lachapelle et al. (2023). To begin, we assume that the observed data $x \in \mathcal{X} \subset \mathbb{R}^{d_x}$ results from applying a diffeomorphic generator $f : \mathcal{Z} \to \mathcal{X}$ to latent vectors $z \in \mathcal{Z} := \mathbb{R}^{d_z}$, sampled from some distribution $p_z$. Concepts underlying $x$ (objects, attributes, events, etc.) are then modelled as $K$ disjoint groups or *slots* of latents $z_{B_k}$ such that $z = (z_{B_1}, ..., z_{B_K})$, where $B_k \subseteq [d_z]$. We assume that $p_z$ is only supported on a subset $\mathcal{Z}_{\mathrm{supp}} \subseteq \mathcal{Z}$ which gives rise to observed data $\mathcal{X}_{\mathrm{supp}} := f(\mathcal{Z}_{\mathrm{supp}})$. This generative process can be summarized as:

$$x = f(z), \qquad z \sim p_z, \qquad \mathrm{supp}(p_z) = \mathcal{Z}_{\mathrm{supp}}. \tag{2.1}$$

Next, consider a model $\hat{f} : \mathcal{Z} \to \mathbb{R}^{d_x}$ trained to be invertible from $\mathcal{X}_{\mathrm{supp}}$ to $\hat{\mathcal{Z}}_{\mathrm{supp}} := \hat{f}^{-1}(\mathcal{X}_{\mathrm{supp}})$, whose inverse $\hat{f}^{-1}$ maps to a representation $\hat{z} \in \hat{\mathcal{Z}}_{\mathrm{supp}} \subseteq \mathcal{Z}$. This model is said to learn a *disentangled* representation of $z \in \mathcal{Z}_{\mathrm{supp}}$ if each model slot $\hat{z}_{B_j}$ captures exactly one concept $z_{B_k}$.

**Definition 2.1** (Disentanglement). Let $f : \mathcal{Z} \to \mathcal{X}$ be a diffeomorphism and $\bar{\mathcal{Z}} \subseteq \mathcal{Z}$. A model $\hat{f}$ *disentangles* $z$ on $\bar{\mathcal{Z}}$ w.r.t. $f$ if there exist a permutation $\pi$ of $[K]$ and slot-wise diffeomorphism $h = (h_1, \ldots, h_K)$ with $h_k : \mathbb{R}^{|B_{\pi(k)}|} \to \mathbb{R}^{|B_k|}$ and $|B_{\pi(k)}| = |B_k|$ such that for all $z \in \bar{\mathcal{Z}}$:

$$\hat{f}\left(h_1\left(z_{B_{\pi(1)}}\right), \ldots, h_K\left(z_{B_{\pi(K)}}\right)\right) = f(z). \tag{2.2}$$

In other words, a representation is disentangled if the model inverts the generator up to permutation and reparametrization of the slots. For *compositional generalization*, we would like this to hold not only on $\mathcal{Z}_{\mathrm{supp}}$ but also for arbitrary combinations of the slots therein. Namely, also on the set

$$\mathcal{Z}_{\mathrm{CPE}} := \mathcal{Z}_1 \times \mathcal{Z}_2 \times \cdots \times \mathcal{Z}_K, \qquad \text{with} \qquad \mathcal{Z}_k := \{z_{B_k} \mid z \in \mathcal{Z}_{\mathrm{supp}}\} \tag{2.3}$$

where $\mathcal{Z}_k$ denote the marginal supports of $p_z$ and $\mathcal{Z}_{\mathrm{CPE}}$ the *Cartesian-product extension* (Lachapelle et al., 2023) of $\mathcal{Z}_{\mathrm{supp}}$. In general, $\mathcal{Z}_{\mathrm{supp}}$ is a subset of $\mathcal{Z}_{\mathrm{CPE}}$. Thus, to generalize compositionally, a model must also achieve disentanglement "out-of-domain" on novel compositions of slots in $\mathcal{Z}_{\mathrm{CPE}}$.

**Definition 2.2** (Compositional Generalization). Let $f : \mathcal{Z} \to \mathcal{X}$ be a diffeomorphism. A model $\hat{f}$ that disentangles $z$ on $\mathcal{Z}_{\mathrm{supp}}$ w.r.t. $f$ (Defn. 2.1) *generalizes compositionally* if it also disentangles $z$ on $\mathcal{Z}_{\mathrm{CPE}}$ w.r.t. $f$.

**On the Necessity of Inductive Biases.** It is well known that only a small subset of invertible models achieve disentanglement on $\mathcal{Z}_{\mathrm{supp}}$ (Hyvärinen and Pajunen, 1999; Locatello et al., 2019) or generalize compositionally to $\mathcal{Z}_{\mathrm{CPE}}$ (Lachapelle et al., 2023). To provably achieve these goals (without explicit supervision), we thus need to further restrict the space of permissible models, i.e., place additional assumptions on the generative process in Eq. (2.1). Such assumptions then translate into inductive biases on a model. To this end, the core challenge is formulating assumptions on $p_z$ or $f$ which *faithfully* reflect properties of concepts, while sufficiently restricting the problem.

**Assumptions on $p_z$.** To guarantee disentanglement, several assumptions on $p_z$ have been proposed, such as conditional independence of latents given an auxiliary variable (Hyvärinen et al., 2019; Khemakhem et al., 2020); particular temporal (Hälvä and Hyvarinen, 2020; Hyvärinen and Morioka, 2016; 2017; Klindt et al., 2021), spatial (Hälvä et al., 2021; 2024), or other latent structures (Kivva et al., 2022; Kori et al., 2024); multiple views (Ahuja et al., 2022; Brehmer et al., 2022; Gresele et al., 2020; Locatello et al., 2020a; von Kügelgen et al., 2021; Yao et al., 2024; Zimmermann et al., 2021); or interventional information (Buchholz et al., 2023; Lachapelle et al., 2022; 2024; Lippe et al., 2022; 2023; Varici et al., 2024; von Kügelgen et al., 2023). While sufficient for disentanglement, such assumptions do not guarantee compositional generalization. The latter requires that the behavior of the generator on $\mathcal{Z}_{\mathrm{CPE}}$ can be determined solely from its behavior on $\mathcal{Z}_{\mathrm{supp}}$ (see Defn. 2.2). In the most extreme case, where the values of each slot $z_{B_k}$ are seen only once, $\mathcal{Z}_{\mathrm{supp}}$ will be a one-dimensional manifold embedded in $\mathcal{Z}$, while $\mathcal{Z}_{\mathrm{CPE}}$ is always $d_z$-dimensional. This highlights that generalizing from $\mathcal{Z}_{\mathrm{supp}}$ to $\mathcal{Z}_{\mathrm{CPE}}$ is only possible if the form of the generator $f$ is restricted.

**Assumptions on $f$.** Restrictions on $f$ which enable compositional generalization have been proposed by Dong and Ma (2022); Lippl and Stachenfeld (2024); Wiedemer et al. (2024b). Yet, these results rely on quite limited function classes and do not address disentanglement, assuming it to be solved a priori. Conversely, several works explore restrictions on $Df$ such as orthogonality (Buchholz et al., 2022; Gresele et al., 2021; Horan et al., 2021) or sparsity (Leemann et al., 2023; Moran

et al., 2022; Zheng and Zhang, 2023) which address disentanglement but not compositional generalization. More recently, Brady et al. (2023) and Lachapelle et al. (2023) proposed assumptions on $\boldsymbol{f}$ which enable both disentanglement and compositional generalization (Wiedemer et al., 2024a). Yet, these assumptions are overly restrictive such that $\boldsymbol{f}$ can only model limited types of concepts, e.g., non-interacting objects, and not more general concepts. We discuss these two works further in § 4.3.

## 3  THE INTERACTION ASYMMETRY PRINCIPLE

In this section, we attempt to formulate assumptions that enable disentanglement and compositional generalization, while capturing more general properties of concepts. To approach this, we take a step back and try to understand what are the defining properties of concepts. Specifically, we consider the question: *Why are some structures in the world recognized as different concepts (e.g., apple vs. dog) and others as part of the same concept?* We propose an answer to this for concepts grounded in sensory data, such as objects (e.g., "car"), events (e.g., "making coffee"), or attributes (e.g., "color").

Sensory-grounded concepts correspond to reoccurring visual or temporal patterns that follow an abstract template. They tend to be modular, such that independently changing one concept generally leaves the structure of other concepts intact (Greff et al., 2015, § 4.1.1; Peters et al., 2017). For example, a car can change position without affecting the structure of the street, buildings, or people around it. Thus, different concepts appear, in some sense, to *not interact*.

On the other hand, parts of the same concept do not seem to possess this modularity. Namely, arbitrarily changing one part of a concept without adjusting other parts is generally not possible without destroying its inherent structure. For example, it is not possible to change the position of the front half of a car, while maintaining something we would still consider a car, without also changing the back half's position. Thus, parts of the same concept seem to *interact*.

This may then lead us to answer our initial question with: Parts of the same concept interact, while different concepts do not. However, this is an oversimplified view, as parts of different concepts can, in fact, interact. For example, in Fig. 1 we see the purple cylinder reflects upon and thus interacts with the golden cube. However, such interactions across concepts appear somehow simpler than interactions within a concept: whereas the latter can alter the concept's structure, the former generally will not. In other words, the complexity of interaction within and across concepts appears to be asymmetric. We formulate this as the following principle (see Appx. G.1 for related principles).

**Principle 3.1** (Interaction Asymmetry). *Parts of the same concept have more complex interactions than parts of different concepts.*

To investigate the implications of Principle 3.1 for disentanglement and compositional generalization, we must first give it a precise formalization. To this end, we need a mathematical definition of the "complexity of interaction" between parts of concepts, i.e., groups of latents from the same or different slots. This can be formalized either through assumptions on the latent distribution $p_{\boldsymbol{z}}$ or on the generator $\boldsymbol{f}$. Since the latter are essential for compositional generalization, this is our focus.

Let us start by imagining what it would mean if two groups of latent components $\boldsymbol{z}_A$ and $\boldsymbol{z}_B$ interact with *no complexity*, i.e., have *no interaction* within $\boldsymbol{f}$. A natural way to formalize this is that $\boldsymbol{z}_A$ and $\boldsymbol{z}_B$ affect distinct output components $f_l$. Mathematically, this is captured as follows.

**Definition 3.2** (At most $0^{\text{th}}$ order/No interaction). Let $\boldsymbol{f} : \mathcal{Z} \to \mathcal{X}$ be $C^1$, and let $A, B \subseteq [d_z]$ be non-empty. $\boldsymbol{z}_A$ and $\boldsymbol{z}_B$ have *no interaction* within $\boldsymbol{f}$ if for all $\boldsymbol{z} \in \mathcal{Z}$, and all $i \in A, j \in B$:
$$D_i \boldsymbol{f}(\boldsymbol{z}) \odot D_j \boldsymbol{f}(\boldsymbol{z}) = \boldsymbol{0} \,. \tag{3.1}$$

To define the next order of interaction complexity, we assume that $\boldsymbol{z}_A$ and $\boldsymbol{z}_B$ *do interact*, i.e., they affect the same output $f_l$ such that $D_i f_l(\boldsymbol{z})$ and $D_j f_l(\boldsymbol{z})$ are non-zero for some $i \in A$, $j \in B$. This interaction, however, should have the lowest possible complexity. A natural way to capture this is to say that $z_i$ can affect the same output $f_l$ as $z_j$ but cannot affect *the way in which $f_l$ depends on $z_j$*. Since the latter is captured by $D_j f_l(\boldsymbol{z})$, this amounts to a question about the $2^{\text{nd}}$ order derivative $D_{i,j}^2 f_l$. We thus arrive at the following definition for the next order of interaction complexity.

**Definition 3.3** (At most $1^{\text{st}}$ order interaction). Let $\boldsymbol{f} : \mathcal{Z} \to \mathcal{X}$ be $C^2$, and let $A, B \subseteq [d_z]$ be non-empty. $\boldsymbol{z}_A$ and $\boldsymbol{z}_B$ have *at most $1^{st}$ order interaction* within $\boldsymbol{f}$ if for all $\boldsymbol{z} \in \mathcal{Z}$, and all $i \in A, j \in B$:
$$D_{i,j}^2 \boldsymbol{f}(\boldsymbol{z}) = \boldsymbol{0} \,. \tag{3.2}$$

Using the same line of reasoning, we can continue to define interactions at increasing orders of complexity. For example, for *at most $2^{nd}$ order interaction*, $z_i$ can affect the derivative $D_j f_l$, such that $D_{i,j}^2 f_l(\boldsymbol{z}) \neq 0$, but cannot affect the way in which $D_j f_l$ depends on any other $z_k$, i.e., $D_{i,j,k}^3 f_l(\boldsymbol{z}) = 0$. This leads to a general definition of interactions with *at most $n^{th}$ order* complexity.

**Definition 3.4** (At most $n^{th}$ order interaction). Let $n \geq 1$ be an integer. Let $\boldsymbol{f} : \mathcal{Z} \to \mathcal{X}$ be $C^{n+1}$. Let $A, B \subseteq [d_z]$ be non-empty. $\boldsymbol{z}_A$ and $\boldsymbol{z}_B$ have *at most $n^{th}$ order interaction* within $\boldsymbol{f}$ if for all $\boldsymbol{z} \in \mathcal{Z}$, all $i \in A, j \in B$, and all multi-indices $\boldsymbol{\alpha} \in \mathbb{N}_0^{d_z}$ with $|\boldsymbol{\alpha}| = n + 1$ and $1 \leq \alpha_i, \alpha_j$:

$$D^{\boldsymbol{\alpha}} \boldsymbol{f}(\boldsymbol{z}) = \boldsymbol{0} \,. \tag{3.3}$$

In other words, $\boldsymbol{z}_A$ and $\boldsymbol{z}_B$ have at most $n^{\text{th}}$ order interaction within $\boldsymbol{f}$ if all higher-than-$n^{\text{th}}$ order cross partial derivatives w.r.t. at least one component of $\boldsymbol{z}_A$ and of $\boldsymbol{z}_B$ are zero everywhere. Otherwise, if the statement in Defn. 3.4 does not hold for some $\boldsymbol{z} \in \mathcal{Z}$, $i \in A$, and $j \in B$, we say that $\boldsymbol{z}_A$ and $\boldsymbol{z}_B$ have $(n+1)^{\text{th}}$ *order interaction at $\boldsymbol{z}$* (and similarly for $1^{\text{st}}$ order interaction if Defn. 3.2 does not hold). With these definitions, we can now provide a precise formalization of Principle 3.1.

**Assumption 3.5** (Interaction asymmetry (formal)). There exists $n \in \mathbb{N}_0$ such that (i) any two distinct slots $\boldsymbol{z}_{B_i}$ and $\boldsymbol{z}_{B_j}$ have *at most $n^{th}$ order interaction* within $\boldsymbol{f}$; and (ii) for all $\boldsymbol{z} \in \mathcal{Z}$, all slots $\boldsymbol{z}_{B_k}$ and all non-empty $A, B$ with $B_k = A \cup B$, $\boldsymbol{z}_A$ and $\boldsymbol{z}_B$ have $(n+1)^{th}$ *order interaction* within $\boldsymbol{f}$ at $\boldsymbol{z}$.

We emphasize that Asm. 3.5 (ii) does not state that *all* subsets of latents within a slot must have $(n+1)^{\text{th}}$ order interaction, but only that a slot cannot be *split* into two parts with at most $n^{\text{th}}$ order interaction, see Fig. 1 (right). We also note that condition (ii) must hold *uniformly* over $\mathcal{Z}$, which resembles the notion of "uniform statistical dependence" among latents introduced by Hyvärinen and Morioka (2017, Defn. 1). For further discussions of Asm. 3.5, see Appx. H.1.

## 4 THEORETICAL RESULTS

We now explore the theoretical implications of Asm. 3.5 for disentanglement on $\mathcal{Z}_{\text{supp}}$ and compositional generalization to $\mathcal{Z}_{\text{CPE}}$. We provide our results for up to at most $2^{\text{nd}}$ order interaction across slots. All results—i.e., at most $0^{\text{th}}$ (no interaction), at most $1^{\text{st}}$, and at most $2^{\text{nd}}$ order interaction—use a unified proof strategy. Thus, we conjecture this strategy can also be used to obtain results for $n \geq 3$. This, however, would require taking $(n + 1) \geq 4$ derivatives of compositions of multivariate functions, which becomes very tedious as $n$ grows. General $n^{\text{th}}$ order results are thus left for future work.

### 4.1 DISENTANGLEMENT

We start by proving disentanglement on $\mathcal{Z}_{\text{supp}}$ for which we will need two additional assumptions.

**Basis-Invariant Interactions.** First, one issue we must address is that our formalization of interaction asymmetry (Asm. 3.5) is not *basis invariant*. Specifically, it is possible that all splits of a slot $\boldsymbol{z}_{B_k}$ have $(n+1)^{\text{th}}$ order interactions while for $\boldsymbol{M}_k \boldsymbol{z}_{B_k}$, with $\boldsymbol{M}_k$ a slot-wise change of basis matrix, they have at most $n^{\text{th}}$ order interactions. Since $\boldsymbol{M}_k$ need not affect interactions across slots, interaction asymmetry may no longer hold in the new basis. This makes it ambiguous whether interaction asymmetry is truly satisfied, as $\boldsymbol{z}_{B_k}$ and $\boldsymbol{M}_k \boldsymbol{z}_{B_k}$ contain the same information. To address this, we assume interaction asymmetry holds for all slot-wise basis changes, or *equivalent generators*.

**Definition 4.1** (Equivalent Generators). A function $\bar{\boldsymbol{f}} : \mathbb{R}^{d_z} \to \mathbb{R}^{d_x}$ is said to be *equivalent* to a generator $\boldsymbol{f}$ if for all $k \in [K]$ there exists an invertible matrix $\boldsymbol{M}_k \in \mathbb{R}^{|B_k| \times |B_k|}$ such that

$$\forall \boldsymbol{z} \in \mathbb{R}^{d_z} : \qquad \bar{\boldsymbol{f}}(\boldsymbol{M}_1 \boldsymbol{z}_{B_1}, \dots, \boldsymbol{M}_K \boldsymbol{z}_{B_K}) = \boldsymbol{f}(\boldsymbol{z}_{B_1}, \dots, \boldsymbol{z}_{B_K}). \tag{4.1}$$

**Sufficient Independence.** We require one additional assumption on $\boldsymbol{f}$ which we call *sufficient independence*. This assumption amounts to a linear independence condition on blocks of higher-order derivatives of $\boldsymbol{f}$. Its main purpose is to remove redundancy in the derivatives of $\boldsymbol{f}$ across slots, which can be interpreted as further constraining the interaction across slots during generation. In the case

---

A *multi-index* is an ordered tuple $\boldsymbol{\alpha} = (\alpha_1, \alpha_2, ..., \alpha_d)$ of non-negative integers $\alpha_i \in \mathbb{N}_0$, with operations $|\boldsymbol{\alpha}| := \alpha_1 + \alpha_2 + ... + \alpha_d$, $\boldsymbol{z}^{\boldsymbol{\alpha}} := z_1^{\alpha_1} z_2^{\alpha_2} ... z_d^{\alpha_d}$, and $D^{\boldsymbol{\alpha}} := \frac{\partial^{\alpha_1}}{\partial z_1^{\alpha_1}} \frac{\partial^{\alpha_2}}{\partial z_2^{\alpha_2}} ... \frac{\partial^{\alpha_d}}{\partial z_d^{\alpha_d}}$, see Appx. B for details.

of $n = 0$ (i.e., no interaction across slots), sufficient independence reduces to linear independence between slot-wise Jacobians of $\boldsymbol{f}$ (Defn. A.8). This is satisfied automatically since $\boldsymbol{f}$ is a diffeomorphism. When $n > 0$, we require an analogous linear independence condition on higher order derivatives of $\boldsymbol{f}$. Below, we present this for the case $n = 2$, while for $n = 1$, it is presented in Defn. A.9.

**Definition 4.2** (Sufficient Independence ($2^{\text{nd}}$ Order)). A $C^3$ function $\boldsymbol{f} : \mathbb{R}^{d_z} \to \mathbb{R}^{d_x}$ with at most $2^{\text{nd}}$ order interactions across slots is said to have *sufficiently independent* derivatives if $\forall \boldsymbol{z} \in \mathbb{R}^{d_z}$:

$$\text{rank}\left(\left[\left[D_i \boldsymbol{f}(\boldsymbol{z})\right]_{i \in B_k} \left[D^2_{i,i'}\boldsymbol{f}(\boldsymbol{z})\right]_{i \in B_k, i' \in [d_z]} \left[D^3_{i,i',i''}\boldsymbol{f}(\boldsymbol{z})\right]_{(i,i',i'') \in B^3_k}\right]_{k \in [K]}\right)$$

$$= \sum_{k \in [K]} \left[\text{rank}\left(\left[\left[D_i \boldsymbol{f}(\boldsymbol{z})\right]_{i \in B_k} \left[D^2_{i,i'}\boldsymbol{f}(\boldsymbol{z})\right]_{i \in B_k, i' \in [d_z]}\right]\right) + \text{rank}\left(\left[D^3_{i,i',i''}\boldsymbol{f}(\boldsymbol{z})\right]_{(i,i',i'') \in B^3_k}\right)\right] .$$

With Defns. 4.1 and 4.2, we can now state our theoretical results; see Appx. A for complete proofs.

**Theorem 4.3** (Disentanglement on $\mathcal{Z}_{\text{supp}}$). *Let $n \in \{0, 1, 2\}$. Let $\boldsymbol{f} : \mathcal{Z} \to \mathcal{X}$ be a $C^{n+1}$ diffeomorphism satisfying interaction asymmetry (Asm. 3.5) for all equivalent generators (Defn. 4.1) and sufficient independence (Appx. A.2). Let $\mathcal{Z}_{\text{supp}}$ be regular closed (Defn. A.3), path-connected (Defn. A.14) and aligned-connected (Defn. A.16). A model $\hat{\boldsymbol{f}} : \mathcal{Z} \to \mathbb{R}^{d_x}$ disentangles $\boldsymbol{z}$ on $\mathcal{Z}_{\text{supp}}$ w.r.t. $\boldsymbol{f}$ (Defn. 2.1) if it is* (i) *a $C^{n+1}$ diffeomorphism between $\hat{\mathcal{Z}}_{\text{supp}}$ and $\mathcal{X}_{\text{supp}}$ with* (ii) *at most $n^{\text{th}}$ order interactions across slots (Defn. 3.4) on $\hat{\mathcal{Z}}_{\text{supp}}$.*

**Intuition.** Assume for a contradiction that $\boldsymbol{h} := \hat{\boldsymbol{f}}^{-1} \circ \boldsymbol{f}$ *entangles* a ground-truth slot $\boldsymbol{z}_{B_k}$, i.e., $D_{B_k}\boldsymbol{h}(\boldsymbol{z})$ has multiple non-zero blocks. Because $\boldsymbol{f}$ and $\hat{\boldsymbol{f}}$ are invertible, $\boldsymbol{h}$ must encode all of $\boldsymbol{z}_{B_k}$ in $\hat{\boldsymbol{z}} := \boldsymbol{h}(\boldsymbol{z})$. Further, because $\boldsymbol{f}$ satisfies interaction asymmetry, $\boldsymbol{z}_{B_k}$ cannot be split into two parts with less than $(n+1)^{\text{th}}$ order interaction. Taken together, this implies that if $\boldsymbol{h}$ entangles $\boldsymbol{z}_{B_k}$, then there exist parts $\boldsymbol{z}_A$ and $\boldsymbol{z}_B$ of $\boldsymbol{z}_{B_k}$, with $(n+1)^{\text{th}}$ order interaction, encoded in different model slots. Since the model $\hat{\boldsymbol{f}}$ is constrained to have at most $n^{\text{th}}$ order interactions *across* slots, it cannot capture this interaction. Thus, the only way that $\hat{\boldsymbol{f}}$ can satisfy (i) and (ii) without achieving disentanglement is if reparametrizing $\boldsymbol{z}$ via $\boldsymbol{h}$ removed the interaction between $\boldsymbol{z}_A$ and $\boldsymbol{z}_B$. This situation is prevented by assuming sufficient independence and that Asm. 3.5 holds for all equivalent generators.

**Conditions on $\mathcal{Z}_{\text{supp}}$.** The regular closed condition on $\mathcal{Z}_{\text{supp}}$ in Thm. 4.3 ensures that equality between two functions on $\mathcal{Z}_{\text{supp}}$ implies equality of their derivatives, while the path-connectedness condition prevents the one-to-one correspondence between the slots of $\boldsymbol{z}$ and those of $\hat{\boldsymbol{z}}$ from changing across different $\boldsymbol{z}$ (Lachapelle et al., 2023). The aligned-connectedness condition is novel and allows one to take integrals to go from *local* to *global* disentanglement (see Appx. A.3 for more details).

## 4.2 COMPOSITIONAL GENERALIZATION

We now show how Asm. 3.5 also enables learning a model that generalizes compositionally (Defn. 2.2), i.e., that equality of $\boldsymbol{f}$ and $\hat{\boldsymbol{f}} \circ \boldsymbol{h}$ on $\mathcal{Z}_{\text{supp}}$ also implies their equality on $\mathcal{Z}_{\text{CPE}}$. As discussed in § 2, such generalization is non-trivial and requires specific restrictions on a function class. A key restriction imposed by interaction asymmetry is that interactions across slots are limited to at most $n^{\text{th}}$ order. In Thm. 4.3, this prevents $\hat{\boldsymbol{f}} \circ \boldsymbol{h}$ from modelling interactions between parts of the same ground-truth slot in different model slots. We now aim to show that limiting the interactions across slots serves the dual role of making $\boldsymbol{f}$ and $\hat{\boldsymbol{f}} \circ \boldsymbol{h}$ "predictable", such that their behavior on $\mathcal{Z}_{\text{CPE}}$ can be determined from $\mathcal{Z}_{\text{supp}}$. To do this, we will require a characterization of the form of functions with at most $n^{\text{th}}$ order interactions across slots, which we prove in Thm. C.2 to be:

$$\boldsymbol{f}(\boldsymbol{z}) = \sum_{k=1}^{K} \boldsymbol{f}^k(\boldsymbol{z}_{B_k}) + \sum_{\boldsymbol{\alpha}:|\boldsymbol{\alpha}| \leq n} \boldsymbol{c}_{\boldsymbol{\alpha}} \boldsymbol{z}^{\boldsymbol{\alpha}} . \tag{4.2}$$

where $\boldsymbol{c}_{\boldsymbol{\alpha}} \in \mathbb{R}^{d_x}$. In the first sum, slots are processed *separately* by functions $\boldsymbol{f}^k$, while in the second, they can interact more explicitly via polynomial functions of components from different slots, with degree determined by the order of interaction, $n$. With this, we can now state our result.

**Theorem 4.4** (Compositional Generalization). *Let $n \in \{0, 1, 2\}$. Let $\mathcal{Z}_{\text{supp}}$ be regular closed (Defn. A.3). Let $\boldsymbol{f} : \mathcal{Z} \to \mathcal{X}$ and $\hat{\boldsymbol{f}} : \mathcal{Z} \to \mathbb{R}^{d_x}$ be $C^3$ diffeomorphisms with at most $n^{\text{th}}$ order interactions across slots on $\mathcal{Z}$. If $\hat{\boldsymbol{f}}$ disentangles $\boldsymbol{z}$ on $\mathcal{Z}_{\text{supp}}$ w.r.t. $\boldsymbol{f}$ (Defn. 2.1), then it generalizes compositionally (Defn. 2.2).*

**Intuition.** Consider the red dotted line in Fig. 2 (left) corresponding to $\{z \in \mathbb{R}^2 \mid z_1 = z_1^*\}$. To generalize compositionally, the behavior of the partial derivative $\frac{\partial f_l}{\partial z_1}(z_1^*, z_2)$ on this line must be predictable from the behavior of $f$ on $\mathcal{Z}_{\text{supp}}$, and similarly for $\hat{f} \circ h$. Because $f$ and, as we show, $\hat{f} \circ h$ have at most $n^{\text{th}}$ order interactions across slots on $\mathbb{R}^{d_z}$, the form of this derivative is constrained to be a fixed-degree polynomial,

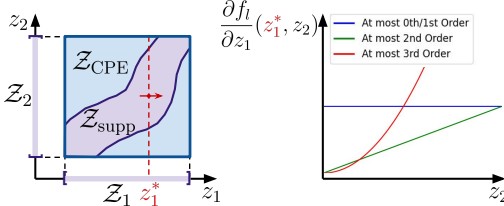

Figure 2: See intuition for Theorem 4.4.

see Eq. (4.2) and Fig. 2 (right). Thus, its global behavior on the dotted line in $\mathbb{R}^{d_z}$ can be determined from its derivative locally in a region in $\mathcal{Z}_{\text{supp}}$. Applying this reasoning to all such line segments intersecting $\mathcal{Z}_{\text{supp}}$, we can show that the behavior of $f$ and $\hat{f} \circ h$ on $\mathcal{Z}_{\text{CPE}}$ can be determined from $\mathcal{Z}_{\text{supp}}$.

## 4.3 UNIFYING AND EXTENDING PRIOR RESULTS

We now show that our theory also recovers the results of Brady et al. (2023) and Lachapelle et al. (2023) as special cases for $n = 0$ and $n = 1$, and extends them to more flexible generative processes.

**At most $0^{\text{th}}$ Order Interaction.** Brady et al. (2023) proposed two properties on $f$, which enable disentanglement and compositional generalization (Wiedemer et al., 2024b): *compositionality* (Defn. E.1) and *irreducibility* (Defn. E.2). Compositionality states that different slots affect distinct output components such that $Df(z)$ has a block-like structure. This is equivalent to $f$ having at most $0^{\text{th}}$ order interaction across slots (Defn. 3.2). Irreducibility is a rank condition on $D_{B_k}f(z)$ which Brady et al. (2023) interpreted as parts of the same object sharing information. In Thm. E.3, we show that irreducibility is equivalent to $f$ having $1^{\text{st}}$ order interaction within slots for all equivalent generators. Thus, the assumptions in Brady et al. (2023) are equivalent to interaction asymmetry for all equivalent generators when $n = 0$. Further, we recover their disentanglement result using a novel proof strategy, unified with proofs for at most $1^{\text{st}}$ / $2^{\text{nd}}$ order interaction across slots (Thm. A.20).

**At most $1^{\text{st}}$ Order Interaction.** Lachapelle et al. (2023) also proposed two assumptions on $f$ for disentanglement and compositional generalization: *additivity* (Defn. E.4) and *sufficient nonlinearity* (Defn. E.5). Additivity is equivalent to $f_l$ having a block-diagonal Hessian for all $l \in [d_x]$ (Lachapelle et al., 2023). This is the same as $f$ having at most $1^{\text{st}}$ order interaction across slots (Defn. 3.4). Sufficient nonlinearity is a linear independence condition on columns of $1^{\text{st}}$ and $2^{\text{nd}}$ derivatives of $f$. In Thm. E.6, we show that sufficient nonlinearity implies that $f$ satisfies sufficient independence for $n = 1$ and has $2^{\text{nd}}$ order interaction within slots for all equivalent generators. Further, we conjecture that the reverse implication does not hold. Thus, the assumptions of Lachapelle et al. (2023) imply, and are conjectured to be stronger than, our assumptions when $n = 1$. We also recover their same disentanglement result using a unified proof strategy (Thm. A.22).

**Allowing More Complex Interactions.** Our theory not only unifies but also extends these prior results to more general function classes. This is clear from considering the form of functions with at most $n^{\text{th}}$ order interactions across slots in Eq. (4.2). For at most $0^{\text{th}}$ (Brady et al., 2023) or $1^{\text{st}}$ order interactions (Lachapelle et al., 2023), the sum of polynomials on the RHS of (4.2) vanishes. Consequently, $f$ reduces to an *additive* function. Such generators can only model concepts with trivial interactions such as non-occluding objects. In contrast, we are able to go beyond additive interactions via the polynomial terms in (4.2). This formally corroborates the "generality" of interaction asymmetry, in that it enables more flexible generative processes where concepts can explicitly interact.

## 5 METHOD: ATTENTION-REGULARIZED TRANSFORMER-VAE

We now explore how our theoretical results in § 4 can inform the design of a practical estimation method. Our theory puts forth two key properties that a model should satisfy: (i) invertibility and (ii) limited interactions across slots of at most $n^{\text{th}}$ order. To achieve disentanglement on $\mathcal{Z}_{\text{supp}}$, (i) and (ii) must hold only "in-domain" on $\hat{\mathcal{Z}}_{\text{supp}}$ and $\mathcal{X}_{\text{supp}}$ (Thm. 4.3), while for compositional generalization, they must also hold out-of-domain, on all of $\mathcal{Z}$ and $\mathcal{X}$ (Thm. 4.4). We will focus on approaches for achieving (i) and (ii) in-domain. Achieving them out-of-domain requires addressing separate practical challenges, which are out of the scope of this work. We discuss this in detail in Appx. H.3.

**On Scalability.** Approaches that enforce (i) and (ii) *exactly* will generally only be computationally tractable in low-dimensional settings. Such computational issues are typical when translating a disentanglement result into an empirical method, often resulting in methods which directly adhere to theory but cannot scale beyond toy data (e.g., Brady et al., 2023; Gresele et al., 2021). Our core motivation, however, is learning representations of concepts underlying *high-dimensional* sensory data, such as images. Thus, to formulate a method which scales to such settings, we do not restrict ourselves to approaches which exactly enforce (i) and (ii) and also explore *approximate* approaches.

**(i) Invertibility.** Our theory requires invertibility between $\mathcal{X}_{\text{supp}} \subseteq \mathbb{R}^{d_x}$ and $\hat{\mathcal{Z}}_{\text{supp}} \subseteq \mathcal{Z} = \mathbb{R}^{d_z}$. For most settings of interest, the observed dimension $d_x$ exceeds the ground-truth latent dimension $d_z$. Thus, we generally cannot use models which are invertible by construction such as normalizing flows (Papamakarios et al., 2021). An alternative is to use an *autoencoder* in which $\hat{f}^{-1}$ and $\hat{f}$ are parameterized separately by an *encoder* $\hat{g} : \mathbb{R}^{d_x} \to \mathbb{R}^{d_{\hat{z}}}$ and a *decoder* $\hat{f} : \mathbb{R}^{d_{\hat{z}}} \to \mathbb{R}^{d_x}$, which are trained to invert each other (on $\hat{\mathcal{Z}}_{\text{supp}}$ and $\mathcal{X}_{\text{supp}}$) by minimizing a reconstruction loss $\mathcal{L}_{\text{rec}} := \mathbb{E}\|\boldsymbol{x} - \hat{f}(\hat{g}(\boldsymbol{x}))\|^2$. Minimizing $\mathcal{L}_{\text{rec}}$ alone, however, does not suffice unless the inferred latent dimension $d_{\hat{z}}$ equals the ground-truth $d_z$. Yet, in practice $d_z$ is unknown. Moreover, choosing $d_{\hat{z}} > d_z$ is important for scalability (Sajjadi et al., 2022a). A viable alternative is thus to employ a soft constraint where $d_{\hat{z}} > d_z$, but the model is encouraged to encode $\boldsymbol{x}$ using minimal latent dimensions. To achieve this, we leverage the well known VAE loss (Kingma and Welling, 2014), which couples $\mathcal{L}_{\text{rec}}$ with a KL-divergence loss $\mathcal{L}_{\text{KL}}$ between a factorized posterior $q(\hat{z}|\boldsymbol{x})$ and prior distribution $p(\hat{z})$, i.e., $\mathcal{L}_{\text{KL}} := \sum_{i \in [d_{\hat{z}}]} D_{\text{KL}}(q(\hat{z}_i|\boldsymbol{x})\|p(\hat{z}_i))$. This loss encourages each $\hat{z}_i$ to be insensitive to changes in $\boldsymbol{x}$ such that unnecessary dimensions should contain no information about $\boldsymbol{x}$ (Rolinek et al., 2019).

**(ii) At Most $n^{\text{th}}$ Order Interactions.** One approach to enforce at most $n^{\text{th}}$ order interactions across slots would be to parameterize the decoder $\hat{f}$ to match the form of such functions (see Thm. C.2) for some fixed $n$. However, this can result in an overly restrictive inductive bias and limit scalability. Moreover, $n$ is generally unknown. Thus, a more promising approach is to *regularize* interactions to be *minimal*. Doing this naively though using gradient descent would require computing gradients of high-order derivatives, which is intractable beyond toy data. This leads to the question: Is there a scalable architecture which permits efficient regularization of the interactions across slots?

**Transformers for Interaction Regularization.** We make the observation that the *Transformer* architecture (Vaswani et al., 2017) provides an efficient means to approximately regularize interactions. In a Transformer, slots are only permitted to interact via an *attention mechanism*. We will focus on a *cross-attention* mechanism, which maps a latent vector $\hat{\boldsymbol{z}}$ to output $\hat{x}_l$ (e.g., a pixel) via:

$$\boldsymbol{K} = \boldsymbol{W}^K[\hat{\boldsymbol{z}}_{B_1} \cdots \hat{\boldsymbol{z}}_{B_K}], \qquad \boldsymbol{V} = \boldsymbol{W}^V[\hat{\boldsymbol{z}}_{B_1} \cdots \hat{\boldsymbol{z}}_{B_K}], \quad \boldsymbol{Q} = \boldsymbol{W}^Q[\boldsymbol{o}_1 \cdots \boldsymbol{o}_{d_x}], \quad (5.1)$$

$$\boldsymbol{A}_{l,k} = \frac{\exp\left(\boldsymbol{Q}_{:,l}^\top \boldsymbol{K}_{:,k}\right)}{\sum_{i \in [K]} \exp\left(\boldsymbol{Q}_{:,l}^\top \boldsymbol{K}_{:,i}\right)}, \quad \bar{\boldsymbol{x}}_l = \boldsymbol{A}_{l,:}\boldsymbol{V}^\top, \qquad \hat{x}_l = \psi(\bar{\boldsymbol{x}}_l). \quad (5.2)$$

In Eq. (5.1), all slots are assumed to have equal size, and key $\boldsymbol{K}_{:,k}$ and value $\boldsymbol{V}_{:,k}$ vectors are computed for each slot $k \in [K]$. Query vectors are computed for output dimensions $l \in [d_x]$ (e.g., pixel coordinates) and each $l$ is assigned a fixed vector $\boldsymbol{o}_l$. In Eq. (5.2), queries and keys are used to compute attention weights $\boldsymbol{A}_{l,k}$. These weights determine the slots pixel $l$ "attends" to when generating pixel token $\bar{\boldsymbol{x}}_l$, which is mapped to a pixel $\hat{x}_l$ by nonlinear function $\psi$; see Appx. F for further details.

Within cross-attention, interactions across slots occur if the query vector for a pixel $l$ attends to multiple slots, i.e., if $\boldsymbol{A}_{l,k}$ is non-zero for more than one $k$. Conversely, if $\boldsymbol{A}_{l,k}$ is non-zero for only one $k$, then, intuitively, no interactions should occur. This intuition can be corroborated formally by computing the Jacobian of cross-attention w.r.t. each slot (see Appx. F.1). Thus, an approximate means to minimize interactions across slots is to regularize $\boldsymbol{A}$ towards having only one non-zero entry for each row $\boldsymbol{A}_{l,:}$. To this end, we propose to minimize the sum of all pairwise products $\boldsymbol{A}_{l,j}\boldsymbol{A}_{l,k}$, where $j \neq k$ (see Fig. 4). This quantity is non-negative and will only be zero when each row of $\boldsymbol{A}$ has exactly one non-zero entry. This resembles the *compositional contrast* of Brady et al. (2023), but computed on $\boldsymbol{A}$, which can be efficiently optimized, as opposed to the Jacobian of $\hat{f}$ which is intractable to optimize. We refer to this regularizer as $\mathcal{L}_{\text{interact}}$, see Eq. (F.9).

**Model.** Combining these different objectives leads us to the following weighted three-part-loss:

$$\mathcal{L}_{\text{disent}}(\hat{f}, \hat{g}, \boldsymbol{x}) = \mathcal{L}_{\text{rec}} + \alpha\mathcal{L}_{\text{interact}} + \beta\mathcal{L}_{\text{KL}}, \qquad (5.3)$$

Figure 3: **(A) Sprites** Normalized slot-wise Jacobians for an unregularized ($\alpha = 0, \beta = 0$) and a regularized ($\alpha > 0, \beta > 0$) Transformer and a Spatial Broadcast Decoder (SBD). The unregularized model encodes objects across multiple slots, while the regularized model matches the disentanglement of the SBD. **(B) CLEVR6** Slot-wise Jacobians for a regularized Transformer and a SBD on objects in CLEVR6 which interact via reflections. As can be seen in reconstructions and Jacobians, the regularized Transformer models reflections, while mostly removing unnecessary interactions, while the SBD fails to model reflections due to its restricted architecture.

We apply this loss to a flexible Transformer-based autoencoder, similar to the models of Jabri et al. (2023); Jaegle et al. (2022); Sajjadi et al. (2022b). For the encoder $\hat{g}$, we first map data $x$ to features using the CNN of Locatello et al. (2020b). These features are processed by a Transformer, which has both self- and cross-attention at every layer, yielding a representation $\hat{z}$. Our decoder $\hat{f}$ then maps $\hat{z}$ to an output $\hat{x}$ using a cross-attention Transformer regularized with $\mathcal{L}_{\text{interact}}$, see Appx. J for details.

**Relationship to Models In Object-Centric Learning.** Existing models for learning disentangled representations of concepts, particularly for disentangling objects without supervision, typically rely on architectural priors rather than regularization (Greff et al., 2019; Locatello et al., 2020b; Seitzer et al., 2023; Singh et al., 2022a). While such priors promote disentanglement, they are often too restrictive. For example, Spatial Broadcast Decoders (Watters et al., 2019b) decode slots separately and only allow for weak interaction through a softmax function, which prevents modelling real-world data where objects exhibit more complex interactions (Singh et al., 2022a). While some works have shown success in disentangling objects using more powerful Transformer decoders (Sajjadi et al., 2022a; Singh et al., 2022a;b), they rely on encoders that use Slot Attention (Locatello et al., 2020b) as an architectural component, which differs from current large-scale models, typically based on Transformers (Anil et al., 2023). In contrast, we explore the more flexible approach of starting with a very general Transformer-based model and regularizing it towards a more constrained model.

## 6 EXPERIMENTS

We now apply our attention-regularized Transformer-VAE (§ 5) for learning representations of concepts. Since this model is designed to enforce the criteria outlined in Thm. 4.3 for disentanglement on $\mathcal{Z}_{\text{supp}}$, we focus on evaluating disentanglement, as opposed to compositional generalization. To this end, we focus on disentangling objects in visual scenes, and leave an empirical study of a wider range of concepts (e.g., attributes, object-parts, events) for future work (see Appx. J for details).

**Data.** We consider two multi-object datasets in our experiments. The first, which we refer to as Sprites (Brady et al., 2023; Watters et al., 2019a; Wiedemer et al., 2024b), consist of images with 2–4 objects set against a black background. The second is the dataset (Johnson et al., 2017), consisting of images with 2–6 objects. In Sprites, objects do not have reflections and rarely occlude such that slots have essentially have no interaction. In CLEVR6, however, objects can cast shadows and reflect upon each other (see Fig. 1 for an example), introducing more complex interactions.

**Metrics.** A common metric for object disentanglement is the Adjusted-Rand Index (ARI; Hubert and Arabie, 1985). The ARI measures the similarity between the set of pixels encoded by a model slot, and the set of ground-truth pixels for a given object in a scene, yielding an optimal score if each slot corresponds to exactly one object. To assign a pixel to a unique model slot, prior works typically choose the slot with the largest attention score (from, e.g., Slot Attention) for that pixel (Seitzer et al., 2023). However, using attention scores can make model comparisons challenging and is also somewhat unprincipled (see Appx. J.2). We thus consider an alternative and compute the ARI using the Jacobian of a decoder (J-ARI). Specifically, we assign a pixel $l$ to the slot with the largest $L_1$ norm for the slot-wise Jacobian $D_{B_k}\hat{f}_l(\hat{z})$ (see Fig. 3 for a visualization of these Jacobians).

While J-ARI indicates which slots are most responsible for encoding each object, it does not indicate if additional slots affect the same object, i.e., $\|D_{B_k}\hat{f}_l(\hat{z})\|_1 \neq 0$ for more than one $k$. To measure

Table 1: **Empirical Results.** We show the mean $\pm$ std. dev. for J-ARI and JIS (in %) over 3 seeds for different choices of encoders and decoders and weights of the loss terms in Eq. (5.3) on Sprites and CLEVR6.

| Model | | | Sprites | | CLEVR6 | |
|---|---|---|---|---|---|---|
| Encoder | Decoder | Loss | J-ARI ($\uparrow$) | JIS ($\uparrow$) | J-ARI ($\uparrow$) | JIS ($\uparrow$) |
| Slot Attention | Spatial-Broadcast | $\alpha = 0, \beta = 0$ | $89.3 \pm 1.5$ | $91.4 \pm 0.8$ | $\mathbf{97.0 \pm 0.2}$ | $\mathbf{95.3 \pm 0.7}$ |
| Slot Attention | Transformer | $\alpha = 0, \beta = 0$ | $90.1 \pm 1.4$ | $73.6 \pm 1.5$ | $95.5 \pm 1.0$ | $63.1 \pm 1.0$ |
| Transformer | Transformer | $\alpha = 0, \beta = 0$ | $80.5 \pm 4.1$ | $57.0 \pm 8.0$ | $92.7 \pm 3.3$ | $54.8 \pm 3.5$ |
| Transformer | Transformer | $\alpha = 0.05, \beta = 0$ | $82.8 \pm 3.6$ | $73.8 \pm 4.0$ | $79.2 \pm 12.8$ | $51.6 \pm 5.9$ |
| Transformer | Transformer | $\alpha = 0, \beta = 0.05$ | $92.6 \pm 2.0$ | $92.8 \pm 0.9$ | $96.6 \pm 0.3$ | $80.3 \pm 0.4$ |
| Transformer | Transformer | $\alpha = 0.05, \beta = 0.05$ **(Ours)** | $\mathbf{93.7 \pm 0.6}$ | $\mathbf{95.0 \pm 1.7}$ | $96.5 \pm 0.4$ | $83.8 \pm 1.1$ |

this, we also introduce the Jacobian Interaction Score (JIS). JIS is computed by taking the maximum of $\|D_{B_k}\hat{f}_l(\hat{z})\|_1$ across slots after normalization, averaged over all pixels. If each pixel is affected by only one slot, JIS is $1$. For datasets where objects essentially do not interact like Sprites, JIS should be close to $1$, whereas for CLEVR6, it should be as high as possible while maintaining invertibility.

## 6.1 RESULTS

$\mathcal{L}_{\text{disent}}$ **Enables Object Disentanglement.** In Tab. 1, we compare the J-ARI and JIS of our regularized Transformer-based model ($\alpha > 0, \beta > 0$) trained with $\mathcal{L}_{\text{disent}}$ (Eq. (5.3)) to the same model trained without regularization ($\alpha = 0, \beta = 0$), i.e., with only $\mathcal{L}_{\text{rec}}$. On Sprites, the regularized model achieves notably higher scores for both J-ARI and JIS. This is corroborated by visualizing the slot-wise Jacobians in Fig. 3A, where we see the regularized model cleanly disentangles objects, whereas the unregularized model often encodes objects across multiple slots. Similarly, on CLEVR6, the regularized model achieves superior disentanglement, as indicated by the higher values for both metrics.

**Comparison to Existing Object-Centric Autoencoders.** In Tab. 1, we also compare our model to existing models using encoders with Slot Attention and Spatial Broadcast Decoders (SBDs). On Sprites, our model achieves higher J-ARI and JIS than these models, despite using a weaker architectural prior. On CLEVR6, our model outperforms Slot Attention with a Transformer decoder in terms of J-ARI and JIS. Models using a SBD, however, achieve a higher and nearly perfect JIS, i.e., the learned slots essentially never affect the same pixel. In Fig 3B, we see this comes at the cost of SBDs failing to model reflections between objects, while our model captures this interaction. This highlights that regularizing a flexible architecture with $\mathcal{L}_{\text{disent}}$ can enable a better balance between restricting interactions and model expressivity.

**Ablation Over Losses.** Lastly, in Tab. 1, we ablate the impact of the regularizers in $\mathcal{L}_{\text{disent}}$. Training without $\mathcal{L}_{\text{KL}}$ ($\alpha > 0, \beta = 0$) can in some cases give improvements in J-ARI and JIS over an unregularized model ($\alpha = 0, \beta = 0$). However, across datasets this loss yields worse disentanglement than $\mathcal{L}_{\text{disent}}$ ($\alpha > 0, \beta > 0$). This highlights that penalizing latent capacity via $\mathcal{L}_{\text{KL}}$ is important for object disentanglement. Training without $\mathcal{L}_{\text{interac}}$ ($\alpha = 0, \beta > 0$) generally yields a drop across both metrics compared to $\mathcal{L}_{\text{disent}}$, though on CLEVR6 this loss achieves a comparable J-ARI. We found that training with $\mathcal{L}_{\text{KL}}$ can, in some cases, implicitly minimize $\mathcal{L}_{\text{interac}}$, explaining this result (Fig. 5).

**More Complex Data.** Tab. 2 in Appx. I presents additional results for the visually complex CLEVR-Tex dataset (Karazija et al., 2021). For these experiments, we follow Seitzer et al. (2023) and reconstruct image representations based on a pre-trained encoder rather than the original images. We find our model to achieve superior J-ARI compared to an unregularized Transformer and a Slot Attention baseline, but slot-wise MLP decoders yield higher JIS. For further details, see Appx. I.

## 7 CONCLUSION

In this work, we proposed interaction asymmetry as a general principle for learning disentangled and composable representations. Formalizing this idea led to a constraint on the partial derivatives of the generator function, which unifies assumptions from prior efforts and extends their results to a more flexible class of generators that allow for non-trivial interactions. These theoretical insights inspired the development of a flexible estimation method based on the Transformer architecture with a novel cross-attention regularizer, which can be efficiently implemented at scale, and which shows promising results on object-centric learning datasets. Future work should seek to further extend our theoretical results, address the empirical challenges for achieving compositional generalization, and test our method on more large-scale data involving not only objects but also other types of concepts.

ACKNOWLEDGEMENTS

JB would like to thank Nicoló Zottino for providing the initial idea (expressed in Fig. 2) that compositional generalization could be extended to functions with at most $n^{\text{th}}$ order interactions. The authors thank Luigi Gresele and the Robust Machine Learning Group for helpful discussions.

This work was supported by the German Federal Ministry of Education and Research (BMBF): Tübingen AI Center, FKZ: 01IS18039A, 01IS18039B. WB acknowledges financial support via an Emmy Noether Grant funded by the German Research Foundation (DFG) under grant no. BR 6382/1-1 and via the Open Philantropy Foundation funded by the Good Ventures Foundation. WB is a member of the Machine Learning Cluster of Excellence, EXC number 2064/1 – Project number 390727645. This work utilized compute resources at the Tübingen Machine Learning Cloud, DFG FKZ INST 37/1057-1 FUGG. Part of this work was done while JvK was affiliated with the Max Planck Institute for Intelligent Systems, Tübingen.

AUTHOR CONTRIBUTIONS

JB initiated and led the project. JB wrote the manuscript in close collaboration with JvK, and with feedback from all authors. JB conceived of the framework of interaction asymmetry and then formalized it together with SL and JvK. JB proved the disentanglement results in Appx. A in collaboration with SL, who improved the clarity of several results and simplified some arguments. SL extended these results from local to global disentanglement in Appx. A.3. SB proved the characterization of functions with at most $n^{\text{th}}$ order interactions in Appx. C, based on an earlier weaker result for analytic functions by JvK. SB proved the compositional generalization results in Appx. D after discussions with JB. JB proved the results for unifying prior works in Appx. E. SB carried out the Jacobian computation in Appx. F.1. JB conceived of the regularizers and method in § 5 and implemented and executed all experiments in § 6 with advising from TK. WB created Fig. 1 and Fig. 3 with insight from JB and feedback from JvK. SL created Fig. 2 based on an outline provided by JB.

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

# Appendices

## Table of Contents

# A DISENTANGLEMENT PROOFS

## A.1 ADDITIONAL DEFINITIONS AND LEMMAS

**Definition A.1** ($C^k$-diffeomorphism). Let $A \subseteq \mathbb{R}^n$ and $B \subseteq \mathbb{R}^m$. A map $\boldsymbol{f} : A \to B$ is said to be a $C^k$-diffeomorphism if it is bijective, $C^k$ and has a $C^k$ inverse.

*Remark* A.2. The property of being differentiable is usually defined only for functions with an open domain of $\mathbb{R}^n$. Note that, in the definition above, both $A$ and $B$ might not be open sets in their respective topologies. For an arbitrary domain $A \subseteq \mathbb{R}^n$, we say that a function $\boldsymbol{f}$ is $C^k$ if it can be extended to a $C^k$ function defined on an open set $U$ containing $A$. More precisely, $\boldsymbol{f} : A \to B$ is $C^k$ if there exists a function $\boldsymbol{g} : U \to \mathbb{R}^m$ such that 1) $U$ is an open set containing $A$, 2) $\boldsymbol{g}$ is $C^k$, and 3) $\boldsymbol{g}(\boldsymbol{a}) = \boldsymbol{f}(\boldsymbol{a})$ for all $\boldsymbol{a} \in A$. See p.199 of Munkres (1991) for details about such constructions.

**Definition A.3** (Regular closed sets). A set $\mathcal{Z}_{\text{supp}} \subseteq \mathbb{R}^{d_z}$ is regular closed if $\mathcal{Z}_{\text{supp}} = \overline{\mathcal{Z}_{\text{supp}}^{\circ}}$, i.e. if it is equal to the closure of its interior (in the standard topology of $\mathbb{R}^n$).

**Lemma A.4** (Lachapelle et al. (2023)). *Let $A, B \subset \mathbb{R}^n$ and suppose there exists an homeomorphism $\boldsymbol{f} : A \to B$. If $A$ is regular closed in $\mathbb{R}^n$, we have that $B \subseteq \overline{B^{\circ}}$.*

The way we defined $C^k$ functions with arbitrary domain is such that a function can be differentiable without having a uniquely defined derivative everywhere on its domain. This happens when the derivative of two distinct extensions differ. The following Lemma states that the derivative of a $C^k$ function is uniquely defined on the closure of the interior of its domain.

**Lemma A.5** (Lachapelle et al. (2023)). *Let $A \subseteq \mathbb{R}^n$ and $\boldsymbol{f} : A \to \mathbb{R}^m$ be a $C^k$ function. Then, its $k$ first derivatives are uniquely defined on $\overline{A^{\circ}}$ in the sense that they do not depend on the specific choice of $C^k$ extension.*

**Notation** For a subset $S \subseteq [d_z]$ and a matrix $\boldsymbol{A} \in \mathbb{R}^{m \times n}$, $\boldsymbol{A}_S$ will denote the sub-matrix consisting of the columns in $\boldsymbol{A}$ indexed by $S$ i.e. $\boldsymbol{A}_S = [\boldsymbol{A}_{.,i}]_{i \in S}$. Similarly, for a vector $\boldsymbol{z}$, $\boldsymbol{z}_S$ will denote the sub-vector of $\boldsymbol{z}$ consisting of components indexed by $S$ i.e. $\boldsymbol{z}_S := (z_i)_{i \in S}$.

**Lemma A.6.** *Let $\boldsymbol{A} \in \mathbb{R}^{m \times n}$ and let $\mathcal{B}$ be a partition of $[n]$. If*

$$rank(\boldsymbol{A}) = \sum_{S \in \mathcal{B}} rank(\boldsymbol{A}_S) \tag{A.1}$$

*Then $\forall \boldsymbol{z} \in \mathbb{R}^n$ s.t. $\boldsymbol{A}\boldsymbol{z} = 0$, $\boldsymbol{A}_S \boldsymbol{z}_S = 0$, for any $S \in \mathcal{B}$.*

*Proof.* Assume for a contradiction that there exist a $\boldsymbol{z} \in \mathbb{R}^n$, s.t. $\boldsymbol{A}\boldsymbol{z} = 0$, and there exist $S_1 \in \mathcal{B}$ s.t. $\boldsymbol{A}_{S_1} \boldsymbol{z}_{S_1} \neq 0$.

Now construct the matrix, denoted, $\boldsymbol{A}_{-S_1}$ consisting of all columns in $\boldsymbol{A}$ except those indexed by $S_1$, i.e.

$$\boldsymbol{A}_{-S_1} := \left[ [\boldsymbol{A}_{.,i}]_{i \in S} \right]_{S \in \mathcal{B} \setminus S_1} \tag{A.2}$$

By using (A.1) and the property that $\text{rank}([\boldsymbol{B}, \boldsymbol{C}]) \leq \text{rank}(\boldsymbol{B}) + \text{rank}(\boldsymbol{C})$, we get

$$\text{rank}(\boldsymbol{A}) = \sum_{S \in \mathcal{B} \setminus S_1} \text{rank}(\boldsymbol{A}_S) + \text{rank}(\boldsymbol{A}_{S_1}) \tag{A.3}$$

$$\geq \text{rank}(\boldsymbol{A}_{-S_1}) + \text{rank}(\boldsymbol{A}_{S_1}) \tag{A.4}$$

$$\geq \text{rank}(\boldsymbol{A}). \tag{A.5}$$

Consequently, we have that:

$$\text{rank}(\boldsymbol{A}) = \text{rank}(\boldsymbol{A}_{-S_1}) + \text{rank}(\boldsymbol{A}_{S_1}) \tag{A.6}$$

This implies that the column spaces of both matrices denoted $\text{range}(\boldsymbol{A}_{-S_1}), \text{range}(\boldsymbol{A}_{S_1})$ respectively, do not intersect, except at the zero vector.

---

A simple example of such a situation is the trivial function $f : \{0\} \to \{0\}$ which is differentiable at 0 but does not have a well defined derivative because $g(x) = x$ and $h(x) = -x$ are both differentiable extensions of $f$ but have different derivatives at $x = 0$.

Now we know that at $\boldsymbol{z}$

$$0 = \boldsymbol{A}\boldsymbol{z} \tag{A.7}$$

$$= \boldsymbol{A}_{-S_1}\boldsymbol{z}_{-S_1} + \boldsymbol{A}_{S_1}\boldsymbol{z}_{S_1} \tag{A.8}$$

Consequently,

$$\boldsymbol{A}_{-S_1}\boldsymbol{z}_{-S_1} = -\boldsymbol{A}_{S_1}\boldsymbol{z}_{S_1} \tag{A.9}$$

and by our assumed contradiction we know that at $\boldsymbol{z}$:

$$\boldsymbol{A}_{S_1}\boldsymbol{z}_{S_1} \neq 0 \tag{A.10}$$

This implies that the column spaces of $\boldsymbol{A}_{-S_1}, \boldsymbol{A}_{S_1}$ must intersect at a point other than the zero vector, which is a contradiction. $\qquad \square$

**Lemma A.7.** *Let $\boldsymbol{A} \in \mathbb{R}^{d \times d}$ be an invertible matrix and $\{B_1, \ldots, B_K\}$ be a partition of $[d]$. Assume there are $k_1, k_2, k \in [K]$ such that:*

$$\boldsymbol{A}_{B_k, B_{k_1}} \neq 0 \neq \boldsymbol{A}_{B_k, B_{k_2}} \tag{A.11}$$

*Then there exists a subset $S \subset [d]$ with cardinality $|B_k|$ that has the following properties:*

1. *The sub-block $\boldsymbol{A}_{B_k, S}$ is invertible.*

2. *$S \nsubseteq B_{k'}$, for any $k' \in [K]$*

*Proof.* We first prove that there must exists an $S$ satisfying point 1. Since $\boldsymbol{A}$ is invertible, each subset of rows is linearly independent and thus $\operatorname{rank}(\boldsymbol{A}_{B_k,:}) = |B_k|$. This implies that there exist a set $S \subset [d]$ with cardinality $|B_k|$ such that $\forall i \in S, \boldsymbol{A}_{B_k, i}$ are linearly independent, and thus form a basis of $\mathbb{R}^{|B_k|}$.

If $S \nsubseteq B_{k'}$ for all $k' \in [K]$, we are done.

We consider the case where there exists a $k'$ such that $S \subseteq B_{k'}$. We will show that we can construct a different $S^*$ from $S$ which satisfies both conditions.

We know by (A.11) that there exist a second block $k^* \neq k'$ such that for some $j^* \in B_{k^*}$, $\boldsymbol{A}_{B_k, j^*} \neq 0$. Since $\{\boldsymbol{A}_{B_k, i}\}_{i \in S}$ forms a basis of $\mathbb{R}^{|B_k|}$, the vector $\boldsymbol{A}_{B_k, j^*}$ can be represented uniquely as

$$\boldsymbol{A}_{B_k, j^*} = \sum_{i \in S} a_i \boldsymbol{A}_{B_k, i}, \tag{A.12}$$

where $a_i \in \mathbb{R}$ for all $i$. Because $\boldsymbol{A}_{B_k, j^*} \neq 0$, there exists $j \in S$ such that $\boldsymbol{a}_j \neq 0$. Because this representation is unique, we know that $\boldsymbol{A}_{B_k, j^*}$ is outside the span of $\{\boldsymbol{A}_{B_k, i}\}_{i \in S \setminus \{j\}}$. This means that, by taking $S^* := (S \setminus \{j\}) \cup \{j^*\}$, we have that $\{\boldsymbol{A}_{B_k, i}\}_{i \in S^*}$ is a basis for $\mathbb{R}^{|B_k|}$ or, in other words, $\boldsymbol{A}_{B_k, S^*}$ is invertible. Also, $S^*$ is not included in a single block since $S \setminus \{j\} \subseteq B_{k'}$ and $j^* \in B_{k^*}$ with $k' \neq k^*$. $\qquad \square$

## A.2 SUFFICIENT INDEPENDENCE ASSUMPTIONS

**Definition A.8** (Sufficient Independence ($0^{\text{th}}$ Order)). Let $\boldsymbol{f} : \mathbb{R}^{d_z} \to \mathbb{R}^{d_x}$ be a $C^1$ function with $0^{\text{th}}$ order interactions between slots (Def. 3.2). The function $\boldsymbol{f}$ is said to have *sufficiently independent* derivatives if $\forall \boldsymbol{z} \in \mathbb{R}^{d_z}$:

$$\operatorname{rank}\left(\left[[D_i \boldsymbol{f}(\boldsymbol{z})]_{i \in B_k}\right]_{k \in [K]}\right) = \sum_{k \in [K]} \operatorname{rank}\left([D_i \boldsymbol{f}(\boldsymbol{z})]_{i \in B_k}\right) \tag{A.13}$$

**Definition A.9** (Sufficient Independence ($1^{\text{st}}$ Order)). Let $\boldsymbol{f} : \mathbb{R}^{d_z} \to \mathbb{R}^{d_x}$ be a $C^2$ function with *at most $1^{\text{st}}$ order interactions* between slots (Def. 3.3). The function $\boldsymbol{f}$ is said to have *sufficiently independent* derivatives if $\forall \boldsymbol{z} \in \mathbb{R}^{d_z}$:

$$\operatorname{rank}\left(\left[\left[D_i \boldsymbol{f}(\boldsymbol{z})\right]_{i \in B_k} \left[D_{i,i'}^2 \boldsymbol{f}(\boldsymbol{z})\right]_{(i,i') \in B_k^2}\right]_{k \in [K]}\right)$$

$$= \sum_{k \in [K]} \left[\operatorname{rank}\left([D_i \boldsymbol{f}(\boldsymbol{z})]_{i \in B_k}\right) + \operatorname{rank}\left(\left[D_{i,i'}^2 \boldsymbol{f}(\boldsymbol{z})\right]_{(i,i') \in B_k^2}\right)\right]$$

**Definition 4.2** (Sufficient Independence ($2^{\text{nd}}$ Order)). A $C^3$ function $\boldsymbol{f} : \mathbb{R}^{d_z} \to \mathbb{R}^{d_x}$ with at most $2^{\text{nd}}$ order interactions across slots is said to have *sufficiently independent* derivatives if $\forall \boldsymbol{z} \in \mathbb{R}^{d_z}$:

$$\text{rank}\left(\left[\left[D_i \boldsymbol{f}(\boldsymbol{z})\right]_{i \in B_k} \left[D^2_{i,i'} \boldsymbol{f}(\boldsymbol{z})\right]_{i \in B_k, i' \in [d_z]} \left[D^3_{i,i',i''} \boldsymbol{f}(\boldsymbol{z})\right]_{(i,i',i'') \in B^3_k}\right]_{k \in [K]}\right)$$

$$= \sum_{k \in [K]} \left[\text{rank}\left(\left[\left[D_i \boldsymbol{f}(\boldsymbol{z})\right]_{i \in B_k} \left[D^2_{i,i'} \boldsymbol{f}(\boldsymbol{z})\right]_{i \in B_k, i' \in [d_z]}\right]\right) + \text{rank}\left(\left[D^3_{i,i',i''} \boldsymbol{f}(\boldsymbol{z})\right]_{(i,i',i'') \in B^3_k}\right)\right] .$$

### A.3 FROM LOCAL TO GLOBAL DISENTANGLEMENT

This section takes care of technical subtleties when one has to go from local to global disentanglement. The disentanglement guarantee of this work is proven by first showing that $D\boldsymbol{h}$, i.e. the Jacobian of $\boldsymbol{h} := \boldsymbol{f}^{-1} \circ \hat{\boldsymbol{f}}$, has a block-permutation structure everywhere, and from there showing that $\boldsymbol{h}$ can be written as $\boldsymbol{h}(\boldsymbol{z}) = (\boldsymbol{h}_1(\boldsymbol{z}_{B_{\pi(1)}}), \boldsymbol{h}_2(\boldsymbol{z}_{B_{\pi(2)}}), \dots, \boldsymbol{h}_K(\boldsymbol{z}_{B_{\pi(K)}}))$ (see Defintion 2.1). Lachapelle et al. (2023) refers to the first condition on $D\boldsymbol{h}$ as *local disentanglement* and the second condition on $\boldsymbol{h}$ as *global disentanglement*, the latter of which corresponds to the definition of disentanglement employed in the present work. The authors also show that going from local to global disentanglement requires special care when considering very general supports $\mathcal{Z}_{\text{supp}}$, like we do in this work, as opposed to the more common assumption that $\mathcal{Z}_{\text{supp}} := \mathbb{R}^{d_z}$ which makes this step more direct (e.g., see Hyvärinen et al. (2019)). This section reuses definitions and lemmata taken from Lachapelle et al. (2023) and introduces a novel sufficient condition on the support of the latent factors, we named *aligned-connectedness*, to guarantee that the jump from local to global disentanglement can be made.

**Definition A.10** (Partition-respecting permutations). Let $\mathcal{B} := \{B_1, B_2, \dots, B_K\}$ be a partition of $\{1, ..., d\}$. A permutation $\pi$ over $\{1, ..., d\}$ respects $\mathcal{B}$ if, for all $B \in \mathcal{B}$, $\pi(B) \in \mathcal{B}$.

**Definition A.11** ($\mathcal{B}$-block permutation matrices). A matrix $\boldsymbol{A} \in \mathbb{R}^{d \times d}$ is a $\mathcal{B}$-block permutation matrix if it is invertible and can be written as $\boldsymbol{A} = \boldsymbol{C}\boldsymbol{P}_\pi$ where $\boldsymbol{P}_\pi$ is the matrix representing the $\mathcal{B}$-respecting permutation $\pi$ (Definition A.10), i.e. $P_\pi \boldsymbol{e}_i = \boldsymbol{e}_{\pi(i)}$, and $\boldsymbol{C} \in \mathbb{R}^{d \times d}$ is such that for all distinct blocks $B, B' \in \mathcal{B}$, $\boldsymbol{C}_{B,B'} = 0$.

**Proposition A.12.** *The inverse of a $\mathcal{B}$-block permutation matrix is also a $\mathcal{B}$-block permutation matrix.*

*Proof.* First note that $\boldsymbol{C}$ must be invertible, otherwise $\boldsymbol{A}$ is not. Also, $\boldsymbol{C}^{-1}$ must also be such that $(\boldsymbol{C}^{-1})_{B,B'} = 0$ for all distinct blocks $B, B' \in \mathcal{B}$. This is because, without loss of generality, we can assume the blocks of $\mathcal{B}$ are contiguous which implies that $\boldsymbol{C}$ is a block diagonal matrix so that $\boldsymbol{C}^{-1}$ is also block diagonal. Since $\pi$ preserves $\mathcal{B}$, we have that $\boldsymbol{P}_\pi^\top \boldsymbol{C}^{-1} \boldsymbol{P}_\pi$ is also block diagonal since, for all distinct $B, B' \in \mathcal{B}$, $(\boldsymbol{P}_\pi^\top \boldsymbol{C}^{-1} \boldsymbol{P}_\pi)_{B,B'} = (\boldsymbol{C}^{-1})_{\pi(B),\pi(B')} = 0$, where we used the fact that the blocks $\pi(B)$ and $\pi(B')$ are in $\mathcal{B}$, because $\pi$ is $\mathcal{B}$-preserving, and are distinct, because $\pi$ is a bijection. We can thus see that

$$\boldsymbol{A}^{-1} = \boldsymbol{P}_\pi^\top \boldsymbol{C}^{-1}$$
$$= \boldsymbol{P}_\pi^\top \boldsymbol{C}^{-1} \boldsymbol{P}_\pi \boldsymbol{P}_\pi^\top$$
$$= \tilde{\boldsymbol{C}} \boldsymbol{P}_\pi^\top$$
$$= \tilde{\boldsymbol{C}} \boldsymbol{P}_{\pi^{-1}} ,$$

where $\tilde{\boldsymbol{C}} := \boldsymbol{P}_\pi^\top \boldsymbol{C}^{-1} \boldsymbol{P}_\pi$ is block diagonal and $\pi^{-1}$ is block-preserving. $\square$

**Definition A.13** (Local disentanglement; Lachapelle et al. (2023)). A learned decoder $\hat{\boldsymbol{f}} : \mathbb{R}^{d_z} \to \mathbb{R}^{d_x}$ is said to be locally disentangled w.r.t. the ground-truth decoder $\boldsymbol{f}$ when $\hat{\boldsymbol{f}} \circ \boldsymbol{h}(\boldsymbol{z}) = \boldsymbol{f}(\boldsymbol{z})$ for all $\boldsymbol{z} \in \mathcal{Z}_{\text{supp}}$ where the mapping $\boldsymbol{h}$ is a diffeomorphism from $\mathcal{Z}_{\text{supp}}$ onto its image satisfying the following property: for all $\boldsymbol{z} \in \mathcal{Z}_{\text{supp}}$, $D\boldsymbol{h}(\boldsymbol{z})$ is a block-permutation matrix respecting $\mathcal{B} := \{B_1, \dots, B_K\}$.

Note that, in the above definition, the permutation of the blocks might change from one $\boldsymbol{z}$ to another (see Example 5 in Lachapelle et al. (2023)). To prevent this possibility, we will assume that $\mathcal{Z}_{\text{supp}}$ is *path-connected*:

**Definition A.14** (Path-connected sets). A set $\mathcal{Z}_{\text{supp}} \subseteq \mathbb{R}^{d_z}$ is path-connected if for all pairs of points $\boldsymbol{z}^0, \boldsymbol{z}^1 \in \mathcal{Z}_{\text{supp}}$, there exists a continuous map $\boldsymbol{\phi} : [0, 1] \to \mathcal{Z}_{\text{supp}}$ such that $\boldsymbol{\phi}(0) = \boldsymbol{z}^0$ and $\boldsymbol{\phi}(1) = \boldsymbol{z}^1$. Such a map is called a path between $\boldsymbol{z}^0$ and $\boldsymbol{z}^1$.

The following Lemma from Lachapelle et al. (2023) can be used to show that when $\boldsymbol{h}$ is a diffeomorphism and $\mathcal{Z}_{\text{supp}}$ is path-connected, the block structure cannot change. This is due to the fact that $D\boldsymbol{h}(\boldsymbol{z})$ is invertible everywhere and a continuous function of $\boldsymbol{z}$. We restate the Lemma without proof.

**Lemma A.15** (Lachapelle et al. (2023)). *Let $\mathcal{C}$ be a path-connected topological space and let $\boldsymbol{M}$ : $\mathcal{C} \to \mathbb{R}^{d \times d}$ be a continuous function. Suppose that, for all $c \in \mathcal{C}$, $\boldsymbol{M}(c)$ is an invertible $\mathcal{B}$-block permutation matrix (Definition A.11). Then, there exists a $\mathcal{B}$-respecting permutation $\pi$ such that for all $c \in \mathcal{C}$ and all distinct $B, B' \in \mathcal{B}$, $\boldsymbol{M}(c)_{\pi(B'),B} = 0$.*

It turns out that, in general, having that $D\boldsymbol{h}$ has a constant block-permutation structure across its support $\mathcal{Z}_{\text{supp}}$ is not enough to make the jump to global disentanglement. See Example 7 from Lachapelle et al. (2023). We now propose a novel condition on the support $\mathcal{Z}_{\text{supp}}$ and will show it is sufficient to guarantee global disentanglement in Lemma A.18.

**Definition A.16** (Aligned-connected sets). A set $\mathcal{A} \subseteq \mathbb{R}^d$ is said to be *aligned-connected* w.r.t. a partition $\{B_1, B_2, \ldots, B_K\}$ if, for all $k \in [K]$ and all $\boldsymbol{a}' \in \mathcal{A}$, the set $\{\boldsymbol{a} \in \mathcal{A} \mid \boldsymbol{a}_{B_k} = \boldsymbol{a}'_{B_k}\}$ is path-connected.

*Remark* A.17 (Relation to path-connectedness). There exist sets that are path-connected but not aligned-connected and vice-versa. Example 7 from Lachapelle et al. (2023) presents a "U-shaped" support that is path-connected but not aligned-connected. Moreover, the set $\mathcal{A} := \mathcal{A}^{(1)} \cup \mathcal{A}^{(2)}$ where $\mathcal{A}^{(1)} := \{\boldsymbol{a} \in \mathbb{R}^2 \mid a_1 \geq 1, a_2 \geq 1\}$ and $\mathcal{A}^{(2)} := \{\boldsymbol{a} \in \mathbb{R}^2 \mid a_1 \leq -1, a_2 \leq -1\}$ is aligned-connected w.r.t. the partition $\mathcal{B} = \{\{1\}, \{2\}\}$ but not path-connected.

We now show how aligned-connectedness combined with path-connectedness is enough to guarantee global disentanglement from local disentanglement.

**Lemma A.18** (Local to global disentanglement). *Suppose $\boldsymbol{h}$ is a diffeomorphism from $\mathcal{Z}_{\text{supp}} \subseteq \mathbb{R}^{d_z}$ to its image and suppose $D\boldsymbol{h}(\boldsymbol{z})$ is a $\mathcal{B}$-block permutation matrix for all $\boldsymbol{z} \in \mathcal{Z}_{\text{supp}}$ (local disentanglement). If $\mathcal{Z}_{\text{supp}}$ is path-connected (Defn. A.14) and aligned-connected set (Defn. A.16), then $\boldsymbol{h}(\boldsymbol{z}) = (\boldsymbol{h}_1(\boldsymbol{z}_{B_{\pi(1)}}), \ldots, \boldsymbol{h}_1(\boldsymbol{z}_{B_{\pi(K)}}))$ for all $\boldsymbol{z} \in \mathcal{Z}_{\text{supp}}$ where the $\boldsymbol{h}_k$ are diffeomorphisms (global disentanglement).*

*Proof.* Since $\boldsymbol{h}$ is a diffeomorphism, $D\boldsymbol{h}$ is continuous and $D\boldsymbol{h}(\boldsymbol{z})$ is invertible for all $\boldsymbol{z} \in \mathcal{Z}_{\text{supp}}$. Since we also have that $\mathcal{Z}_{\text{supp}}$ is path-connected, we can apply Lemma A.15 to get that there exists a permutation $\pi : [K] \to [K]$ such that, for all $\boldsymbol{z} \in \mathcal{Z}_{\text{supp}}$ and all distinct $k, k' \in [K]$, we have $D\boldsymbol{h}(\boldsymbol{z})_{B_k, B_{\pi(k')}} = 0$. In other words, $D_{B_{\pi(k')}}\boldsymbol{h}_{B_k}(\boldsymbol{z}) = 0$. We must now show that $\boldsymbol{h}_{B_k}(\boldsymbol{z})$ depends solely on $\boldsymbol{z}_{B_{\pi(k)}}$. Consider another point $\boldsymbol{z}' \in \mathcal{Z}_{\text{supp}}$ such that $\boldsymbol{z}_{B_{\pi(k)}} = \boldsymbol{z}'_{B_{\pi(k)}}$. We will now show that $\boldsymbol{h}_{B_k}(\boldsymbol{z}) = \boldsymbol{h}_{B_k}(\boldsymbol{z}')$, i.e. changing $\boldsymbol{z}_{B_{\pi(k)}^c}$ does not influence $\boldsymbol{h}_{B_k}(\boldsymbol{z})$.

Because $\mathcal{Z}_{\text{supp}}$ is aligned-connected, there exists a continuous path $\phi : [0, 1] \to \mathcal{Z}_{\text{supp}}$ such that $\phi(0) = \boldsymbol{z}'$, $\phi(1) = \boldsymbol{z}$ and $\phi_{B_{\pi(k)}}(t) = \boldsymbol{z}_{B_{\pi(k)}} = \boldsymbol{z}'_{B_{\pi(k)}}$ for all $t \in [0, 1]$. By the fundamental

---

This lemma also holds if $\mathcal{C}$ is connected.

theorem of calculus, we have that

$$
\begin{aligned}
\boldsymbol{h}_{B_k}(\boldsymbol{z}) - \boldsymbol{h}_{B_k}(\boldsymbol{z}') &= \int_0^1 (\boldsymbol{h}_{B_k} \circ \phi)'(t)dt \\
&= \int_0^1 D\boldsymbol{h}_{B_k}(\phi(t))\phi'(t)dt \\
&= \int_0^1 \left( D_{B_{\pi(k)}}\boldsymbol{h}_{B_k}(\phi(t))\phi'_{B_{\pi(k)}}(t) + \sum_{k' \neq k} D_{B_{\pi(k')}}\boldsymbol{h}_{B_k}(\phi(t))\phi'_{B_{\pi(k')}}(t) \right) dt \\
&= \int_0^1 \left( D_{B_{\pi(k)}}\boldsymbol{h}_{B_k}(\phi(t))\boldsymbol{0} + \sum_{k' \neq k} \boldsymbol{0}\phi'_{B_{\pi(k')}}(t) \right) dt \\
&= \boldsymbol{0} \,,
\end{aligned}
$$

where we used the fact that $\phi_{B_{\pi(k)}}(t)$ is a constant function of $t$ and $D_{B_{\pi(k')}}\boldsymbol{h}_{B_k}(\boldsymbol{z}) = 0$ for distinct $k, k'$.

We conclude that, for all $k$, we can write $\boldsymbol{h}_{B_k}(\boldsymbol{z}) = \boldsymbol{h}_{B_k}(\boldsymbol{z}_{B_{\pi(k)}})$, which is the desired result.

Additionally, the functions $\boldsymbol{h}_{B_k}(\boldsymbol{z}_{B_{\pi(k)}})$ are diffeomorphisms because their Jacobians must be invertible otherwise the Jacobian of $\boldsymbol{h}$ (which is block diagonal) would not be invertible (which would violate the fact that it is a diffeomorphism). $\square$

**Contrasting with Lachapelle et al. (2023).** Instead of assuming aligned-connectedness, Lachapelle et al. (2023) assumed that the block-specific decoders, which would correspond to the $\boldsymbol{f}^k(\boldsymbol{z}_{B_k})$ in (4.2), are injective which, when combined with path-connectedness, is also enough to go from local to global disentanglement in the context of additive decoders ($n = 1$). Whether a similar strategy could be adapted for more general decoders with at most $n$th order interactions is left as future work.

## A.4   DISENTANGLEMENT (AT MOST $0^{\text{TH}}$ ORDER/NO INTERACTION)

**Lemma A.19.** *Let $\mathcal{Z}_{\text{supp}} \subseteq \mathcal{Z}$ be a regular closed set (Defn. A.3). Let $\boldsymbol{f} : \mathcal{Z} \to \mathcal{X}$ be $C^1$ and $\boldsymbol{h} : \hat{\mathcal{Z}}_{\text{supp}} \to \mathcal{Z}_{\text{supp}}$ be a diffeomorphism. Let $\hat{\boldsymbol{f}} := \boldsymbol{f} \circ \boldsymbol{h}$. If $\boldsymbol{f}$ has no interaction (Definition 3.4 with $n = 0$), then, for all $j, j' \in [d_z]$ and $\boldsymbol{z} \in \hat{\mathcal{Z}}_{\text{supp}}$, we have*

$$
D_j \hat{\boldsymbol{f}}(\boldsymbol{z}) \odot D_{j'} \hat{\boldsymbol{f}}(\boldsymbol{z}) = \boldsymbol{W}^{\boldsymbol{f}}(\boldsymbol{h}(\boldsymbol{z}))\boldsymbol{m}^{\boldsymbol{h}}(\boldsymbol{z}, (j, j')) , \tag{A.14}
$$

*where*

$$
\begin{aligned}
\boldsymbol{W}^{\boldsymbol{f}}(\boldsymbol{z}) &:= [\boldsymbol{W}_k^{\boldsymbol{f}}(\boldsymbol{z})]_{k \in [K]} \\
\boldsymbol{W}_k^{\boldsymbol{f}}(\boldsymbol{z}) &:= [D_{i_1}\boldsymbol{f}(\boldsymbol{z}) \odot D_{i_2}\boldsymbol{f}(\boldsymbol{z})]_{(i_1, i_2) \in B_k^2} \\
\boldsymbol{m}^{\boldsymbol{h}}(\boldsymbol{z}, (j, j')) &:= [\boldsymbol{m}_k^{\boldsymbol{h}}(\boldsymbol{z}, (j, j'))]_{k \in [K]} \\
\boldsymbol{m}_k^{\boldsymbol{h}}(\boldsymbol{z}, (j, j')) &:= [D_{j'}\boldsymbol{h}_{i_1}(\boldsymbol{z})D_j \boldsymbol{h}_{i_2}(\boldsymbol{z})]_{(i_1, i_2) \in B_k^2} \,.
\end{aligned}
$$

*Proof.* We have that

$$
\hat{\boldsymbol{f}}(\boldsymbol{z}) = \boldsymbol{f} \circ \boldsymbol{h}(\boldsymbol{z}), \ \forall \boldsymbol{z} \in \hat{\mathcal{Z}}_{\text{supp}} \,.
$$

Following the same line of argument as Lachapelle et al. (2023), we can use Lemma A.5 to say that the function $\hat{\boldsymbol{f}}(\boldsymbol{z}) = \boldsymbol{f} \circ \boldsymbol{h}(\boldsymbol{z})$ has well-defined derivatives on $\overline{(\hat{\mathcal{Z}}_{\text{supp}})^\circ}$. Since $\boldsymbol{h}^{-1}$ is a diffeomorphism from $\mathcal{Z}_{\text{supp}}$ (which is regular closed) to $\hat{\mathcal{Z}}_{\text{supp}}$, Lemma A.4 implies that $\hat{\mathcal{Z}}_{\text{supp}} \subseteq \overline{(\hat{\mathcal{Z}}_{\text{supp}})^\circ}$. This means that the function $\hat{\boldsymbol{f}}(\boldsymbol{z}) = \boldsymbol{f} \circ \boldsymbol{h}(\boldsymbol{z})$ has well-defined derivatives for all $\boldsymbol{z} \in \hat{\mathcal{Z}}_{\text{supp}}$.

By taking the derivative w.r.t. $\boldsymbol{z}_j$ on both sides of $\hat{\boldsymbol{f}}(\boldsymbol{z}) = \boldsymbol{f} \circ \boldsymbol{h}(\boldsymbol{z})$, we get

$$
D_j \hat{\boldsymbol{f}}(\boldsymbol{z}) = \sum_{k \in [K]} \sum_{i \in B_k} D_i \boldsymbol{f}(\boldsymbol{h}(\boldsymbol{z}))D_j \boldsymbol{h}_i(\boldsymbol{z}) \tag{A.15}
$$

We thus have that

$$
\begin{aligned}
D_j \hat{\boldsymbol{f}}(\boldsymbol{z}) D_{j'} \hat{\boldsymbol{f}}(\boldsymbol{z}) &= \left( \sum_{k_1 \in [K]} \sum_{i_1 \in B_{k_1}} D_{i_1} \boldsymbol{f}(\boldsymbol{h}(\boldsymbol{z})) D_j \boldsymbol{h}_{i_1}(\boldsymbol{z}) \right) \odot \left( \sum_{k_2 \in [K]} \sum_{i_2 \in B_{k_2}} D_{i_2} \boldsymbol{f}(\boldsymbol{h}(\boldsymbol{z})) D_{j'} \boldsymbol{h}_{i_2}(\hat{\boldsymbol{z}}) \right) \\
&= \sum_{k_1 \in [K]} \sum_{i_1 \in B_{k_1}} \sum_{k_2 \in [K]} \sum_{i_2 \in B_{k_2}} D_{i_1} \boldsymbol{f}(\boldsymbol{h}(\boldsymbol{z})) \odot D_{i_2} \boldsymbol{f}(\boldsymbol{h}(\boldsymbol{z})) D_j \boldsymbol{h}_{i_1}(\boldsymbol{z}) D_{j'} \boldsymbol{h}_{i_2}(\boldsymbol{z}) \\
&= \sum_{k_1 \in [K]} \sum_{i_1 \in B_{k_1}} \sum_{i_2 \in B_{k_1}} D_{i_1} \boldsymbol{f}(\boldsymbol{h}(\boldsymbol{z})) \odot D_{i_2} \boldsymbol{f}(\boldsymbol{h}(\boldsymbol{z})) D_j \boldsymbol{h}_{i_1}(\boldsymbol{z}) D_{j'} \boldsymbol{h}_{i_2}(\boldsymbol{z}) ,
\end{aligned}
$$

where the last equality used the fact that $\boldsymbol{f}$ has no interaction (Definition 3.2). We conclude by noticing

$$
\begin{aligned}
D_j \hat{\boldsymbol{f}}(\boldsymbol{z}) D_{j'} \hat{\boldsymbol{f}}(\boldsymbol{z}) &= \sum_{k_1 \in [K]} \sum_{(i_1, i_2) \in B_{k_1}^2} D_{i_1} \boldsymbol{f}(\boldsymbol{h}(\boldsymbol{z})) D_{i_2} \boldsymbol{f}(\boldsymbol{h}(\boldsymbol{z})) D_j \boldsymbol{h}_{i_1}(\boldsymbol{z}) D_{j'} \boldsymbol{h}_{i_2}(\boldsymbol{z}) \\
&= \boldsymbol{W}^{\boldsymbol{f}}(\boldsymbol{h}(\boldsymbol{z})) \boldsymbol{m}^{\boldsymbol{h}}(\boldsymbol{z}, (j, j')) .
\end{aligned}
$$

$\square$

**Theorem A.20.** *Let $\boldsymbol{f} : \mathcal{Z} \to \mathcal{X}$ be a $C^1$ diffeomorphism satisfying interaction asymmetry (Asm. 3.5) for all equivalent generators (Defn. 4.1) for $n = 0$. Let $\mathcal{Z}_{\mathrm{supp}} \subseteq \mathcal{Z}$ be regular closed (Defn. A.3), path-connected (Defn. A.14) and aligned-connected (Defn. A.16). A model $\hat{\boldsymbol{f}} : \mathcal{Z} \to \mathbb{R}^{d_x}$ disentangles $\boldsymbol{z}$ on $\mathcal{Z}_{\mathrm{supp}}$ w.r.t. $\boldsymbol{f}$ (Defn. 2.1) if it is (i) a $C^1$ diffeomorphism between $\hat{\mathcal{Z}}_{\mathrm{supp}}$ and $\mathcal{X}_{\mathrm{supp}}$ with (ii) at most $0^{th}$ order interactions across slots (Defn. 3.4) on $\hat{\mathcal{Z}}_{\mathrm{supp}}$.*

*Proof.* As mentioned in Section A.3, the proofs will proceed in two steps: First, we show local disentanglement (Definition A.13) and then we show (global) disentanglement via Lemma A.18. We first show local disentanglement.

**Remark:** We will use the following notation below:

$$
D_{i,j}^1 \boldsymbol{f}(\boldsymbol{z}) := D_j \boldsymbol{f}(\boldsymbol{z}) \odot D_i \boldsymbol{f}(\boldsymbol{z}) \in \mathbb{R}^m \tag{A.16}
$$

We first define the function $\boldsymbol{h} : \hat{\mathcal{Z}}_{\mathrm{supp}} \to \mathcal{Z}_{\mathrm{supp}}$ relating the latent spaces of these functions on $\hat{\mathcal{Z}}_{\mathrm{supp}}$:

$$
\boldsymbol{h} := \boldsymbol{f}^{-1} \circ \hat{\boldsymbol{f}} \tag{A.17}
$$

The function $\hat{\boldsymbol{f}}$ can then be written in terms of $\boldsymbol{f}$ and $\boldsymbol{h}$ on $\hat{\mathcal{Z}}_{\mathrm{supp}}$:

$$
\hat{\boldsymbol{f}} = \boldsymbol{f} \circ \boldsymbol{h} \tag{A.18}
$$

Because $\boldsymbol{f}, \hat{\boldsymbol{f}}$ are both $C^1$ diffeomorphism between $\mathcal{Z}_{\mathrm{supp}}, \mathcal{X}_{\mathrm{supp}}$ and $\hat{\mathcal{Z}}_{\mathrm{supp}}, \mathcal{X}_{\mathrm{supp}}$, respectively, we have that $\boldsymbol{h}$ is a $C^1$ diffeomorphism.

By Lemma A.19, for all $\boldsymbol{z} \in \hat{\mathcal{Z}}_{\mathrm{supp}}$, $j, j' \in [d_z]$, we have:

$$
D_j \hat{\boldsymbol{f}}(\boldsymbol{z}) \odot D_{j'} \hat{\boldsymbol{f}}(\boldsymbol{z}) = \boldsymbol{W}^{\boldsymbol{f}}(\boldsymbol{h}(\boldsymbol{z})) \boldsymbol{m}^{\boldsymbol{h}}(\boldsymbol{z}, (j, j')) \tag{A.19}
$$

where $\boldsymbol{w}^{\boldsymbol{f}}$ and $\boldsymbol{m}^{\boldsymbol{h}}$ are defined in Lemma A.19.

Define the sets

$$
\mathcal{D} := \bigcup_{k \in [K]} B_k^2, \qquad \mathcal{D}^c := \{1, \ldots, d_z\}^2 \setminus \mathcal{D} \tag{A.20}
$$

Because $\hat{\boldsymbol{f}}$ has no interaction (Definition 3.2), we have that, for all $(j, j') \in \mathcal{D}^c$

$$
\begin{aligned}
0 &= \boldsymbol{W}^{\boldsymbol{f}}(\boldsymbol{h}(\boldsymbol{z})) \boldsymbol{m}^{\boldsymbol{h}}(\boldsymbol{z}, (j, j')) \\
&= \sum_{k \in [K]} \boldsymbol{W}_k^{\boldsymbol{f}}(\boldsymbol{h}(\boldsymbol{z})) \boldsymbol{m}_k^{\boldsymbol{h}}(\boldsymbol{z}, (j, j')) .
\end{aligned}
$$

Because $\boldsymbol{f}$ has no interaction, each row $\boldsymbol{W}_k^{\boldsymbol{f}}(\boldsymbol{h}(\boldsymbol{z}))_{n,\cdot}$ is non-zero for at most one $k \in [K]$ (although this $k$ can change for different values of $n$ and $\boldsymbol{z}$). This implies that for all $\boldsymbol{z} \in \hat{\mathcal{Z}}_{\text{supp}}, (j, j') \in \mathcal{D}^c$, $k \in [K]$:

$$0 = \boldsymbol{W}_k^{\boldsymbol{f}}(\boldsymbol{h}(\boldsymbol{z}))\boldsymbol{m}_k^{\boldsymbol{h}}(\boldsymbol{z}, (j, j')) \tag{A.21}$$

**Case 1:** $|B_k| = 1$ (One-Dimensional Slots)

When $|B_k| = 1$, for all $k \in [d_z]$, the matrix $\boldsymbol{W}_k^{\boldsymbol{f}}(\boldsymbol{h}(\boldsymbol{z}))$ can be written as:

$$\boldsymbol{W}_k^{\boldsymbol{f}}(\boldsymbol{h}(\boldsymbol{z})) = [D_k \boldsymbol{f}(\boldsymbol{z}) \odot D_k \boldsymbol{f}(\boldsymbol{z})] \tag{A.22}$$

This matrix has a single column, which must be non-zero since $\boldsymbol{f}$ is a $C^1$ diffeomorphism. Thus, $\boldsymbol{W}_k^{\boldsymbol{f}}(\boldsymbol{h}(\boldsymbol{z}))$ has full column rank and thus has a null space equal to 0. Using Eq. (A.21), we conclude that for all $(j, j') \in \mathcal{D}^c$, $k \in [d_z]$:

$$0 = \boldsymbol{m}_k^{\boldsymbol{h}}(\boldsymbol{z}, (j, j')) \tag{A.23}$$

Applying the definition of $\boldsymbol{m}_k^{\boldsymbol{h}}(\boldsymbol{z}, (j, j'))$, this implies that for all $(j, j') \in \mathcal{D}^c$, $k \in [d_z]$:

$$0 = D_{j'} \boldsymbol{h}_k(\boldsymbol{z}) D_j \boldsymbol{h}_k(\boldsymbol{z}) \tag{A.24}$$

This means each row of the Jacobian matrix $D\boldsymbol{h}(\boldsymbol{z})$ cannot have more than one nonzero value. Since the Jacobian is invertible, these nonzero values must all be different for different rows, otherwise a whole column would be zero. Hence $D\boldsymbol{h}(\boldsymbol{z})$ is a permutation-scaling matrix, i.e. we have local disentanglement.

**Case 2:** $|B_k| > 1$ (Multi-Dimensional Slots)

Assume for a contradiction that $\hat{\boldsymbol{f}}$ is not locally disentangled on $\mathcal{Z}_{\text{supp}}$ w.r.t $\boldsymbol{f}$. This implies that there exist a $\boldsymbol{z}^* \in \hat{\mathcal{Z}}_{\text{supp}}, k, k', k'' \in [K]$ for $k' \neq k''$, such that:

$$D_{B_{k'}} \boldsymbol{h}_{B_k}(\boldsymbol{z}^*) \neq 0, \quad D_{B_{k''}} \boldsymbol{h}_{B_k}(\boldsymbol{z}^*) \neq 0 \tag{A.25}$$

Because $\boldsymbol{f}, \hat{\boldsymbol{f}}$ are $C^1$ diffeomorphisms, we know that $\boldsymbol{h}$ is also a $C^1$ diffeomorphism. Coupling this with Eq. (A.25), Lemma A.7 tells us that there exist an $S \subset [d_z]$ with cardinality $|B_k|$ such that:

$$\forall B \in \mathcal{B},\ S \not\subseteq B, \quad \text{and} \quad \forall i \in S, D_i \boldsymbol{h}_{B_k}(\boldsymbol{z}^*)\ \text{are linearly independent.} \tag{A.26}$$

Now choose any $\bar{B} \in \mathcal{B}$ such that $S_1 := S \cap \bar{B} \neq \emptyset$. Furthermore, define the set $S_2 := S \setminus S_1$. Because $S \not\subseteq \bar{B}$, we know that $S_2$ is non-empty. Further, by construction $S = S_1 \cup S_2$. In other words, $S_1$ and $S_2$ are non-empty, form a partition of $S$, and do not contain any indices from the same slot.

Now construct the matrices, denoted $\boldsymbol{A}_{S_1}$ and $\boldsymbol{A}_{S_2}$ as follows:

$$\boldsymbol{A}_{S_1} := D_{S_1} \boldsymbol{h}_{B_k}(\boldsymbol{z}^*), \quad \boldsymbol{A}_{S_2} := D_{S_2} \boldsymbol{h}_{B_k}(\boldsymbol{z}^*) \tag{A.27}$$

And the matrix denoted $\boldsymbol{A}_k$ as:

$$\boldsymbol{A}_k := [\boldsymbol{A}_{S_1}, \boldsymbol{A}_{S_2}] \tag{A.28}$$

Note that because, $\forall i \in S, D_i \boldsymbol{h}_{B_k}(\hat{\boldsymbol{z}}^*)$ are linearly independent (Eq. (A.26)), we know that $\boldsymbol{A}_k$ is invertible.

Now, define the following block diagonal matrix $\boldsymbol{A} \in \mathbb{R}^{d_z \times d_z}$ as follows:

$$\boldsymbol{A} := \begin{bmatrix} \boldsymbol{A}_1 & 0 & \dots & 0 \\ 0 & \boldsymbol{A}_2 & \dots & 0 \\ \vdots & \vdots & \ddots & \vdots \\ 0 & 0 & \dots & \boldsymbol{A}_K \end{bmatrix} \tag{A.29}$$

where $\forall i \in [K] \setminus \{k\}$, $\boldsymbol{A}_i$ is the identity matrix, and thus invertible, while $\boldsymbol{A}_k$ is defined according to Eq. (A.28).

Define $\bar{\mathcal{Z}} := \boldsymbol{A}^{-1}\mathcal{Z}$, the function $\bar{\boldsymbol{h}} : \bar{\mathcal{Z}} \to \mathcal{Z}$ as $\bar{\boldsymbol{h}}(\boldsymbol{z}) := \boldsymbol{A}\boldsymbol{z}$ and the function $\bar{\boldsymbol{f}} : \bar{\mathcal{Z}} \to \mathcal{X}$ as $\bar{\boldsymbol{f}} := \boldsymbol{f} \circ \bar{\boldsymbol{h}}$. By construction we have

$$\forall \boldsymbol{z} \in \mathcal{Z}, \quad \bar{\boldsymbol{f}}\left(\boldsymbol{A}_1^{-1}\boldsymbol{z}_{B_1}, \dots, \boldsymbol{A}_K^{-1}\boldsymbol{z}_{B_K}\right) = \boldsymbol{f}(\boldsymbol{z}_{B_1}, \dots, \boldsymbol{z}_{B_K}). \tag{A.30}$$

Because all $\boldsymbol{A}_i^{-1}$ are invertible, then $\bar{\boldsymbol{f}}$ is *equivalent* to $\boldsymbol{f}$ in the sense of Def. (4.1).

We can now apply Lemma A.19 to $\bar{\boldsymbol{f}} = \boldsymbol{f} \circ \bar{\boldsymbol{h}}$ to obtain, for all $j, j' \in [d_z]$:

$$D_j \bar{\boldsymbol{f}}(\boldsymbol{z}) \odot D_{j'} \bar{\boldsymbol{f}}(\boldsymbol{z}) = \boldsymbol{W}^{\boldsymbol{f}}(\bar{\boldsymbol{h}}(\boldsymbol{z})) \boldsymbol{m}^{\bar{\boldsymbol{h}}}(\boldsymbol{z}, (j, j')) . \tag{A.31}$$

Choose $\bar{\boldsymbol{z}} \in \bar{\mathcal{Z}}$ such that $\bar{\boldsymbol{h}}(\bar{\boldsymbol{z}}) = \boldsymbol{h}(\boldsymbol{z}^*)$, which is possible because $\boldsymbol{h}(\boldsymbol{z}^*) \in \mathcal{Z}$ and $\bar{\boldsymbol{h}}$ is a bijection from $\bar{\mathcal{Z}}$ to $\mathcal{Z}$). We can then write

$$D_j \bar{\boldsymbol{f}}(\bar{\boldsymbol{z}}) \odot D_{j'} \bar{\boldsymbol{f}}(\bar{\boldsymbol{z}}) = \boldsymbol{W}^{\boldsymbol{f}}(\boldsymbol{h}(\boldsymbol{z}^*)) \boldsymbol{m}^{\bar{\boldsymbol{h}}}(\bar{\boldsymbol{z}}, (j, j')) . \tag{A.32}$$

Let $J, J' \subseteq B_k$ be a partition of $B_k$ such that $J$ is the set of columns of $\boldsymbol{A}$ corresponding to $\boldsymbol{A}_{S_1}$ and $J'$ be the set of columns of $\boldsymbol{A}$ corresponding to $\boldsymbol{A}_{S_2}$. More formally, we have

$$\boldsymbol{A}_{B_k, J} = \boldsymbol{A}_{S_1} \qquad \text{and} \qquad \boldsymbol{A}_{B_k, J'} = \boldsymbol{A}_{S_2}$$

Since $\boldsymbol{A}_{S_1} = D_{S_1} \boldsymbol{h}_{B_k}(\boldsymbol{z}^*)$ and $\boldsymbol{A}_{S_2} = D_{S_2} \boldsymbol{h}_{B_k}(\boldsymbol{z}^*)$, we have that

$$\boldsymbol{A}_{B_k, J} = D_{S_1} \boldsymbol{h}_{B_k}(\boldsymbol{z}^*) \quad \text{and} \quad \boldsymbol{A}_{B_k, J'} = D_{S_2} \boldsymbol{h}_{B_k}(\boldsymbol{z}^*)$$

Since $D\bar{\boldsymbol{h}}(\bar{\boldsymbol{z}}) = \boldsymbol{A}$, we have

$$D_J \bar{\boldsymbol{h}}_{B_k}(\bar{\boldsymbol{z}}) = D_{S_1} \boldsymbol{h}_{B_k}(\boldsymbol{z}^*) \quad \text{and} \quad D_{J'} \bar{\boldsymbol{h}}_{B_k}(\bar{\boldsymbol{z}}) = D_{S_2} \boldsymbol{h}_{B_k}(\boldsymbol{z}^*) .$$

Choose some $(j, j') \in J \times J'$. We know there must exist $(s, s') \in S_1 \times S_2$ such that

$$D_j \bar{\boldsymbol{h}}_{B_k}(\bar{\boldsymbol{z}}) = D_s \boldsymbol{h}_{B_k}(\boldsymbol{z}^*) \quad \text{and} \quad D_{j'} \bar{\boldsymbol{h}}_{B_k}(\bar{\boldsymbol{z}}) = D_{s'} \boldsymbol{h}_{B_k}(\boldsymbol{z}^*) .$$

which implies

$$\boldsymbol{m}_k^{\bar{\boldsymbol{h}}}(\bar{\boldsymbol{z}}, (j, j')) = \boldsymbol{m}_k^{\boldsymbol{h}}(\boldsymbol{z}^*, (s, s')) . \tag{A.33}$$

Moreover, since the Jacobian of $\bar{\boldsymbol{h}}$ is block diagonal, we have that $\boldsymbol{m}_{k'}^{\bar{\boldsymbol{h}}}(\boldsymbol{z}, (j, j')) = 0$ for all $k' \neq k$ (recall that $j, j' \in B_k$). This means we can rewrite (A.32) as

$$D_j \bar{\boldsymbol{f}}(\bar{\boldsymbol{z}}) \odot D_{j'} \bar{\boldsymbol{f}}(\bar{\boldsymbol{z}}) = \boldsymbol{W}_k^{\boldsymbol{f}}(\boldsymbol{h}(\boldsymbol{z}^*)) \boldsymbol{m}_k^{\bar{\boldsymbol{h}}}(\bar{\boldsymbol{z}}, (j, j')) . \tag{A.34}$$

Plugging (A.33) into the above equation yields

$$D_j \bar{\boldsymbol{f}}(\bar{\boldsymbol{z}}) \odot D_{j'} \bar{\boldsymbol{f}}(\bar{\boldsymbol{z}}) = \boldsymbol{W}_k^{\boldsymbol{f}}(\boldsymbol{h}(\boldsymbol{z}^*)) \boldsymbol{m}_k^{\boldsymbol{h}}(\boldsymbol{z}^*, (s, s')) . \tag{A.35}$$

Since $(s, s') \in S_1 \times S_2 \subseteq \mathcal{D}^c$, we can apply (A.21) to get

$$D_j \bar{\boldsymbol{f}}(\bar{\boldsymbol{z}}) \odot D_{j'} \bar{\boldsymbol{f}}(\bar{\boldsymbol{z}}) = \boldsymbol{W}_k^{\boldsymbol{f}}(\boldsymbol{h}(\boldsymbol{z}^*)) \boldsymbol{m}_k^{\boldsymbol{h}}(\boldsymbol{z}^*, (s, s')) = 0 . \tag{A.36}$$

In other words, we found a partition $J, J'$ of the block $B_k$ such that $D_j \bar{\boldsymbol{f}}(\bar{\boldsymbol{z}}) \odot D_{j'} \bar{\boldsymbol{f}}(\bar{\boldsymbol{z}}) = 0$ for all $(j, j') \in J \times J'$. This means that the blocks $J$ and $J'$ have *no interaction* in $\bar{\boldsymbol{f}}$ at $\bar{\boldsymbol{z}}$. This is a contradiction with Assm. 3.5. Hence, we have local disentanglement.

**Local to global disentanglement.** We showed that $D\boldsymbol{h}(\boldsymbol{z})$ is a block-permutation matrix for all $\boldsymbol{z} \in \hat{\mathcal{Z}}_{\text{supp}}$, i.e. local disentanglement. Consider the inverse $\boldsymbol{h}$, $\boldsymbol{v} := \boldsymbol{h}^{-1}$. The Jacobian of $\boldsymbol{v}$ is given by $D\boldsymbol{v}(\boldsymbol{z}) = D\boldsymbol{h}^{-1}(\boldsymbol{z}) = D\boldsymbol{h}(\boldsymbol{v}(\boldsymbol{z}))^{-1}$, by the inverse function theorem. By Proposition A.12, this means $D\boldsymbol{v}(\boldsymbol{z})$ is also a block permutation matrix for all $\boldsymbol{z} \in \mathcal{Z}_{\text{supp}}$. Since $\mathcal{Z}_{\text{supp}}$ is aligned-connected (Definition A.16), Lemma A.18 guarantees that we can write $\boldsymbol{v}(\boldsymbol{z}) = (\boldsymbol{v}_1(\boldsymbol{z}_{B_{\pi(1)}}), \ldots, \boldsymbol{v}_K(\boldsymbol{z}_{B_{\pi(K)}}))$ for all $\boldsymbol{z} \in \mathcal{Z}_{\text{supp}}$ where the $\boldsymbol{v}_k$ are diffeomorphisms. This implies that $\boldsymbol{h}(\boldsymbol{z}) = (\boldsymbol{v}_1^{-1}(\boldsymbol{z}_{B_{\pi^{-1}(1)}}), \ldots, \boldsymbol{v}_K^{-1}(\boldsymbol{z}_{B_{\pi^{-1}(K)}}))$ for all $\boldsymbol{z} \in \hat{\mathcal{Z}}_{\text{supp}}$, which concludes the proof. $\qquad \square$

## A.5 DISENTANGLEMENT (AT MOST 1$^{\text{ST}}$ ORDER INTERACTION)

**Lemma A.21.** *Let $\mathcal{Z}_{\text{supp}} \subseteq \mathcal{Z}$ be a regular closed set (Defn. A.3). Let $\boldsymbol{f} : \mathcal{Z} \to \mathcal{X}$ be $C^1$ and $\boldsymbol{h} : \hat{\mathcal{Z}}_{\text{supp}} \to \mathcal{Z}_{\text{supp}}$ be a diffeomorphism. Let $\hat{\boldsymbol{f}} := \boldsymbol{f} \circ \boldsymbol{h}$. If $\boldsymbol{f}$ has at most 1$^{\text{st}}$ order interaction (Definition 3.4 with $n = 1$), then, for all $j, j' \in [d_z]$ and $\boldsymbol{z} \in \hat{\mathcal{Z}}_{\text{supp}}$, we have*

$$D_{j, j'}^2 \hat{\boldsymbol{f}}(\boldsymbol{z}) = \boldsymbol{W}^{\boldsymbol{f}}(\boldsymbol{h}(\boldsymbol{z})) \boldsymbol{m}^{\boldsymbol{h}}(\boldsymbol{z}, (j, j')) , \tag{A.37}$$

*where*

$$\boldsymbol{W}^{\boldsymbol{f}}(\boldsymbol{z}) := [\boldsymbol{W}_k^{\boldsymbol{f}}(\boldsymbol{z}))]_{k \in [K]}$$

$$\boldsymbol{W}_k^{\boldsymbol{f}}(\boldsymbol{z}) := \left[ [D_{i_1} \boldsymbol{f}(\boldsymbol{z})]_{i_1 \in B_k}, [D_{i_1,i_2}^2 \boldsymbol{f}(\boldsymbol{z})]_{(i_1,i_2) \in B_k^2} \right]$$

$$\boldsymbol{m}^{\boldsymbol{h}}(\boldsymbol{z}, (j, j')) := [\boldsymbol{m}_k^{\boldsymbol{h}}(\boldsymbol{z}, (j, j'))]_{k \in [K]}$$

$$\boldsymbol{m}_k^{\boldsymbol{h}}(\boldsymbol{z}, (j, j')) := \left[ [D_{j,j'}^2 \boldsymbol{h}_{i_1}(\boldsymbol{z})]_{i_1 \in B_k}, [D_{j'} \boldsymbol{h}_{i_2}(\boldsymbol{z}) D_j \boldsymbol{h}_{i_1}(\boldsymbol{z})]_{(i_1,i_2) \in B_k^2} \right].$$

*Proof.* The exact same argument as the one presented in Lemma A.19 (based on Lachapelle et al. (2023)) guarantees that, $\hat{\boldsymbol{f}}$ and $\boldsymbol{f} \circ \boldsymbol{h}$ have equal derivatives on $\hat{\mathcal{Z}}_{\text{supp}}$. We leverage this fact next.

By taking the derivative w.r.t. $\boldsymbol{z}_j$ on both sides of $\hat{\boldsymbol{f}}(\boldsymbol{z}) = \boldsymbol{f} \circ \boldsymbol{h}(\boldsymbol{z})$, we get

$$D_j \hat{\boldsymbol{f}}(\boldsymbol{z}) = \sum_{k \in [K]} \sum_{i \in B_k} D_i \boldsymbol{f}(\boldsymbol{h}(\boldsymbol{z})) D_j \boldsymbol{h}_i(\boldsymbol{z}) \tag{A.38}$$

Now take another derivative w.r.t. $\boldsymbol{z}_{j'}$ for some $j' \in [d_z]$ to get

$$D_{j,j'}^2 \hat{\boldsymbol{f}}(\boldsymbol{z}) = \sum_{k_1 \in [K]} \sum_{i_1 \in B_{k_1}} \left[ D_{i_1} \boldsymbol{f}(\boldsymbol{h}(\boldsymbol{z})) D_{j,j'}^2 \boldsymbol{h}_{i_1}(\boldsymbol{z}) + \sum_{k_2 \in [K]} \sum_{i_2 \in B_{k_2}} D_{i_1,i_2}^2 \boldsymbol{f}(\boldsymbol{h}(\boldsymbol{z})) D_{j'} \boldsymbol{h}_{i_2}(\boldsymbol{z}) D_j \boldsymbol{h}_{i_1}(\boldsymbol{z}) \right]$$

Because we have *at most first order interactions* (Def. 3.4 with $n = 1$), the second sum over $[K]$ drops, and we are left with:

$$D_{j,j'}^2 \hat{\boldsymbol{f}}(\boldsymbol{z}) = \sum_{k_1 \in [K]} \sum_{i_1 \in B_{k_1}} \left[ D_{i_1} \boldsymbol{f}(\boldsymbol{h}(\boldsymbol{z})) D_{j,j'}^2 \boldsymbol{h}_{i_1}(\boldsymbol{z}) + \sum_{i_2 \in B_{k_1}} D_{i_1,i_2}^2 \boldsymbol{f}(\boldsymbol{h}(\boldsymbol{z})) D_{j'} \boldsymbol{h}_{i_2}(\boldsymbol{z}) D_j \boldsymbol{h}_{i_1}(\boldsymbol{z}) \right]$$

$$= \sum_{k_1 \in [K]} \left[ \sum_{i_1 \in B_{k_1}} D_{i_1} \boldsymbol{f}(\boldsymbol{h}(\boldsymbol{z})) D_{j,j'}^2 \boldsymbol{h}_{i_1}(\boldsymbol{z}) + \sum_{(i_1,i_2) \in B_{k_1}^2} D_{i_1,i_2}^2 \boldsymbol{f}(\boldsymbol{h}(\boldsymbol{z})) D_{j'} \boldsymbol{h}_{i_2}(\boldsymbol{z}) D_j \boldsymbol{h}_{i_1}(\boldsymbol{z}) \right]$$

$$= \boldsymbol{W}^{\boldsymbol{f}}(\boldsymbol{h}(\boldsymbol{z})) \boldsymbol{m}^{\boldsymbol{h}}(\boldsymbol{z}, (j, j')),$$

which concludes the proof. $\qquad\square$

**Theorem A.22.** *Let $\boldsymbol{f} : \mathcal{Z} \to \mathcal{X}$ be a $C^2$ diffeomorphism satisfying interaction asymmetry (Asm. 3.5) for all equivalent generators (Defn. 4.1) for $n = 1$ and sufficient independence (Defn. A.9). Let $\mathcal{Z}_{\text{supp}} \subseteq \mathcal{Z}$ be regular closed (Defn. A.3), path-connected (Defn. A.14) and aligned-connected (Defn. A.16). A model $\hat{\boldsymbol{f}} : \mathcal{Z} \to \mathbb{R}^{d_x}$ disentangles $\boldsymbol{z}$ on $\mathcal{Z}_{\text{supp}}$ w.r.t. $\boldsymbol{f}$ (Defn. 2.1) if it is (i) a $C^2$ diffeomorphism between $\hat{\mathcal{Z}}_{\text{supp}}$ and $\mathcal{X}_{\text{supp}}$ with (ii) at most $1^{st}$ order interactions across slots (Defn. 3.4) on $\hat{\mathcal{Z}}_{\text{supp}}$.*

*Proof.* As mentioned in Section A.3, the proofs will proceed in two steps: First, we show local disentanglement (Definition A.13) and then we show (global) disentanglement via Lemma A.18. We first show local disentanglement.

We first define the function $\boldsymbol{h} : \hat{\mathcal{Z}}_{\text{supp}} \to \mathcal{Z}_{\text{supp}}$ relating the latent spaces of these functions on $\hat{\mathcal{Z}}_{\text{supp}}$:

$$\boldsymbol{h} := \boldsymbol{f}^{-1} \circ \hat{\boldsymbol{f}} \tag{A.39}$$

The function $\hat{\boldsymbol{f}}$ can then be written in terms of $\boldsymbol{f}$ and $\boldsymbol{h}$ on $\hat{\mathcal{Z}}_{\text{supp}}$:

$$\hat{\boldsymbol{f}} = \boldsymbol{f} \circ \boldsymbol{h} \tag{A.40}$$

Because $\boldsymbol{f}, \hat{\boldsymbol{f}}$ are both $C^2$ diffeomorphism between $\mathcal{Z}_{\text{supp}}, \mathcal{X}_{\text{supp}}$ and $\hat{\mathcal{Z}}_{\text{supp}}, \mathcal{X}_{\text{supp}}$, respectively, we have that $\boldsymbol{h}$ is a $C^2$ diffeomorphism.

Since $\boldsymbol{f}$ has at most 1st order interactions, we can apply Lemma A.21 to obtain, for all $\boldsymbol{z} \in \hat{\mathcal{Z}}_{\text{supp}}$, $j, j' \in [d_z]$,

$$D_{j,j'}^2 \hat{\boldsymbol{f}}(\boldsymbol{z}) = \boldsymbol{W^f}(\boldsymbol{h}(\boldsymbol{z})) \boldsymbol{m^h}(\boldsymbol{z}, (j, j'))\,.$$

Since $\hat{\boldsymbol{f}}$ has at most 1st order interaction, we have that, for all $(j, j') \in \mathcal{D}^c$

$$0 = \boldsymbol{W^f}(\boldsymbol{h}(\boldsymbol{z})) \boldsymbol{m^h}(\boldsymbol{z}, (j, j'))\,. \tag{A.41}$$

By defining

$$\boldsymbol{W}_k^{\boldsymbol{f},\text{rest}}(\boldsymbol{z}) := [D_{i_1} \boldsymbol{f}(\boldsymbol{z})]_{i_1 \in B_k}$$
$$\boldsymbol{W}_k^{\boldsymbol{f},\text{high}}(\boldsymbol{z}) := [D_{i_1,i_2}^2 \boldsymbol{f}(\boldsymbol{z})]_{(i_1,i_2) \in B_k^2}$$
$$\boldsymbol{m}_k^{\boldsymbol{h},\text{rest}}(\boldsymbol{z}, (j, j')) := [D_{j,j'}^2 \boldsymbol{h}_{i_1}(\boldsymbol{z})]_{i_1 \in B_k}$$
$$\boldsymbol{m}_k^{\boldsymbol{h},\text{high}}(\boldsymbol{z}, (j, j')) := [D_{j'} \boldsymbol{h}_{i_2}(\boldsymbol{z}) D_j \boldsymbol{h}_{i_1}(\boldsymbol{z})]_{(i_1,i_2) \in B_k^2}$$

we can restate the sufficiently independent derivative assumption (Def. A.9) as, for all $\boldsymbol{z} \in \mathcal{Z}$

$$\text{rank}\left(\boldsymbol{W^f}(\boldsymbol{z})\right) = \sum_{k \in [K]} \left[\text{rank}\left(\boldsymbol{W}_k^{\boldsymbol{f},\text{rest}}(\boldsymbol{z})\right) + \text{rank}\left(\boldsymbol{W}_k^{\boldsymbol{f},\text{high}}(\boldsymbol{z})\right)\right]$$

This condition allows us to apply Lemma A.6 to go from (A.41) to, for all $(j, j') \in \mathcal{D}^c$, $k \in [K]$:

$$0 = \boldsymbol{W}_k^{\boldsymbol{f},\text{high}}(\boldsymbol{h}(\boldsymbol{z})) \boldsymbol{m}_k^{\boldsymbol{h},\text{high}}(\boldsymbol{z}, (j, j')) \tag{A.42}$$

**Case 1:** $|B_k| = 1$ (One-Dimensional Slots) By Assumption 3.5.ii, (with $A = B = \{i\}$) $D_{i,i}^2 \boldsymbol{f}(\boldsymbol{z}) \neq 0$. Note that $\boldsymbol{W}_k^{\boldsymbol{f},\text{high}}(\boldsymbol{h}(\boldsymbol{z})) = D_{k,k}^2 \boldsymbol{f}(\boldsymbol{z})$. Hence, (A.42) implies that $\boldsymbol{m}_k^{\boldsymbol{h},\text{high}}(\boldsymbol{z}, (j, j')) = 0$ (which is a scalar). This means $\boldsymbol{m}_k^{\boldsymbol{h},\text{high}}(\boldsymbol{z}, (j, j')) = D_{j'} \boldsymbol{h}_k(\boldsymbol{z}) D_j \boldsymbol{h}_k(\boldsymbol{z}) = 0$. Since this is true for all $k$ and all distinct $j, j'$, this means each row has at most one nonzero entry. Since $D\boldsymbol{h}(\boldsymbol{z})$ is invertible, these nonzero entries must appear on different columns, otherwise a column will be filled with zeros. This means $D\boldsymbol{h}(\boldsymbol{z})$ is a permutation-scaling matrix, i.e. we have local disentanglement (Definition A.13).

**Case 2:** $|B_k| > 1$ (Multi-Dimensional Slots)

Assume for a contradiction that $\hat{\boldsymbol{f}}$ is not locally disentangled on $\mathcal{Z}_{\text{supp}}$ w.r.t. $\boldsymbol{f}$. This implies that there exist a $\boldsymbol{z}^* \in \hat{\mathcal{Z}}_{\text{supp}}$, $k, k', k'' \in [K]$ with $k' \neq k''$ such that:

$$D_{B_{k'}} \boldsymbol{h}_{B_k}(\boldsymbol{z}^*) \neq 0, \quad D_{B_{k''}} \boldsymbol{h}_{B_k}(\boldsymbol{z}^*) \neq 0 \tag{A.43}$$

Because $\boldsymbol{f}, \hat{\boldsymbol{f}}$ are $C^1$ diffeomorphisms, we know that $\boldsymbol{h}$ is also a $C^1$ diffeomorphism. Coupling this with Eq. (A.43), Lemma A.7 tells us that there exist an $S \subset [d_z]$ with cardinality $|B_k|$ such that:

$$\forall B \in \mathcal{B}, S \nsubseteq B, \quad \text{and} \quad \forall i \in S, D_i \boldsymbol{h}_{B_k}(\boldsymbol{z}^*) \text{ are linearly independent.} \tag{A.44}$$

Now choose any $\bar{B} \in \mathcal{B}$ such that $S_1 := \{S \cap \bar{B}\} \neq \emptyset$. Furthermore, define the set $S_2 := S \setminus S_1$. Because $S \nsubseteq \bar{B}$, we know that $S_2$ is non-empty. Further, by construction $S = S_1 \cup S_2$. In other words, $S_1$ and $S_2$ are non-empty, form a partition of $S$, and do not contain any indices from the same slot.

Now construct the matrices, denoted $\boldsymbol{A}_{S_1}$ and $\boldsymbol{A}_{S_2}$ as follows:

$$\boldsymbol{A}_{S_1} := D_{S_1} \boldsymbol{h}_{B_k}(\boldsymbol{z}^*), \quad \boldsymbol{A}_{S_2} := D_{S_2} \boldsymbol{h}_{B_k}(\boldsymbol{z}^*) \tag{A.45}$$

And the matrix denoted $\boldsymbol{A}_k$ as:

$$\boldsymbol{A}_k := [\boldsymbol{A}_{S_1}, \boldsymbol{A}_{S_2}] \tag{A.46}$$

Note that because, $\forall i \in S$, $D_i \boldsymbol{h}_{B_k}(\hat{\boldsymbol{z}}^*)$ are linearly independent (Eq. A.44), we know that $\boldsymbol{A}_k$ is invertible.

Now, define the following block diagonal matrix $\boldsymbol{A} \in \mathbb{R}^{d_z \times d_z}$ as follows:

$$\boldsymbol{A} := \begin{bmatrix} \boldsymbol{A}_1 & 0 & \dots & 0 \\ 0 & \boldsymbol{A}_2 & \dots & 0 \\ \vdots & \vdots & \ddots & \vdots \\ 0 & 0 & \dots & \boldsymbol{A}_K \end{bmatrix} \tag{A.47}$$

where $\forall i \in [K] \setminus \{k\}$, $\boldsymbol{A}_i$ is the identity matrix, and thus invertible, while $\boldsymbol{A}_k$ is defined according to Eq. (A.46).

Define $\bar{\mathcal{Z}} := \boldsymbol{A}^{-1}\mathcal{Z}$, the function $\bar{\boldsymbol{h}} : \bar{\mathcal{Z}} \to \mathcal{Z}$ as $\bar{\boldsymbol{h}}(\boldsymbol{z}) := \boldsymbol{A}\boldsymbol{z}$ and the function $\bar{\boldsymbol{f}} : \bar{\mathcal{Z}} \to \mathcal{X}$ as $\bar{\boldsymbol{f}} := \boldsymbol{f} \circ \bar{\boldsymbol{h}}$. By construction we have

$$\forall \boldsymbol{z} \in \mathcal{Z}, \qquad \bar{\boldsymbol{f}}\left(\boldsymbol{A}_1^{-1}\boldsymbol{z}_{B_1}, \ldots, \boldsymbol{A}_K^{-1}\boldsymbol{z}_{B_K}\right) = \boldsymbol{f}(\boldsymbol{z}_{B_1}, \ldots, \boldsymbol{z}_{B_K}). \tag{A.48}$$

Because all $\boldsymbol{A}_i^{-1}$ are invertible, then $\bar{\boldsymbol{f}}$ is *equivalent* to $\boldsymbol{f}$ in the sense of Def. (4.1).

We can now apply Lemma A.21 to $\bar{\boldsymbol{f}} = \boldsymbol{f} \circ \bar{\boldsymbol{h}}$ to obtain, for all $j, j' \in [d_z]$:

$$D_{j,j'}^2 \bar{\boldsymbol{f}}(\boldsymbol{z}) = \boldsymbol{W}^{\boldsymbol{f}}(\bar{\boldsymbol{h}}(\boldsymbol{z}))\boldsymbol{m}^{\bar{\boldsymbol{h}}}(\boldsymbol{z},(j,j')). \tag{A.49}$$

Choose $\bar{\boldsymbol{z}} \in \bar{\mathcal{Z}}$ such that $\bar{\boldsymbol{h}}(\bar{\boldsymbol{z}}) = \boldsymbol{h}(\boldsymbol{z}^*)$, which is possible because $\boldsymbol{h}(\boldsymbol{z}^*) \in \mathcal{Z}$ and $\bar{\boldsymbol{h}}$ is a bijection from $\bar{\mathcal{Z}}$ to $\mathcal{Z}$. We can then write

$$D_{j,j'}^2 \bar{\boldsymbol{f}}(\bar{\boldsymbol{z}}) = \boldsymbol{W}^{\boldsymbol{f}}(\boldsymbol{h}(\boldsymbol{z}^*))\boldsymbol{m}^{\bar{\boldsymbol{h}}}(\bar{\boldsymbol{z}},(j,j')). \tag{A.50}$$

Let $J, J' \subseteq B_k$ be a partition of $B_k$ such that $J$ is the set of columns of $\boldsymbol{A}$ corresponding to $\boldsymbol{A}_{S_1}$ and $J'$ be the set of columns of $\boldsymbol{A}$ corresponding to $\boldsymbol{A}_{S_2}$. More formally, we have

$$\boldsymbol{A}_{B_k,J} = \boldsymbol{A}_{S_1} \qquad \text{and} \qquad \boldsymbol{A}_{B_k,J'} = \boldsymbol{A}_{S_2}.$$

Since $\boldsymbol{A}_{S_1} = D_{S_1}\boldsymbol{h}_{B_k}(\boldsymbol{z}^*)$ and $\boldsymbol{A}_{S_2} = D_{S_2}\boldsymbol{h}_{B_k}(\boldsymbol{z}^*)$, we have that

$$\boldsymbol{A}_{B_k,J} = D_{S_1}\boldsymbol{h}_{B_k}(\boldsymbol{z}^*) \quad \text{and} \quad \boldsymbol{A}_{B_k,J'} = D_{S_2}\boldsymbol{h}_{B_k}(\boldsymbol{z}^*)$$

Since $D\bar{\boldsymbol{h}}(\bar{\boldsymbol{z}}) = \boldsymbol{A}$, we have

$$D_J\bar{\boldsymbol{h}}_{B_k}(\bar{\boldsymbol{z}}) = D_{S_1}\boldsymbol{h}_{B_k}(\boldsymbol{z}^*) \quad \text{and} \quad D_{J'}\bar{\boldsymbol{h}}_{B_k}(\bar{\boldsymbol{z}}) = D_{S_2}\boldsymbol{h}_{B_k}(\boldsymbol{z}^*).$$

For all $(j,j') \in J \times J'$, there must exist $(s,s') \in S_1 \times S_2$ such that

$$D_j\bar{\boldsymbol{h}}_{B_k}(\bar{\boldsymbol{z}}) = D_s\boldsymbol{h}_{B_k}(\boldsymbol{z}^*) \quad \text{and} \quad D_{j'}\bar{\boldsymbol{h}}_{B_k}(\bar{\boldsymbol{z}}) = D_{s'}\boldsymbol{h}_{B_k}(\boldsymbol{z}^*).$$

This implies that, for all $(j,j') \in J \times J'$, there exists $(s,s') \in S_1 \times S_2$ such that

$$\boldsymbol{m}_k^{\bar{\boldsymbol{h}},\text{high}}(\bar{\boldsymbol{z}},(j,j')) = \boldsymbol{m}_k^{\boldsymbol{h},\text{high}}(\boldsymbol{z}^*,(s,s')). \tag{A.51}$$

Moreover, since $\bar{\boldsymbol{h}}$ is a block-wise function we have that, for all $(j,j') \in J \times J' \subseteq B_k$ and $k' \in [K] \setminus \{k\}$, $\boldsymbol{m}_{k'}^{\bar{\boldsymbol{h}}}(\bar{\boldsymbol{z}},(j,j')) = 0$. We can thus write:

$$D_{j,j'}^2\bar{\boldsymbol{f}}(\bar{\boldsymbol{z}}) = \boldsymbol{W}_k^{\boldsymbol{f}}(\boldsymbol{h}(\boldsymbol{z}^*))\boldsymbol{m}_k^{\bar{\boldsymbol{h}}}(\bar{\boldsymbol{z}},(j,j')). \tag{A.52}$$

Since $\bar{\boldsymbol{h}}$ is linear, we have that $\boldsymbol{m}_k^{\bar{\boldsymbol{h}},\text{rest}}(\bar{\boldsymbol{z}},(j,j')) = 0$, and thus

$$D_{j,j'}^2\bar{\boldsymbol{f}}(\bar{\boldsymbol{z}}) = \boldsymbol{W}_k^{\boldsymbol{f},\text{high}}(\boldsymbol{h}(\boldsymbol{z}^*))\boldsymbol{m}_k^{\bar{\boldsymbol{h}},\text{high}}(\bar{\boldsymbol{z}},(j,j')). \tag{A.53}$$

Plug the (A.51) into the above to obtain that, for all $(j,j') \in J \times J'$,

$$D_{j,j'}^2\bar{\boldsymbol{f}}(\bar{\boldsymbol{z}}) = \boldsymbol{W}_k^{\boldsymbol{f},\text{high}}(\boldsymbol{h}(\boldsymbol{z}^*))\boldsymbol{m}_k^{\boldsymbol{h},\text{high}}(\boldsymbol{z}^*,(s,s')) = 0, \tag{A.54}$$

where the very last "$= 0$" is due to (A.42) (recall $(s,s') \in S_1 \times S_2 \subseteq \mathcal{D}^c$).

In other words, we found a partition $J, J'$ of the block $B_k$ and a value $\bar{\boldsymbol{z}}$ such that $D_{j,j'}^2\bar{\boldsymbol{f}}(\bar{\boldsymbol{z}}) = 0$ for all $(j,j') \in J \times J'$. This means that the blocks $J$ and $J'$ have *no second order interaction* in $\bar{\boldsymbol{f}}$ at $\bar{\boldsymbol{z}}$. This is a contradiction with Assm. 3.5. Hence, we have local disentanglement.

**From local to global disentanglement.** The same argument as in the proof of Theorem A.20 applies.

$\square$

## A.6 DISENTANGLEMENT (AT MOST 2$^{\text{ND}}$ ORDER INTERACTION)

**Lemma A.23.** *Let $\mathcal{Z}_{\text{supp}} \subseteq \mathcal{Z}$ be a regular closed set (Defn. A.3). Let $\boldsymbol{f} : \mathcal{Z} \to \mathcal{X}$ be $C^1$ and $\boldsymbol{h} : \hat{\mathcal{Z}}_{\text{supp}} \to \mathcal{Z}_{\text{supp}}$ be a diffeomorphism. Let $\hat{\boldsymbol{f}} := \boldsymbol{f} \circ \boldsymbol{h}$. If $\boldsymbol{f}$ has at most $2^{nd}$ order interaction (Definition 3.4 with $n = 2$), then, for all $j, j' \in [d_z]$ and $\boldsymbol{z} \in \hat{\mathcal{Z}}_{\text{supp}}$, we have*

$$D^3_{j,j',j''}\hat{\boldsymbol{f}}(\boldsymbol{z}) = \boldsymbol{W}^{\boldsymbol{f}}(\boldsymbol{h}(\boldsymbol{z}))\boldsymbol{m}^{\boldsymbol{h}}(\boldsymbol{z}, (j, j', j'')) , \tag{A.55}$$

*where*

$$\begin{aligned}
\boldsymbol{W}^{\boldsymbol{f}}(\boldsymbol{z}) &:= [\boldsymbol{W}^{\boldsymbol{f}}_k(\boldsymbol{z}))]_{k \in [K]} \\
\boldsymbol{W}^{\boldsymbol{f}}_k(\boldsymbol{z}) &:= \Big[ [D_{i_1}\boldsymbol{f}(\boldsymbol{z})]_{i_1 \in B_k}, \\
&\qquad [D^2_{i_1,i_2}\boldsymbol{f}(\boldsymbol{z})]_{i_1 \in B_k, i_2 \in [d_z]}, \\
&\qquad [D^3_{i_1,i_2,i_3}\boldsymbol{f}(\boldsymbol{z})]_{(i_1,i_2,i_3) \in B^3_k} \Big] \\
\boldsymbol{m}^{\boldsymbol{h}}(\boldsymbol{z}, (j, j', j'')) &:= [\boldsymbol{m}^{\boldsymbol{h}}_k(\boldsymbol{z}, (j, j', j''))]_{k \in [K]} \\
\boldsymbol{m}^{\boldsymbol{h}}_k(\boldsymbol{z}, (j, j', j'')) &:= \Big[ [D^3_{j,j',j''}\boldsymbol{h}_{i_1}(\boldsymbol{z})]_{i_1 \in B_k}, \\
&\qquad [D_j\boldsymbol{h}_{i_1}(\boldsymbol{z})D^2_{j',j''}\boldsymbol{h}_{i_2}(\boldsymbol{z}) + D_{j'}\boldsymbol{h}_{i_2}(\boldsymbol{z})D^2_{j,j''}\boldsymbol{h}_{i_1}(\boldsymbol{z}) + D_{j''}\boldsymbol{h}_{i_2}(\boldsymbol{z})D^2_{j,j'}\boldsymbol{h}_{i_1}(\boldsymbol{z})]_{i_1 \in B_k, i_2 \in [d_z]} \\
&\qquad [D_{j''}\boldsymbol{h}_{i_3}(\boldsymbol{z})D_{j'}\boldsymbol{h}_{i_2}(\boldsymbol{z})D_j\boldsymbol{h}_{i_1}(\boldsymbol{z})]_{(i_1,i_2,i_3) \in B^3_k} \Big] .
\end{aligned}$$

*Proof.* As argued in Lemma A.21, differentiating $\hat{\boldsymbol{f}}(\boldsymbol{z}) = \boldsymbol{f} \circ \boldsymbol{h}(\boldsymbol{z})$ w.r.t. $\boldsymbol{z}_j$ and $\boldsymbol{z}_{j'}$ on both sides yields

$$D^2_{j,j'}\hat{\boldsymbol{f}}(\boldsymbol{z}) = \sum_{k_1 \in [K]} \sum_{i_1 \in B_{k_1}} \left[ D_{i_1}\boldsymbol{f}(\boldsymbol{h}(\boldsymbol{z}))D^2_{j,j'}\boldsymbol{h}_{i_1}(\boldsymbol{z}) + \sum_{k_2 \in [K]} \sum_{i_2 \in B_{k_2}} D^2_{i_1,i_2}\boldsymbol{f}(\boldsymbol{h}(\boldsymbol{z}))D_{j'}\boldsymbol{h}_{i_2}(\boldsymbol{z})D_j\boldsymbol{h}_{i_1}(\boldsymbol{z}) \right]$$

Now take another derivative with respect to $z_{j''}$ to compute $D^3_{j,j',j''}\hat{\boldsymbol{f}}(\boldsymbol{z})$. For the first term in the sum, we have:

$$\sum_{k_1 \in [K]} \sum_{i_1 \in B_{k_1}} \left[ \sum_{k_2 \in [K]} \sum_{i_2 \in B_{k_2}} D^2_{i_1,i_2}\boldsymbol{f}_n(\boldsymbol{h}(\boldsymbol{z}))D_{j''}\boldsymbol{h}_{i_2}(\boldsymbol{z})D^2_{j,j'}\boldsymbol{h}_{i_1}(\boldsymbol{z}) + D_{i_1}\boldsymbol{f}_n(\boldsymbol{h}(\boldsymbol{z}))D^3_{j,j',j''}\boldsymbol{h}_{i_1}(\boldsymbol{z}) \right]$$

And for the second term in the sum (the nested sum), we have:

$$\sum_{k_1 \in [K]} \sum_{i_1 \in B_{k_1}} \sum_{k_2 \in [K]} \sum_{i_2 \in B_{k_2}} \left[ \sum_{k_3 \in [K]} \sum_{i_3 \in B_{k_3}} D^3_{i_1,i_2,i_3}\boldsymbol{f}_n(\boldsymbol{h}(\boldsymbol{z}))D_{j''}\boldsymbol{h}_{i_3}(\boldsymbol{z})D_{j'}\boldsymbol{h}_{i_2}(\boldsymbol{z})D_j\boldsymbol{h}_{i_1}(\boldsymbol{z}) + \right.$$

$$\left. D^2_{i_1,i_2}\boldsymbol{f}_n(\boldsymbol{h}(\boldsymbol{z}))\Big[D^2_{j',j''}\boldsymbol{h}_{i_2}(\boldsymbol{z})D_j\boldsymbol{h}_{i_1}(\boldsymbol{z}) + D_{j'}\boldsymbol{h}_{i_2}(\boldsymbol{z})D^2_{j,j''}\boldsymbol{h}_{i_1}(\boldsymbol{z})\Big] \right]$$

Because we have at most second order interactions (Def. 3.4 with $n = 2$), this term can be rewritten as:

$$\sum_{k_1 \in [K]} \sum_{i_1 \in B_{k_1}} \left[ \sum_{i_2 \in B_{k_1}} \sum_{i_3 \in B_{k_1}} D^3_{i_1,i_2,i_3}\boldsymbol{f}_n(\boldsymbol{h}(\boldsymbol{z}))D_{j''}\boldsymbol{h}_{i_3}(\boldsymbol{z})D_{j'}\boldsymbol{h}_{i_2}(\boldsymbol{z})D_j\boldsymbol{h}_{i_1}(\boldsymbol{z}) + \right.$$

$$\left. \sum_{k_2 \in [K]} \sum_{i_2 \in B_{k_2}} D^2_{i_1,i_2}\boldsymbol{f}_n(\boldsymbol{h}(\boldsymbol{z}))\Big[D^2_{j',j''}\boldsymbol{h}_{i_2}(\boldsymbol{z})D_j\boldsymbol{h}_{i_1}(\boldsymbol{z}) + D_{j'}\boldsymbol{h}_{i_2}(\boldsymbol{z})D^2_{j,j''}\boldsymbol{h}_{i_1}(\boldsymbol{z})\Big] \right]$$

Combining the first and second terms, we get:

$$
\begin{aligned}
D^3_{j,j',j''}\hat{\boldsymbol{f}}(\boldsymbol{z}) = \sum_{k_1\in[K]}\sum_{i_1\in B_{k_1}}\Bigg[ &D_{i_1}\boldsymbol{f}(\boldsymbol{h}(\boldsymbol{z}))D^3_{j,j',j''}\boldsymbol{h}_{i_1}(\boldsymbol{z}) + \\
\sum_{k_2\in[K]}\sum_{i_2\in B_{k_2}}&D^2_{i_1,i_2}\boldsymbol{f}(\boldsymbol{h}(\boldsymbol{z}))\Big(D_j\boldsymbol{h}_{i_1}(\boldsymbol{z})D^2_{j',j''}\boldsymbol{h}_{i_2}(\boldsymbol{z}) + D_{j'}\boldsymbol{h}_{i_2}(\boldsymbol{z})D^2_{j,j''}\boldsymbol{h}_{i_1}(\boldsymbol{z}) + D_{j''}\boldsymbol{h}_{i_2}(\boldsymbol{z})D^2_{j,j'}\boldsymbol{h}_{i_1}(\boldsymbol{z})\Big) + \\
\sum_{i_2\in B_{k_1}}\sum_{i_3\in B_{k_1}}&D^3_{i_1,i_2,i_3}\boldsymbol{f}(\boldsymbol{h}(\boldsymbol{z}))D_{j''}\boldsymbol{h}_{i_3}(\boldsymbol{z})D_{j'}\boldsymbol{h}_{i_2}(\boldsymbol{z})D_j\boldsymbol{h}_{i_1}(\boldsymbol{z})\Bigg] \\
= \sum_{k_1\in[K]}\Bigg[\sum_{i_1\in B_{k_1}}&D_{i_1}\boldsymbol{f}(\boldsymbol{h}(\boldsymbol{z}))D^3_{j,j',j''}\boldsymbol{h}_{i_1}(\boldsymbol{z}) + \\
\sum_{i_1\in B_{k_1}}\sum_{i_2\in[d_z]}&D^2_{i_1,i_2}\boldsymbol{f}(\boldsymbol{h}(\boldsymbol{z}))\Big(D_j\boldsymbol{h}_{i_1}(\boldsymbol{z})D^2_{j',j''}\boldsymbol{h}_{i_2}(\boldsymbol{z}) + D_{j'}\boldsymbol{h}_{i_2}(\boldsymbol{z})D^2_{j,j''}\boldsymbol{h}_{i_1}(\boldsymbol{z}) + D_{j''}\boldsymbol{h}_{i_2}(\boldsymbol{z})D^2_{j,j'}\boldsymbol{h}_{i_1}(\boldsymbol{z})\Big) + \\
\sum_{(i_1,i_2,i_3)\in B^3_{k_1}}&D^3_{i_1,i_2,i_3}\boldsymbol{f}(\boldsymbol{h}(\boldsymbol{z}))D_{j''}\boldsymbol{h}_{i_3}(\boldsymbol{z})D_{j'}\boldsymbol{h}_{i_2}(\boldsymbol{z})D_j\boldsymbol{h}_{i_1}(\boldsymbol{z})\Bigg] \\
= \boldsymbol{W}^{\boldsymbol{f}}(\boldsymbol{h}(\boldsymbol{z}))&\boldsymbol{m}^{\boldsymbol{h}}(\boldsymbol{z},(j,j',j''))\,.
\end{aligned}
$$

$\square$

**Theorem A.24.** *Let $\boldsymbol{f} : \mathcal{Z} \to \mathcal{X}$ be a $C^3$ diffeomorphism satisfying interaction asymmetry (Asm. 3.5) for all equivalent generators (Defn. 4.1) for $n = 2$ and sufficient independence (Defn. 4.2). Let $\mathcal{Z}_{\mathrm{supp}} \subseteq \mathcal{Z}$ be regular closed (Defn. A.3), path-connected (Defn. A.14) and aligned-connected (Defn. A.16). A model $\hat{\boldsymbol{f}} : \mathcal{Z} \to \mathbb{R}^{d_x}$ disentangles $\boldsymbol{z}$ on $\mathcal{Z}_{\mathrm{supp}}$ w.r.t. $\boldsymbol{f}$ (Defn. 2.1) if it is (i) a $C^3$ diffeomorphism between $\hat{\mathcal{Z}}_{\mathrm{supp}}$ and $\mathcal{X}_{\mathrm{supp}}$ with (ii) at most $2^{nd}$ order interactions across slots (Defn. 3.4) on $\hat{\mathcal{Z}}_{\mathrm{supp}}$.*

*Proof.* As mentioned in Section A.3, the proofs will proceed in two steps: First, we show local disentanglement (Definition A.13) and then we show (global) disentanglement via Lemma A.18. We first show local disentanglement.

We first define the function $\boldsymbol{h} : \hat{\mathcal{Z}}_{\mathrm{supp}} \to \mathcal{Z}_{\mathrm{supp}}$ relating the latent spaces of these functions on $\hat{\mathcal{Z}}_{\mathrm{supp}}$:

$$\boldsymbol{h} := \boldsymbol{f}^{-1} \circ \hat{\boldsymbol{f}} \tag{A.56}$$

The function $\hat{\boldsymbol{f}}$ can then be written in terms of $\boldsymbol{f}$ and $\boldsymbol{h}$ on $\hat{\mathcal{Z}}_{\mathrm{supp}}$:

$$\hat{\boldsymbol{f}} = \boldsymbol{f} \circ \boldsymbol{h} \tag{A.57}$$

Because $\boldsymbol{f}, \hat{\boldsymbol{f}}$ are both $C^2$ diffeomorphism between $\mathcal{Z}_{\mathrm{supp}}, \mathcal{X}_{\mathrm{supp}}$ and $\hat{\mathcal{Z}}_{\mathrm{supp}}, \mathcal{X}_{\mathrm{supp}}$, respectively, we have that $\boldsymbol{h}$ is a $C^2$ diffeomorphism.

Since $\hat{\boldsymbol{f}}$ has at most $2^{nd}$ order interaction, we have that, for all $\boldsymbol{z} \in \hat{\mathcal{Z}}_{\mathrm{supp}}$, $(j, j', j'') \in \mathcal{D}^c \times [d_z]$,

$$0 = \boldsymbol{W}^{\boldsymbol{f}}(\boldsymbol{h}(\boldsymbol{z}))\boldsymbol{m}^{\boldsymbol{h}}(\boldsymbol{z},(j,j',j''))\,. \tag{A.58}$$

By defining

$$\boldsymbol{W}_k^{\boldsymbol{f},\text{rest}}(\boldsymbol{z}) := \Bigg[ [D_{i_1}\boldsymbol{f}(\boldsymbol{z})]_{i_1 \in B_k},$$

$$[D_{i_1,i_2}^2\boldsymbol{f}(\boldsymbol{z})]_{i_1 \in B_k, i_2 \in [d_z]}\Bigg]$$

$$\boldsymbol{W}_k^{\boldsymbol{f},\text{high}}(\boldsymbol{z}) := [D_{i_1,i_2,i_3}^3\boldsymbol{f}(\boldsymbol{z})]_{(i_1,i_2,i_3) \in B_k^3}$$

$$\boldsymbol{m}_k^{\boldsymbol{h},\text{rest}}(\boldsymbol{z},(j,j',j'')) := \Bigg[ [D_{j,j',j''}^3\boldsymbol{h}_{i_1}(\boldsymbol{z})]_{i_1 \in B_k},$$

$$[D_j\boldsymbol{h}_{i_1}(\boldsymbol{z})D_{j',j''}^2\boldsymbol{h}_{i_2}(\boldsymbol{z}) + D_{j'}\boldsymbol{h}_{i_2}(\boldsymbol{z})D_{j,j''}^2\boldsymbol{h}_{i_1}(\boldsymbol{z}) + D_{j''}\boldsymbol{h}_{i_2}(\boldsymbol{z})D_{j,j'}^2\boldsymbol{h}_{i_1}(\boldsymbol{z})]_{i_1 \in B_k, i_2 \in [d_z]}\Bigg]$$

$$\boldsymbol{m}_k^{\boldsymbol{h},\text{high}}(\boldsymbol{z},(j,j',j'')) := [D_{j''}\boldsymbol{h}_{i_3}(\boldsymbol{z})D_{j'}\boldsymbol{h}_{i_2}(\boldsymbol{z})D_j\boldsymbol{h}_{i_1}(\boldsymbol{z})]_{(i_1,i_2,i_3) \in B_k^3},$$

we can restate the sufficiently independent derivative assumption (Def. 4.2) as, for all $\boldsymbol{z} \in \mathcal{Z}$

$$\text{rank}\left(\boldsymbol{W}^{\boldsymbol{f}}(\boldsymbol{z})\right) = \sum_{k \in [K]}\left[\text{rank}\left(\boldsymbol{W}_k^{\boldsymbol{f},\text{rest}}(\boldsymbol{z})\right) + \text{rank}\left(\boldsymbol{W}_k^{\boldsymbol{f},\text{high}}(\boldsymbol{z})\right)\right]$$

This condition allows us to apply Lemma A.6 to go from (A.58) to, for all $(j,j',j'') \in \mathcal{D}^c \times [d_z]$, $k \in [K]$:

$$0 = \boldsymbol{W}_k^{\boldsymbol{f},\text{high}}(\boldsymbol{h}(\boldsymbol{z}))\boldsymbol{m}_k^{\boldsymbol{h},\text{high}}(\boldsymbol{z},(j,j',j'')) \tag{A.59}$$

**Case 1:** $|B_k| = 1$ (One-Dimensional Slots) By Assumption 3.5.ii (with $A = B = \{i\}$), we have that $D_{i,i,i}^3\boldsymbol{f}(\boldsymbol{z}) \neq 0$. Note that $\boldsymbol{W}_k^{\boldsymbol{f},\text{high}}(\boldsymbol{h}(\boldsymbol{z})) = D_{k,k,k}^3\boldsymbol{f}(\boldsymbol{z})$. Hence, (A.42) implies that $\boldsymbol{m}_k^{\boldsymbol{h},\text{high}}(\boldsymbol{z},(j,j',j'')) = 0$ (which is a scalar). This means $\boldsymbol{m}_k^{\boldsymbol{h},\text{high}}(\boldsymbol{z},(j,j',j'')) = D_{j''}\boldsymbol{h}_k(\boldsymbol{z})D_{j'}\boldsymbol{h}_k(\boldsymbol{z})D_j\boldsymbol{h}_k(\boldsymbol{z}) = 0$ for all $(j,j',j'') \in \mathcal{D}^c \times [d_z]$. In particular, we have

$$D_{j'}\boldsymbol{h}_k(\boldsymbol{z})^2 D_j\boldsymbol{h}_k(\boldsymbol{z}) = 0,$$

for all $(j,j') \in \mathcal{D}^c$. Since this is true for all $k$ and all distinct $j,j'$, this means each row of $D\boldsymbol{h}(\boldsymbol{z})$ has at most one nonzero entry. Since $D\boldsymbol{h}(\boldsymbol{z})$ is invertible, these nonzero entries must appear on different columns, otherwise a column would be filled with zeros. This means $D\boldsymbol{h}(\boldsymbol{z})$ is a permutation-scaling matrix, i.e. we have local disentanglement (Definition A.13).

**Case 2:** $|B_k| > 1$ (Multi-Dimensional Slots)

Assume for a contradiction that $\hat{\boldsymbol{f}}$ does not disentangled $\boldsymbol{z}$ on $\mathcal{Z}_{\text{supp}}$ w.r.t. $\boldsymbol{f}$. This implies that there exist a $\boldsymbol{z}^* \in \hat{\mathcal{Z}}_{\text{supp}}$, $k,k',k'' \in [K]$ with $k' \neq k''$ such that:

$$D_{B_{k'}}\boldsymbol{h}_{B_k}(\boldsymbol{z}^*) \neq 0, \quad D_{B_{k''}}\boldsymbol{h}_{B_k}(\boldsymbol{z}^*) \neq 0 \tag{A.60}$$

Because $\boldsymbol{f}, \hat{\boldsymbol{f}}$ are $C^3$ diffeomorphisms, we know that $\boldsymbol{h}$ is also a $C^3$ diffeomorphism. Coupling this with Eq. (A.60), Lemma A.7 tells us that there exist an $S \subset [d_z]$ with cardinality $|B_k|$ such that:

$$\forall B \in \mathcal{B}, \ S \not\subseteq B, \ \text{and} \ \forall i \in S, D_i\boldsymbol{h}_{B_k}(\boldsymbol{z}^*) \text{ are linearly independent.} \tag{A.61}$$

Now choose any $\bar{B} \in \mathcal{B}$ such that $S_1 := \{S \cap \bar{B}\} \neq \emptyset$. Furthermore, define the set $S_2 := S \setminus S_1$. Because $S \not\subseteq \bar{B}$, we know that $S_2$ is non-empty. Further, by construction $S = S_1 \cup S_2$. In other words, $S_1$ and $S_2$ are non-empty, form a partition of $S$, and do not contain any indices from the same slot.

Now construct the matrices, denoted $\boldsymbol{A}_{S_1}$ and $\boldsymbol{A}_{S_2}$ as follows:

$$\boldsymbol{A}_{S_1} := D_{S_1}\boldsymbol{h}_{B_k}(\boldsymbol{z}^*), \quad \boldsymbol{A}_{S_2} := D_{S_2}\boldsymbol{h}_{B_k}(\boldsymbol{z}^*) \tag{A.62}$$

And the matrix denoted $\boldsymbol{A}_k$ as:

$$\boldsymbol{A}_k := [\boldsymbol{A}_{S_1}, \boldsymbol{A}_{S_2}] \tag{A.63}$$

Note that because, $\forall i \in S, D_i\boldsymbol{h}_{B_k}(\hat{\boldsymbol{z}}^*)$ are linearly independent (Eq. (A.61)), we know that $\boldsymbol{A}_k$ is invertible.

Now, define the following block diagonal matrix $\boldsymbol{A} \in \mathbb{R}^{d_z \times d_z}$ as follows:

$$\boldsymbol{A} := \begin{bmatrix} \boldsymbol{A}_1 & 0 & \dots & 0 \\ 0 & \boldsymbol{A}_2 & \dots & 0 \\ \vdots & \vdots & \ddots & \vdots \\ 0 & 0 & \dots & \boldsymbol{A}_K \end{bmatrix} \tag{A.64}$$

where $\forall i \in [K] \setminus \{k\}$, $\boldsymbol{A}_i$ is the identity matrix, and thus invertible, while $\boldsymbol{A}_k$ is defined according to Eq. (A.63).

Define $\bar{\mathcal{Z}} := \boldsymbol{A}^{-1}\mathcal{Z}$, the function $\bar{\boldsymbol{h}} : \bar{\mathcal{Z}} \to \mathcal{Z}$ as $\bar{\boldsymbol{h}}(\boldsymbol{z}) := \boldsymbol{A}\boldsymbol{z}$ and the function $\bar{\boldsymbol{f}} : \bar{\mathcal{Z}} \to \mathcal{X}$ as $\bar{\boldsymbol{f}} := \boldsymbol{f} \circ \bar{\boldsymbol{h}}$. By construction we have

$$\forall \boldsymbol{z} \in \mathcal{Z}, \qquad \bar{\boldsymbol{f}}\left(\boldsymbol{A}_1^{-1}\boldsymbol{z}_{B_1}, \dots, \boldsymbol{A}_K^{-1}\boldsymbol{z}_{B_K}\right) = \boldsymbol{f}(\boldsymbol{z}_{B_1}, \dots, \boldsymbol{z}_{B_K}). \tag{A.65}$$

Because all $\boldsymbol{A}_i^{-1}$ are invertible, then $\bar{\boldsymbol{f}}$ is *equivalent* to $\boldsymbol{f}$ in the sense of Def. (4.1).

We can now apply Lemma A.23 to $\bar{\boldsymbol{f}} = \boldsymbol{f} \circ \bar{\boldsymbol{h}}$ to obtain, for all $j, j', j'' \in [d_z]$:

$$D_{j,j',j''}^3 \bar{\boldsymbol{f}}(\boldsymbol{z}) = \boldsymbol{W}^{\boldsymbol{f}}(\bar{\boldsymbol{h}}(\boldsymbol{z}))\boldsymbol{m}^{\bar{\boldsymbol{h}}}(\boldsymbol{z}, (j, j', j'')). \tag{A.66}$$

Choose $\bar{\boldsymbol{z}} \in \bar{\mathcal{Z}}$ such that $\bar{\boldsymbol{h}}(\bar{\boldsymbol{z}}) = \boldsymbol{h}(\boldsymbol{z}^*)$, which is possible because $\boldsymbol{h}(\boldsymbol{z}^*) \in \mathcal{Z}$ and $\bar{\boldsymbol{h}}$ is a bijection from $\bar{\mathcal{Z}}$ to $\mathcal{Z}$. We can then write

$$D_{j,j',j''}^3 \bar{\boldsymbol{f}}(\bar{\boldsymbol{z}}) = \boldsymbol{W}^{\boldsymbol{f}}(\boldsymbol{h}(\boldsymbol{z}^*))\boldsymbol{m}^{\bar{\boldsymbol{h}}}(\bar{\boldsymbol{z}}, (j, j', j'')). \tag{A.67}$$

Let $J, J' \subseteq B_k$ be a partition of $B_k$ such that $J$ is the set of columns of $\boldsymbol{A}$ corresponding to $\boldsymbol{A}_{S_1}$ and $J'$ be the set of columns of $\boldsymbol{A}$ corresponding to $\boldsymbol{A}_{S_2}$. More formally, we have

$$\boldsymbol{A}_{B_k,J} = \boldsymbol{A}_{S_1} \qquad \text{and} \qquad \boldsymbol{A}_{B_k,J'} = \boldsymbol{A}_{S_2}$$

Since $\boldsymbol{A}_{S_1} = D_{S_1}\boldsymbol{h}_{B_k}(\boldsymbol{z}^*)$ and $\boldsymbol{A}_{S_2} = D_{S_2}\boldsymbol{h}_{B_k}(\boldsymbol{z}^*)$, we have that

$$\boldsymbol{A}_{B_k,J} = D_{S_1}\boldsymbol{h}_{B_k}(\boldsymbol{z}^*) \quad \text{and} \quad \boldsymbol{A}_{B_k,J'} = D_{S_2}\boldsymbol{h}_{B_k}(\boldsymbol{z}^*)$$

Since $D\bar{\boldsymbol{h}}(\bar{\boldsymbol{z}}) = \boldsymbol{A}$, we have

$$D_J\bar{\boldsymbol{h}}_{B_k}(\bar{\boldsymbol{z}}) = D_{S_1}\boldsymbol{h}_{B_k}(\boldsymbol{z}^*) \quad \text{and} \quad D_{J'}\bar{\boldsymbol{h}}_{B_k}(\bar{\boldsymbol{z}}) = D_{S_2}\boldsymbol{h}_{B_k}(\boldsymbol{z}^*).$$

For all $(j, j', j'') \in J \times J' \times B_k$ there must exist $(s, s', s'') \in S_1 \times S_2 \times S$ such that

$$D_j\bar{\boldsymbol{h}}_{B_k}(\bar{\boldsymbol{z}}) = D_s\boldsymbol{h}_{B_k}(\boldsymbol{z}^*), \quad D_{j'}\bar{\boldsymbol{h}}_{B_k}(\bar{\boldsymbol{z}}) = D_{s'}\boldsymbol{h}_{B_k}(\boldsymbol{z}^*), \text{ and } D_{j''}\bar{\boldsymbol{h}}_{B_k}(\bar{\boldsymbol{z}}) = D_{s''}\boldsymbol{h}_{B_k}(\boldsymbol{z}^*).$$

This implies that for all $(j, j', j'') \in J \times J' \times B_k$ there must exist $(s, s', s'') \in S_1 \times S_2 \times S$ such that

$$\boldsymbol{m}_k^{\bar{\boldsymbol{h}},\text{high}}(\bar{\boldsymbol{z}}, (j, j', j'')) = \boldsymbol{m}_k^{\boldsymbol{h},\text{high}}(\boldsymbol{z}^*, (s, s', s'')). \tag{A.68}$$

Moreover, since $\bar{\boldsymbol{h}}$ is a block-wise function, we have that, for all $(j, j', j'') \in J \times J' \times B_k \subseteq B_k^3$ and all $k' \in [K] \setminus \{k\}$, we have $\boldsymbol{m}_{k'}^{\bar{\boldsymbol{h}}}(\bar{\boldsymbol{z}}, (j, j', j'')) = 0$, which allows us to rewrite (A.67) as

$$D_{j,j',j''}^3 \bar{\boldsymbol{f}}(\bar{\boldsymbol{z}}) = \boldsymbol{W}_k^{\boldsymbol{f}}(\boldsymbol{h}(\boldsymbol{z}^*))\boldsymbol{m}_k^{\bar{\boldsymbol{h}}}(\bar{\boldsymbol{z}}, (j, j', j'')). \tag{A.69}$$

Since $\bar{\boldsymbol{h}}$ is linear, we have that $\boldsymbol{m}_k^{\bar{\boldsymbol{h}},\text{rest}}(\bar{\boldsymbol{z}}, (j, j', j'')) = 0$, and thus

$$D_{j,j',j''}^3 \bar{\boldsymbol{f}}(\bar{\boldsymbol{z}}) = \boldsymbol{W}_k^{\boldsymbol{f},\text{high}}(\boldsymbol{h}(\boldsymbol{z}^*))\boldsymbol{m}_k^{\bar{\boldsymbol{h}},\text{high}}(\bar{\boldsymbol{z}}, (j, j', j'')). \tag{A.70}$$

Plug (A.68) into the above to obtain that for all $(j, j', j'') \in J \times J' \times B_k$, there exists $(s, s', s'') \in S_1 \times S_2 \times S$ such that

$$D_{j,j',j''}^3 \bar{\boldsymbol{f}}(\bar{\boldsymbol{z}}) = \boldsymbol{W}^{\boldsymbol{f},\text{high}}(\boldsymbol{h}(\boldsymbol{z}^*))\boldsymbol{m}_k^{\boldsymbol{h},\text{high}}(\boldsymbol{z}^*, (s, s', s'')) = 0, \tag{A.71}$$

where the very last "$= 0$" is due to (A.59) (recall $(s, s', s'') \in S_1 \times S_2 \times S \subseteq \mathcal{D}^c \times [d_z]$).

In other words, we found a partition $J, J'$ of the block $B_k$ and a value $\bar{z}$ such that $D^3_{j,j',j''}\bar{\boldsymbol{f}}(\bar{z}) = 0$ for all $(j, j', j'') \in J \times J' \times B_k$. One can show that $\bar{\boldsymbol{f}}$ as no 3rd order interaction across blocks because it is equivalent to $\boldsymbol{f}$, which also has no 3rd order interactions across blocks. We thus have that $D^3_{j,j',j''}\bar{\boldsymbol{f}}(\bar{z}) = 0$ for all $(j, j', j'') \in J \times J' \times [d_z]$. This means that the blocks $J$ and $J'$ have *no third order interaction* in $\bar{\boldsymbol{f}}$ at $\bar{z}$. This is a contradiction with Assm. 3.5.

**From local to global disentanglement.** The same argument as in the proof of Theorem A.20 applies.

$\square$

## B    MULTI-INDEX NOTATION

Multi-index notation is a convenient shorthand to denote higher order derivatives. A multi-index of dimension $d$ is an ordered tuple $\boldsymbol{\alpha} = (\alpha_1, \ldots, \alpha_d) \in \mathbb{N}^d$. We introduce the shorthands

$$|\boldsymbol{\alpha}| = \sum_{i=1}^{d} \alpha_i, \quad \boldsymbol{\alpha}! = \prod_{i=1}^{d} \alpha_i! \tag{B.1}$$

and we write $\boldsymbol{\alpha} \geq \boldsymbol{\beta}$ if $\alpha_i \geq \beta_i$ for all $i$ and $\boldsymbol{\alpha} \pm \boldsymbol{\beta}$ denotes the element wise sum (difference) of the entries. We write

$$D^{\boldsymbol{\alpha}} = \frac{\partial^{\alpha_1}}{\partial z_1^{\alpha_1}} \cdots \frac{\partial^{\alpha_d}}{\partial z_d^{\alpha_d}} \tag{B.2}$$

and

$$\boldsymbol{z}^{\boldsymbol{\alpha}} = \prod_{i=1}^{d} z_i^{\alpha_i}. \tag{B.3}$$

We will need the important property that

$$D^{\boldsymbol{\alpha}} \boldsymbol{z}^{\boldsymbol{\beta}} = \begin{cases} \frac{\boldsymbol{\beta}!}{(\boldsymbol{\beta}-\boldsymbol{\alpha})!} \boldsymbol{z}^{\boldsymbol{\beta}-\boldsymbol{\alpha}} & \text{if } \boldsymbol{\beta} \geq \boldsymbol{\alpha} \\ 0 & \text{otherwise.} \end{cases} \tag{B.4}$$

Consider now a partition of $d_z$ into slots $B_1, \ldots, B_k$. We define the set of interaction multi-indices of order $n$ for $n \geq 2$ by

$$I_n = \{\boldsymbol{\alpha} \in \mathbb{N}^{d_z} : |\boldsymbol{\alpha}| = n, \exists i_1, i_2 \text{ s. t. } i_1 \in B_{k_1}, i_2 \in B_{k_2} \text{ with } k_1 \neq k_2 \text{ and } \alpha_{i_1}, \alpha_{i_2} > 0\}, \tag{B.5}$$

i.e., the set of all multi-indices such that the non-zero components are contained in at least two blocks. Clearly $I_n$ depends on the block partition which we do not reflect in the notation. We also consider

$$I_{\leq n} = \bigcup_{2 \leq m \leq n} I_m. \tag{B.6}$$

Clearly, if $\boldsymbol{\alpha} \in I_{|\boldsymbol{\alpha}|}$ and $\boldsymbol{\beta}$ is any multi-index, then $\boldsymbol{\alpha} + \boldsymbol{\beta} \in I_{|\boldsymbol{\alpha}|+|\boldsymbol{\beta}|}$.

## C    CHARACTERIZATION OF FUNCTIONS WITH AT MOST $n^{\text{TH}}$ ORDER INTERACTIONS

In this section we characterize functions with interaction of at most $n^{\text{th}}$ order by proving Theorem C.2. Our characterization relies on the notion of aligned-connectedness introduced in Definition A.16 and the following topological notion.

**Definition C.1.** A topological space $X$ is contractible if there is a continuous function $F : X \times [0, 1] \to X$ such that $F(x, 0) = x$ and $F(x, 1) = x_0$ for a point $x_0 \in X$. We call a subset of $\mathbb{R}^d$ contractible if it is contractible as a topological space with respect to the induced subspace topology.

Roughly, contractibility means that we can transform a topological space continuously into a point, which is possible if the space has no holes. Note that, e.g., all one dimensional connected sets and all convex sets are contractible. Sets that are not contractible are, e.g., spheres and disconnected sets. Note that the characterization in the following theorem generalizes Proposition 7 in Lachapelle et al. (2023) by allowing higher order interactions and showing the result for more general domains. We denote, similar to (2.3), for $\Omega \subset \mathbb{R}^{d_x}$ by $\Omega_i = \{z_{B_i} : z \in \Omega\}$ the projections of $\Omega$ on the blocks.

**Theorem C.2** (Characterization of functions with at most $n^{\text{th}}$ order interactions across slots.)**.** *Let $\Omega$ be an open connected and aligned-connected set such that $\Omega_k$ is contractible. Let $f(z) = f(z_{B_1}, z_{B_2}, ..., z_{B_K})$ be a $C^{n+1}$ function on $\Omega$ for an integer $n \in \mathbb{Z}_{\geq 1}$. Then any distinct slots $z_{B_i}$ and $z_{B_j}$ have at most $n^{\text{th}}$ order interaction within $f$ (Defn. 3.4) if and only if, for some constants $\left\{ c_{\boldsymbol{\alpha}} \in \mathbb{R}^{d_x} \right\}_{\boldsymbol{\alpha} \in I_{\leq n}}$ and some $C^{n+1}$ functions $f^k : \Omega_k \to \mathbb{R}^{d_x}$ such that for all $z \in \mathcal{Z}$*

$$f(z) = \sum_{k=1}^{K} f^k(z_{B_k}) + \sum_{\boldsymbol{\alpha} \in I_{\leq n}} c_{\boldsymbol{\alpha}} z^{\boldsymbol{\alpha}} . \tag{C.1}$$

*Remark* C.3. To avoid unnecessary complications we focus on the case where the ground truth $f$ is defined on $\mathcal{Z} = \mathbb{R}^{d_z}$. Then $\Omega = \mathcal{Z}$ clearly satisfies the assumptions and actually the proof is slightly simpler. The more general result here would allows us to handle also $\mathcal{Z} \subsetneq \mathbb{R}^{d_z}$ in Appendix D with minor changes.

The proof can be essentially decomposed in two steps: We show how to reduce from interaction of at most order $n$ to interaction of at most order $n - 1$ and then we establish the induction base for $n = 2$.

**Lemma C.4.** *Suppose $f : \Omega \to \mathbb{R}^{d_x}$ is a $C^{n+1}$ function and $\Omega$ open and connected. Assume that $f$ has interaction of at most order $n$ between any two different slots for some $n \geq 2$. Let $z_0 \in \Omega$ be any point. Then the function*

$$f(z) - \sum_{\boldsymbol{\alpha} \in I_n} \frac{D^{\boldsymbol{\alpha}} f(z_0)}{\boldsymbol{\alpha}!} z^{\boldsymbol{\alpha}} \tag{C.2}$$

*has interaction of order at most $n - 1$.*

*Proof.* First we observe that $f$ having interaction at most $n$ implies that $D^{\boldsymbol{\alpha}} f$ is constant in $\Omega$ for $\boldsymbol{\alpha} \in I_n$. Indeed, since $\boldsymbol{\alpha} \in I_n$ we conclude $\boldsymbol{\alpha} + e_i \in I_{n+1}$ where $e_i$ denotes the tuple with $i$-th entry 1 and all other entries 0. Then, by definition of having interaction at most in $n$ in Definition 3.4, we conclude that

$$\partial_i D^{\boldsymbol{\alpha}} f(z) = D^{\boldsymbol{\alpha} + e_i} f(z) = 0. \tag{C.3}$$

This implies that the total derivative of $D^{\boldsymbol{\alpha}} f$ vanishes on $\Omega$, which implies that $D^{\boldsymbol{\alpha}} f$ is constant because $\Omega$ is connected. Consider now any $\boldsymbol{\beta} \in I_n$. Then we find using (B.4)

$$D^{\boldsymbol{\beta}} \left( f(z) - \sum_{\boldsymbol{\alpha} \in I_n} \frac{D^{\boldsymbol{\alpha}} f(z_0)}{\boldsymbol{\alpha}!} z^{\boldsymbol{\alpha}} \right) = D^{\boldsymbol{\beta}} f(z) - \frac{D^{\boldsymbol{\beta}} f(z_0)}{\boldsymbol{\beta}!} \boldsymbol{\beta}! = 0 \tag{C.4}$$

where we used that $D^{\boldsymbol{\beta}} f$ is constant and $D^{\boldsymbol{\beta}} z^{\boldsymbol{\alpha}} = 0$ for $\boldsymbol{\alpha} \neq \boldsymbol{\beta}$ if $|\boldsymbol{\alpha}| = |\boldsymbol{\beta}|$. This ends the proof. $\square$

We now establish the functional form for interaction of at most order 1. This is essentially a similar statement as in Proposition 7 in Lachapelle et al. (2023) except that we consider more general domains so that their proof does not apply.

**Lemma C.5.** *Assume $\Omega$ is an open connected and aligned-connected set such that $\Omega_k$ is contractible. If $f$ is a function such that different slots have interaction at most of order 1 then there are functions $f^k$ such that*

$$f(z) = \sum_{k=1}^{K} f^k(z_{B_k}). \tag{C.5}$$

*Proof.* Fix a $1 \le k \le K$. With slight abuse of notation we write $\boldsymbol{z} = (\boldsymbol{z}_{B_k}, \boldsymbol{z}_{B_k^c})$. Fix now some value $\boldsymbol{z}_{B_k}$. We claim that for all $\boldsymbol{z}_{B_k^c}, \boldsymbol{z}'_{B_k^c}$ such that $(\boldsymbol{z}_{B_k}, \boldsymbol{z}_{B_k^c}), (\boldsymbol{z}_{B_k}, \boldsymbol{z}'_{B_k^c}) \in \Omega$

$$D_{\boldsymbol{z}_{B_k}} \boldsymbol{f}((\boldsymbol{z}_{B_k}, \boldsymbol{z}_{B_k^c})) = D_{\boldsymbol{z}_{B_k}} \boldsymbol{f}((\boldsymbol{z}_{B_k}, \boldsymbol{z}'_{B_k^c})). \tag{C.6}$$

By assumption we indeed know that

$$D_{\boldsymbol{z}_{B_k^c}} D_{\boldsymbol{z}_{B_k}} \boldsymbol{f}((\boldsymbol{z}_{B_k}, \boldsymbol{z}_{B_k^c})) = 0. \tag{C.7}$$

Moreover, by aligned-connectedness we know that the set $\Omega_{\boldsymbol{z}_{B_k}} = \{z_{B_k^c} : (\boldsymbol{z}_{B_k}, \boldsymbol{z}_{B_k^c}) \in \Omega\}$ is connected so we conclude that the function

$$\Omega_{\boldsymbol{z}_{B_k^c}} \to \mathbb{R}^{|B_k| \times d_x}, \quad \boldsymbol{z}_{B_k^c} \to D_{\boldsymbol{z}_{B_k}} \boldsymbol{f}((\boldsymbol{z}_{B_k}, \boldsymbol{z}_{B_k^c})) \tag{C.8}$$

is indeed constant. This implies that there is a function $\boldsymbol{g}^k$ depending on $\boldsymbol{z}_{B_k}$ such that

$$\boldsymbol{g}^k(\boldsymbol{z}_{B_k}) = D_{\boldsymbol{z}_{B_k}} \boldsymbol{f}((\boldsymbol{z}_{B_k}, \boldsymbol{z}_{B_k^c})) \tag{C.9}$$

for all $\boldsymbol{z} = (\boldsymbol{z}_{B_k}, \boldsymbol{z}_{B_k^c}) \in \Omega$. Locally $\boldsymbol{g}^k$ is the gradient of a function, but by assumption $\Omega_k$ is contractible and therefore, by the Poincaré-Lemma, there is a function $\boldsymbol{f}^k$ such that $D\boldsymbol{f}^k = \boldsymbol{g}^k$. Then we find

$$D_{\boldsymbol{z}_{B_k}} \boldsymbol{f}((\boldsymbol{z}_{B_k}, \boldsymbol{z}_{B_k^c})) = \boldsymbol{g}(\boldsymbol{z}_{B_k}) = D_{\boldsymbol{z}_{B_k}} \boldsymbol{f}^k(\boldsymbol{z}_{B_k}) = D_{\boldsymbol{z}_{B_k}} \left( \sum_{k'=1}^K \boldsymbol{f}^{k'}(\boldsymbol{z}_{B_{k'}}) \right). \tag{C.10}$$

Thus the difference $\boldsymbol{f} - \sum_{k=1}^K \boldsymbol{f}^k$ has vanishing derivative on $\Omega$ and since $\Omega$ is connected we conclude that it is constant. This implies (C.5) after shifting one $\boldsymbol{f}^k$ by this constant. $\qquad\square$

Based on these two lemmas the proof of Theorem C.2 is straightforward.

*Proof of Theorem C.2.* In the first step we show that if the at most $n$-th order interaction condition holds then $\boldsymbol{f}$ can be written as in (C.1), i.e., '$\Rightarrow$'. Applying inductively Lemma C.4 we conclude that there are constants $\boldsymbol{c_\alpha} \in \mathbb{R}^{d_x}$ such that

$$\boldsymbol{f}(\boldsymbol{z}) - \sum_{m=2}^n \sum_{\boldsymbol{\alpha} \in I_m} \boldsymbol{c_\alpha} \boldsymbol{z}^{\boldsymbol{\alpha}} \tag{C.11}$$

has interaction of order at most 1. Thus, we can apply Lemma C.5 which implies that a representation as in (C.1) exists on $\Omega$. For the reverse direction '$\Leftarrow$' we observe that clearly the functional form implies for $\boldsymbol{\beta} \in I_{n+1}$ the relation

$$D^{\boldsymbol{\beta}} \boldsymbol{f} = 0. \tag{C.12}$$
$$\square$$

Let us show through examples that the topological conditions on the set $\Omega$ are neccessary. The following examples shows that the condition that $\Omega_k$ is contractible cannot be dropped.

*Example* C.6. For every $\boldsymbol{z} \in \mathbb{R}^2 \setminus \{0\}$ we denote by $\theta(\boldsymbol{z}) \in [0, 2\pi)$ the argument (i.e., the angle to the positive $x$-axis in radian) and by $r(\boldsymbol{z}) = |\boldsymbol{z}|$ the radius of $\boldsymbol{z}$. We consider $\Omega \subset \mathbb{R}^4$ and $B_1 = \{1, 2\}, B_2 = \{3, 4\}$ given by

$$\Omega = \{\boldsymbol{z} : r(\boldsymbol{z}_{B_1}), r(\boldsymbol{z}_{B_2}) \in (1, 2), (\theta(\boldsymbol{z}_{B_1}) - \theta(\boldsymbol{z}_{B_2}) \mod 2\pi) \in (0, \pi)\} \tag{C.13}$$

and the function

$$\boldsymbol{f} : \Omega \to \mathbb{R}, \quad \boldsymbol{f}(\boldsymbol{z}) = \theta(\boldsymbol{z}_{B_1}) - \theta(\boldsymbol{z}_{B_2}) \mod 2\pi. \tag{C.14}$$

Then $\Omega$ is aligned-connected because the sets in questions are annular sectors and in particular path connected. Moreover, $\boldsymbol{f}$ is smooth because $\theta(\boldsymbol{z}_{B_1}) - \theta(\boldsymbol{z}_{B_2}) \mod 2\pi \in (0, \pi)$ so it does not jump and $D_{\boldsymbol{z}_{B_1}} D_{\boldsymbol{z}_{B_2}} \boldsymbol{f} = 0$ because it is locally additive. However it is not globally additive as in (C.1).

The necessity of the aligned connctedness condition can be shown by an example that is similar to Example 7 in Lachapelle et al. (2023).

*Example* C.7. Consider $\Omega = ([-1, 0] \times [-2, 2]) \cup ([0, 1] \times [1, 2]) \cup ([0, 1] \times [-2, -1])$ and $\boldsymbol{f} : \Omega \to \mathbb{R}$ given by

$$\boldsymbol{f}(\boldsymbol{z}) = \begin{cases} z_1^3 & \text{if } z_1, z_2 > 0 \\ 0 & \text{otherwise.} \end{cases} \tag{C.15}$$

Then $\boldsymbol{f}$ is $C^2$, $\boldsymbol{f}$ has interaction of order at most 1 but $\boldsymbol{f}$ cannot be written as in (C.1). Note that $\Omega$ is not aligned-connected because $\{z_2 : (1/2, z_2) \in \Omega\} = [-2, -1] \cup [1, 2]$ is not connected.

# D    COMPOSITIONAL GENERALIZATION PROOFS

In this appendix we prove extrapolation result Theorem 4.4. Based on the functional form derived in Theorem C.2 we relate two different disentangled representations.

**Lemma D.1.** *Let $\boldsymbol{f} : \mathcal{Z} \to \mathbb{R}^{d_x}$ be a $C^3$ diffeomorphism of the form:*

$$\boldsymbol{f}(\boldsymbol{z}) = \sum_{k=1}^{K} \boldsymbol{f}^k(\boldsymbol{z}_{B_k}) + \sum_{\boldsymbol{\alpha} \in I_2} \boldsymbol{c}_{\boldsymbol{\alpha}} \boldsymbol{z}^{\boldsymbol{\alpha}} \tag{D.1}$$

*for some $\boldsymbol{f}^i$ in $C^3$. Let $\hat{\boldsymbol{f}} : \mathcal{Z} \to \mathbb{R}^{d_x}$ be a diffeomorphism of the same functional form. Let $\boldsymbol{h} : \mathcal{Z}_{\mathrm{supp}} \to \mathcal{Z}$ be such that $\boldsymbol{f} = \hat{\boldsymbol{f}} \circ \boldsymbol{h}$ on $\mathcal{Z}_{\mathrm{supp}}$. If $\boldsymbol{h}$ is a slot-wise function, i.e. for all $k \in [K], \boldsymbol{h}_k(\boldsymbol{z}) = \boldsymbol{h}_k(\boldsymbol{z}_{B_k})$ and $\mathcal{Z}_{\mathrm{supp}}$ is regularly closed then for all $\boldsymbol{z} \in \mathcal{Z}_{\mathrm{supp}}$*

$$\sum_{k=1}^{K} \boldsymbol{f}^k(\boldsymbol{z}_{B_k}) = \sum_{k=1}^{K} \hat{\boldsymbol{f}}^{\pi(k)}(\boldsymbol{h}_k(\boldsymbol{z}_{B_k})) + L(\boldsymbol{z}) \tag{D.2}$$

*for some affine function $L : \mathbb{R}^{d_z} \to \mathbb{R}^{d_x}$.*

*Remark D.2.* We note that it is not possible to remove the affine function $L$ from the statement. Indeed if all slots have dimension 1 and $h_1(z_1) = z_1 + 1$, $h_2(z_2) = z_2 + 1$ then $h_1(z_1)h_2(z_2) - z_1 z_2 = z_1 + z_2 + 1$ is an additive function. Moreover, we cannot in general prove that $\boldsymbol{h}$ itself is slotwise affine because the coefficients $\boldsymbol{c}$ can be zero. In this case $\boldsymbol{h}$ can be any slot-wise diffeomorphism.

*Proof.* First we remark that the polynomial part of the functional form in (D.1) contains all terms $z_i z_j$ where $i, j$ are in different slots, thus it can be equivalently written as

$$\sum_{\boldsymbol{\alpha} \in I_2} \boldsymbol{c}_{\boldsymbol{\alpha}} \boldsymbol{z}^{\boldsymbol{\alpha}} = \sum_{k=1}^{K} \sum_{k'=k+1}^{K} \left( \boldsymbol{z}_{B_k} \otimes \boldsymbol{z}_{B_{k'}} \right) \boldsymbol{A}_{kk'} \tag{D.3}$$

for some constant matrices $\{ \boldsymbol{A}_{kk'} \in \mathbb{R}^{(|B_k| \cdot |B_{k'}|) \times N} \}_{k < k' \in [K]}$, where $\otimes$ denotes the Kronecker product (e.g., $[z_1, z_2] \otimes [z_3, z_4] = [z_1 z_3, z_1 z_4, z_2 z_3, z_2 z_4]$).

We assume that the permutation $\pi$ is the identity. We know that $\boldsymbol{f}, \hat{\boldsymbol{f}}$ are diffeomorphisms between the same spaces and can thus be related by the function $\boldsymbol{h}$ via:

$$\boldsymbol{f} = \hat{\boldsymbol{f}} \circ \boldsymbol{h} \tag{D.4}$$

Inserting the functional forms for $\boldsymbol{f}, \hat{\boldsymbol{f}}$ and leveraging that $\boldsymbol{h}$ is a slot-wise function and $\pi$ is the identity, we have for all $\boldsymbol{z} \in \mathcal{Z}$

$$\sum_{k=1}^{K} \boldsymbol{f}^k(\boldsymbol{z}_{B_k}) + \sum_{k=1}^{K} \sum_{k'=k+1}^{K} \left( \boldsymbol{z}_{B_k} \otimes \boldsymbol{z}_{B_{k'}} \right) \boldsymbol{A}_{kk'}$$
$$= \sum_{k=1}^{K} \hat{\boldsymbol{f}}^k(\boldsymbol{h}_{B_k}(\boldsymbol{z}_{B_k})) + \sum_{k=1}^{K} \sum_{k'=k+1}^{K} \left( \boldsymbol{h}^k(\boldsymbol{z}_{B_k}) \otimes \boldsymbol{h}^{k'}(\boldsymbol{z}_{B_{k'}}) \right) \hat{\boldsymbol{A}}_{kk'}. \tag{D.5}$$

To prove the claim we now consider the expression

$$\begin{aligned} L(\boldsymbol{z}) &= \sum_{k=1}^{K} \boldsymbol{f}^k(\boldsymbol{z}_{B_k}) - \sum_{k=1}^{K} \hat{\boldsymbol{f}}^k(h_{B_k}(\boldsymbol{z}_{B_k})) \\ &= \sum_{i=k}^{K} \sum_{k'=k+1}^{K} \left( \boldsymbol{h}^k(\boldsymbol{z}_{B_k}) \otimes \boldsymbol{h}^{k'}(\boldsymbol{z}_{B_{k'}}) \right) \hat{\boldsymbol{A}}_{kk'} - \sum_{k=1}^{K} \sum_{k'=k+1}^{K} \left( \boldsymbol{z}_{B_k} \otimes \boldsymbol{z}_{B_{k'}} \right) \boldsymbol{A}_{kk'} \end{aligned} \tag{D.6}$$

and prove that $L(\boldsymbol{z})$ is an affine function. To show this it is sufficient to prove that the second derivative $D^2 L$ vanishes because $\mathcal{Z}_{\text{supp}}$ is path-connected. Thus we consider all partial derivatives. Consider first the case where $i \in B_k$ and $i' \in B_{k'}$ for $k < k'$. Then we find that

$$D_i D_{i'} L(\boldsymbol{z}) = D_i D_{i'} \left( \sum_{k=1}^{K} \boldsymbol{f}^k (\boldsymbol{z}_{B_k}) - \sum_{k=1}^{K} \hat{\boldsymbol{f}}^k (\boldsymbol{h}_{B_k}(\boldsymbol{z}_{B_k})) \right) = 0. \tag{D.7}$$

It remains to consider derivatives of the form $D_i D_{i'}$ where $i, i' \in B_k$ for some slot $i$. Then we clearly have

$$D_i D_{i'} \sum_{k=1}^{K} \sum_{k'=k+1}^{K} \left( \boldsymbol{z}_{B_k} \otimes \boldsymbol{z}_{B_{k'}} \right) \boldsymbol{A}_{kk'} = 0 \tag{D.8}$$

because this is a linear expression in $z_{B_k}$. Next, we want to show that

$$D_i D_{i'} \left( \boldsymbol{h}^k(\boldsymbol{z}_{B_k}) \otimes \boldsymbol{h}^{k'}(\boldsymbol{z}_{B_{k'}}) \right) \hat{\boldsymbol{A}}_{kk'} = 0 \tag{D.9}$$

for all $k < k$. To prove this we show the more general statement (that will be used in the proof of Theorem 4.4 below) that for any $k \neq k'$ and any vector $v \in \mathbb{R}^{B_{k'}}$ the functions

$$\boldsymbol{z}_{B_k} \to (\boldsymbol{h}^k(\boldsymbol{z}_{B_k}) \otimes v) \hat{\boldsymbol{A}}_{kk'} \tag{D.10}$$

are affine on $\mathcal{Z}_k$ or equivalently that

$$D_i D_{i'} \left( \boldsymbol{h}^k(\boldsymbol{z}_{B_k}) \otimes v \right) \hat{\boldsymbol{A}}_{kk'} = 0 \tag{D.11}$$

for every $v \in \mathbb{R}^{B_{k'}}$. To prove this we consider any $j \in B_{k'}$ and apply the derivative $D_i D_{i'} D_j$ to (D.5) to get

$$0 = \left( D_i D_{i'} \boldsymbol{h}^k(\boldsymbol{z}_{B_k}) \otimes D_j \boldsymbol{h}^{k'}(\boldsymbol{z}_{B_k}) \right) \hat{\boldsymbol{A}}_{kk'} \tag{D.12}$$

for every $\boldsymbol{z} \in \mathring{\mathcal{Z}}_{\text{supp}}$. Now we use that by assumption $\boldsymbol{h}$ is a diffeomorphism. Using the block structure of $\boldsymbol{h}$ we find that also $\boldsymbol{h}^k$ are diffeomorphisms. In particular, this implies that for any $\boldsymbol{z} \in \mathring{\mathcal{Z}}_{\text{supp}}$ the vectors $(D_j \boldsymbol{h}^{k'}(\boldsymbol{z}_{B_{k'}}))_{j \in B_{k'}}$ are linearly independent vectors in $\mathbb{R}^{|B_{k'}|}$ and they thus generate $\mathbb{R}^{|B_{k'}|}$. Therefore we can find coefficients $\alpha_j$ (depending on $\boldsymbol{z}_{B_{k'}}$) such that

$$\sum_{j \in B_{k'}} \alpha_j D_j \boldsymbol{h}^{k'}(\boldsymbol{z}_{B_{k'}}) = v \tag{D.13}$$

Then we get using (D.12)

$$D_i D_{i'} \left( \boldsymbol{h}^k(\boldsymbol{z}_{B_k}) \otimes v \right) \hat{\boldsymbol{A}}_{kk'} = D_i D_{i'} \left( \boldsymbol{h}^k(\boldsymbol{z}_{B_k}) \otimes \left( \sum_{j \in B_{k'}} \alpha_j D_j \boldsymbol{h}^{k'}(\boldsymbol{z}_{B_{k'}}) \right) \right) \hat{\boldsymbol{A}}_{kk'}$$
$$= \sum_{j \in B_{k'}} \alpha_j \left( D_i D_{i'} \boldsymbol{h}^k(\boldsymbol{z}_{B_k}) \otimes D_j \boldsymbol{h}^{k'}(\boldsymbol{z}_{B_{k'}}) \right) \hat{\boldsymbol{A}}_{kk'} = 0. \tag{D.14}$$

So (D.10) holds and thus also (D.9) (we actually only get this for points $z_k \in \mathcal{Z}_k$ such that there is $\boldsymbol{z} \in \mathring{\mathcal{Z}}_{\text{supp}}$ with $z_k = \boldsymbol{z}_{B_k}$ but by continuity and since $\mathcal{Z}_{\text{supp}}$ is regularly closed this actually holds on $\mathcal{Z}_k$). The same reasoning shows that this is also true if $i, i' \in B_{k'}$ (instead of $i, i' \in B_k$). We then find that for $i, i' \in B_k$

$$D_i D_{i'} \sum_{k=1}^{K} \sum_{k'=k+1}^{K} \left( \boldsymbol{h}^k(\boldsymbol{z}_{B_k}) \otimes \boldsymbol{h}^{k'}(\boldsymbol{z}_{B_{k'}}) \right) \hat{\boldsymbol{A}}_{kk'} =$$
$$= \sum_{k=1}^{K} \sum_{k'=k+1}^{K} D_i D_{i'} \left( \boldsymbol{h}^k(\boldsymbol{z}_{B_k}) \otimes \boldsymbol{h}^{k'}(\boldsymbol{z}_{B_{k'}}) \right) \hat{\boldsymbol{A}}_{kk'} = 0. \tag{D.15}$$

The last display together with (D.8) and (D.7) imply that $D^2 L = 0$ and thus $L$ is affine. When $\pi$ is not the identity the proof is similar. □

We also need the following simple lemma which states that we have unique Cartesian-product extension of functions with interaction of order at most $n$ between different slots.

**Lemma D.3.** *Let $\boldsymbol{f} : \mathcal{Z} \to \mathbb{R}^{d_x}$ be a $C^3$ diffeomorphism with interaction at most $n$ between different slots such that $\mathcal{Z}_{\text{supp}}$ is regularly closed and for $\boldsymbol{z} \in \mathcal{Z}_{\text{supp}}$*

$$\boldsymbol{f}(\boldsymbol{z}) = \sum_{k=1}^{K} \boldsymbol{f}^k (\boldsymbol{z}_{B_k}) + \sum_{2 \leq m \leq n} \sum_{\boldsymbol{\alpha} \in I_m} \boldsymbol{c}_{\boldsymbol{\alpha}} \boldsymbol{z}^{\boldsymbol{\alpha}} \tag{D.16}$$

*for some $\boldsymbol{f}^i$ in $C^3$. Then this relation holds on $\mathcal{Z}_{\text{CPE}}$.*

*Proof.* We know by Theorem C.2 that a representation as in (D.16) holds on $\mathcal{Z} = \mathbb{R}^{d_z}$ and thus can be restricted to $\mathcal{Z}_{\text{CPE}}$, however it might not be the same representation but involve functions $\tilde{\boldsymbol{f}}^k$ and constants $\tilde{\boldsymbol{c}}_{\boldsymbol{\alpha}}$. Taking the difference and setting $\bar{\boldsymbol{f}}^k = \boldsymbol{f}^k - \tilde{\boldsymbol{f}}^k$ and $\bar{\boldsymbol{c}}_{\boldsymbol{\alpha}} = \boldsymbol{c}_{\boldsymbol{\alpha}} - \tilde{\boldsymbol{c}}_{\boldsymbol{\alpha}}$ we find that on $\mathcal{Z}_{\text{supp}}$

$$0 = \sum_{k=1}^{K} \bar{\boldsymbol{f}}^k (\boldsymbol{z}_{B_k}) + \sum_{2 \leq m \leq n} \sum_{\boldsymbol{\alpha} \in I_m} \bar{\boldsymbol{c}}_{\boldsymbol{\alpha}} \boldsymbol{z}^{\boldsymbol{\alpha}}. \tag{D.17}$$

But by applying $D^{\boldsymbol{\alpha}}$ for $\boldsymbol{\alpha} \in I_m$ for $m = n$ down to $m = 2$ we find $\bar{\boldsymbol{c}}_{\boldsymbol{\alpha}} = 0$ for all $\boldsymbol{\alpha} \in I_{\leq n}$ and thus the polynomial term vanishes. Next, we apply $D$ and find that $\bar{\boldsymbol{f}}^k$ is constant on $\mathcal{Z}_k$ (because $\mathcal{Z}_{\text{supp}}$ is regularly closed). This implies that (D.17) holds in $\mathcal{Z}_{\text{CPE}}$ and thus (D.16) holds on $\mathcal{Z}_{\text{CPE}}$. □

Using the previous lemmas we can prove Theorem 4.4.

**Theorem 4.4** (Compositional Generalization). *Let $n \in \{0, 1, 2\}$. Let $\mathcal{Z}_{\text{supp}}$ be regular closed (Defn. A.3). Let $\boldsymbol{f} : \mathcal{Z} \to \mathcal{X}$ and $\hat{\boldsymbol{f}} : \mathcal{Z} \to \mathbb{R}^{d_x}$ be $C^3$ diffeomorphisms with at most $n^{\text{th}}$ order interactions across slots on $\mathcal{Z}$. If $\hat{\boldsymbol{f}}$ disentangles $\boldsymbol{z}$ on $\mathcal{Z}_{\text{supp}}$ w.r.t. $\boldsymbol{f}$ (Defn. 2.1), then it generalizes compositionally (Defn. 2.2).*

*Proof of Theorem 4.4.* Note that Corollary 3 in Lachapelle et al. (2023) already handles the case $n = 0, 1$ but the proof below is more general, and also covers the case of $n = 0, 1$, since functions with at most $0^{\text{th}}$ and $1^{\text{st}}$ order interactions are special cases of functions with at most $2^{\text{nd}}$ order interactions assuming $\boldsymbol{f}$ is a $C^3$ diffeomorphism.

We can apply Theorem C.2 to $\boldsymbol{f}$ which implies that $\boldsymbol{f}$ can be written on $\mathcal{Z} = \mathbb{R}^{d_z}$ as in (C.1) and as explained in Lemma D.1 an equivalent representation is

$$\boldsymbol{f}(\boldsymbol{z}) = \sum_{k=1}^{K} \boldsymbol{f}^k (\boldsymbol{z}_{B_k}) + \sum_{k=1}^{K} \sum_{k'=k+1}^{K} (\boldsymbol{z}_{B_k} \otimes \boldsymbol{z}_{B_{k'}}) \boldsymbol{A}_{kk'}. \tag{D.18}$$

and we have similarly

$$\hat{\boldsymbol{f}}(\boldsymbol{z}) = \sum_{k=1}^{K} \hat{\boldsymbol{f}}^k (\boldsymbol{z}_{B_k}) + \sum_{k=1}^{K} \sum_{k'=k+1}^{K} (\boldsymbol{z}_{B_k} \otimes \boldsymbol{z}_{B_{k'}}) \hat{\boldsymbol{A}}_{kk'}. \tag{D.19}$$

By assumption we have $\boldsymbol{f} = \hat{\boldsymbol{f}} \circ \boldsymbol{h}$ on $\mathcal{Z}_{\text{supp}}$ where $\boldsymbol{h}(\boldsymbol{z}) := \left( \boldsymbol{h}_1 (\boldsymbol{z}_{B_{\pi(1)}}), \ldots, \boldsymbol{h}_K (\boldsymbol{z}_{B_{\pi(K)}}) \right)$ and the functions $\boldsymbol{h}_k : \mathbb{R}^{|B_{\pi(k)}|} \to \mathbb{R}^{|B_k|}$ are diffeomorphisms. Our goal is to show that this relation actually holds on the Cartesian-product extensions $\mathcal{Z}_{\text{CPE}}$. Let $\mathcal{U}$ be the set of points such that $\boldsymbol{f}(\boldsymbol{z}) = \hat{\boldsymbol{f}} \circ \boldsymbol{h}(\boldsymbol{z})$ for $\boldsymbol{z} \in \mathcal{U}$. We claim that if $\boldsymbol{z} = (\boldsymbol{z}_{B_1}, \ldots, \boldsymbol{z}_{B_K}) \in \mathring{\mathcal{U}}$ then $\boldsymbol{z}' = (\boldsymbol{z}_{B_1}, \ldots, \boldsymbol{z}'_{B_l}, \ldots, \boldsymbol{z}_{B_K}) \in \mathcal{U}$ for any $\boldsymbol{z}'_{B_l} \in \mathcal{Z}_l$. Let us define the map $e^{\boldsymbol{z}} : \mathcal{Z}_l \to \mathcal{Z}$ given by $e^{\boldsymbol{z}}(\boldsymbol{z}'_{B_l}) = \boldsymbol{z}'$. We know by Lemma D.1 that the function

$$\boldsymbol{z} \to \sum_{k=1}^{K} \boldsymbol{f}^k(\boldsymbol{z}_{B_k}) - \sum_{k=1}^{K} \hat{\boldsymbol{f}}^{\pi(k)}(\boldsymbol{h}_k(\boldsymbol{z})_{B_k})) = L(\boldsymbol{z}) \tag{D.20}$$

is affine on $\mathcal{Z}_{\text{supp}}$. Applying Lemma D.3 the same holds on $\mathcal{Z}_{\text{CPE}}$. Thus we conclude that

$$\boldsymbol{z}'_{B_l} \to \sum_{k=1}^{K} \boldsymbol{f}^k(e^{\boldsymbol{z}}(\boldsymbol{z}'_{B_l})_{B_k}) - \sum_{k=1}^{K} \hat{\boldsymbol{f}}^{\pi(k)}(\boldsymbol{h}_k(e^{\boldsymbol{z}}(\boldsymbol{z}'_{B_l})_{B_k})) = L(e^{\boldsymbol{z}}(\boldsymbol{z}'_{B_i})) \tag{D.21}$$

is affine on $\mathcal{Z}_l$. Moreover,

$$\boldsymbol{z}'_{B_l} \to \sum_{k=1}^{K} \sum_{k'=k+1}^{K} \left( e^{\boldsymbol{z}}(\boldsymbol{z}'_{B_l})_{B_k} \otimes e^{\boldsymbol{z}}(\boldsymbol{z}'_{B_l})_{B_{k'}} \right) \boldsymbol{A}_{kk'} \tag{D.22}$$

is clearly affine on $\mathcal{Z}_l$ and by (D.10) the same holds for

$$\boldsymbol{z}'_{B_l} \to \sum_{k=1}^{K} \sum_{k'=k+1}^{K} \left( \boldsymbol{h}^k(e^{\boldsymbol{z}}(\boldsymbol{z}'_{B_l})_{B_k}) \otimes \boldsymbol{h}^{k'}(e^{\boldsymbol{z}}(\boldsymbol{z}'_{B_l})_{B_{k'}}) \right) \boldsymbol{A}_{kk'}. \tag{D.23}$$

The last three displays together imply that

$$\boldsymbol{z}'_{B_l} \to \boldsymbol{f}(e^{\boldsymbol{z}}(\boldsymbol{z}'_{B_l})) - \hat{\boldsymbol{f}} \circ \boldsymbol{h}(e^{\boldsymbol{z}}(\boldsymbol{z}'_{B_l})) \tag{D.24}$$

is affine on $\mathcal{Z}_l$ and since it is zero in a neighbourhood of $\boldsymbol{z}'_{B_l} = \boldsymbol{z}_{B_l}$ (because $\boldsymbol{z} \in \mathring{\mathcal{U}}$) it is equal to zero on $\mathcal{Z}_l$. Since this is true for any slot $B_l$ we can now conclude that $\mathcal{U} = \mathcal{Z}$. Indeed, pick any open rectangle $\mathcal{Z}'_1 \times \mathcal{Z}'_2 \times \ldots \times \mathcal{Z}'_K \subset \mathcal{Z}_{\text{supp}} \subset \mathcal{U}$. We then infer that $\mathring{\mathcal{Z}}_1 \times \mathcal{Z}'_2 \times \ldots \times \mathcal{Z}'_K \subset \mathcal{U}$ and by inducting over the slots and applying continuity at the boundary we obtain the claim. $\qquad\square$

## E    UNIFYING ASSUMPTIONS FROM PRIOR WORK

### E.1    AT MOST $0^{\text{TH}}$ ORDER INTERACTION ACROSS SLOTS

To prove that the assumptions in Brady et al. (2023) are a special case of our assumptions for $n = 0$, we first restate their assumptions formally. To this end, we first define the following set:

$$\forall S \subseteq [d_z] \quad I_S(\boldsymbol{z}) := \{ l \in [d_x] : D_S \boldsymbol{f}_l(\boldsymbol{z}) \neq 0 \}. \tag{E.1}$$

The assumption of *compositionality* in Brady et al. (2023) can now be stated:

**Definition E.1** (Compositionality). A differentiable function $\boldsymbol{f} : \mathcal{Z} \to \mathcal{X}$, is said to be *compositional* if:

$$\forall \boldsymbol{z} \in \mathcal{Z}, k, j \neq k \in [K] : I_k(\boldsymbol{z}) \cap I_j(\boldsymbol{z}) = \emptyset. \tag{E.2}$$

We now state the second assumption in Brady et al. (2023), deemed *irreducibility*.

**Definition E.2** (Irreducibility). A differentiable function $\boldsymbol{f} : \mathcal{Z} \to \mathcal{X}$, is said to be *irreducible* if for all $\boldsymbol{z} \in \mathcal{Z}$ and $k \in [K]$ and any partition $I_k(\boldsymbol{z}) = S_1 \cup S_2$ (i.e., $S_1 \cap S_2 = \emptyset$ and $S_1, S_2 \neq \emptyset$), we have:

$$\text{rank}\left(D\boldsymbol{f}_{S_1}(\boldsymbol{z})\right) + \text{rank}\left(D\boldsymbol{f}_{S_2}(\boldsymbol{z})\right) > \text{rank}\left(D\boldsymbol{f}_{I_k}(\boldsymbol{z})\right). \tag{E.3}$$

We now prove that compositionality and irreducibility are equivalent to $\boldsymbol{f}$ having satisfying interaction asymmetry (3.5) for all equivalent generators (4.1) for $n = 0$.

**Theorem E.3.** *A $C^1$ diffeomorphism $\boldsymbol{f} : \mathcal{Z} \to \mathcal{X}$ satisfies compositionality (Def. E.1) and irreducibility (Def. E.2) if and only if $\boldsymbol{f}$ has at most $0^{th}$ order interaction across slots (Defn. 3.2) and satisfies interaction asymmetry (Assm. 3.5) for all equivalent generators (4.1).*

*Proof.* We start by proving the forward direction, i.e., that compositionality and irreducibility imply that $\boldsymbol{f}$ has at most $0^{\text{th}}$ order interaction across slots and satisfies interaction asymmetry for all equivalent generators.

The definitions of compositionality and at most $0^{\text{th}}$ order interaction across slots are precisely equivalent, thus we only need to show that compositionality and irreducibility imply that $\boldsymbol{f}$ satisfies interaction asymmetry for all equivalent generators. To show this we will prove the following contraposition: that if $\boldsymbol{f}$ has at most $0^{\text{th}}$ order interaction across slots and *does not* satisfy interaction asymmetry for all equivalent generators, then $\boldsymbol{f}$ is not irreducible.

Since $\boldsymbol{f}$ has at most $0^{\text{th}}$ order interaction across slots and *does not* satisfy interaction asymmetry for all equivalent generators, this implies that there exists a matrix $\boldsymbol{A} \in \mathbb{R}^{|B_k \times B_k|}$ and a partition of $B_k$, into $A, B$ ($A \cup B = B_k, A \cap B = \emptyset$) such that within the function $\bar{\boldsymbol{f}}$ defined as:

$$\forall \boldsymbol{z} \in \mathcal{Z}, \qquad \bar{\boldsymbol{f}}\left(\boldsymbol{A}_1 \boldsymbol{z}_{B_1}, \ldots, \boldsymbol{A}_K \boldsymbol{z}_{B_K}\right) = \boldsymbol{f}(\boldsymbol{z}_{B_1}, \ldots, \boldsymbol{z}_{B_K}). \tag{E.4}$$

where $\boldsymbol{A}_i$ such that $i \neq k$ is the identity matrix, the latents $\bar{\boldsymbol{z}}_A, \bar{\boldsymbol{z}}_B$ have no interaction. This implies that under $\bar{\boldsymbol{f}}$, $I_A(\bar{\boldsymbol{z}})$ does not intersect with $I_B(\bar{\boldsymbol{z}})$. Further, because $\bar{\boldsymbol{f}}$ is invertible, we know that $D\bar{\boldsymbol{f}}_{B_k}(\bar{\boldsymbol{z}})$ is full column rank. Coupling these two properties, we conclude that $\text{rank}(D\bar{\boldsymbol{f}}_{B_k}(\bar{\boldsymbol{z}})) = \text{rank}(D\bar{\boldsymbol{f}}_A(\bar{\boldsymbol{z}})) + \text{rank}(D\bar{\boldsymbol{f}}_B(\bar{\boldsymbol{z}}))$. Furthermore, the Jacobians $D\bar{\boldsymbol{f}}(\bar{\boldsymbol{z}})$ and $D\boldsymbol{f}(\boldsymbol{z})$ will be related by an invertible linear map by construction. Thus, $D\bar{\boldsymbol{f}}_S(\bar{\boldsymbol{z}})$ and $D\boldsymbol{f}_S(\boldsymbol{z})$ have equal rank for any subset $S \subseteq [d_z]$. Therefore, we conclude that $\text{rank}(D\boldsymbol{f}_{B_k}(\boldsymbol{z})) = \text{rank}(D\boldsymbol{f}_A(\boldsymbol{z})) + \text{rank}(D\boldsymbol{f}_B(\boldsymbol{z}))$. Because $A$ and $B$ form a partition of $B_k$ we conclude that $\boldsymbol{f}$ is not irreducible.

We now prove the reverse direction if $\boldsymbol{f}$ has at most $0^{\text{th}}$ order interaction across slots and satisfies interaction asymmetry for all equivalent generators then $\boldsymbol{f}$ satisfies compositionality and irreducibility. As noted before, the definitions of compositionality and at most $0^{\text{th}}$ order interaction across slots are precisely equivalent. Thus, we only need to show that if $\boldsymbol{f}$ has at most $0^{\text{th}}$ order interaction across slots and satisfies interaction asymmetry then this implies $\boldsymbol{f}$ satisfies irreducibility. To show this, will prove the following contraposition: that if $\boldsymbol{f}$ does not satisfy irreducibility, then it *does not* satisfy interaction asymmetry for all equivalent generators with $n = 0$.

Since $\boldsymbol{f}$ is not irreducible, we know that there exist a $\boldsymbol{z}$, a slot $k \in [K]$, and a partition of $B_k$ into $A, B$ such that $\text{rank}(D\boldsymbol{f}_{B_k}(\boldsymbol{z})) = \text{rank}(D\boldsymbol{f}_A(\boldsymbol{z})) + \text{rank}(D\boldsymbol{f}_B(\boldsymbol{z}))$. Because $D\boldsymbol{f}_{B_k}(\boldsymbol{z})$ is full column rank this implies that $\text{rank}(D\boldsymbol{f}_A(\boldsymbol{z})) = |A|$ and $\text{rank}(D\boldsymbol{f}_B(\boldsymbol{z})) = |B|$. Now take two matrices $\boldsymbol{M}_{S_1} \in \mathbb{R}^{d_x \times |A|}$ and $\boldsymbol{M}_{S_2} \in \mathbb{R}^{d_x \times |B|}$ such that the column space of $\boldsymbol{M}_{S_1}$ is the same as $D\boldsymbol{f}_A(\boldsymbol{z})$ and the columns space of $\boldsymbol{M}_{S_2}$ is the same as $D\boldsymbol{f}_B(\boldsymbol{z})$. Now construct the following matrix $\boldsymbol{M} \in \mathbb{R}^{d_x \times |B_k|}$ as follows:

$$\boldsymbol{M} := [\boldsymbol{M}_{S_1}, \boldsymbol{M}_{S_2}] \tag{E.5}$$

Note that by construction this matrix has a block structure such that rows for $\boldsymbol{M}_{S_1}$ are never non-zero for the same rows as $\boldsymbol{M}_{S_2}$. Because $\boldsymbol{M}$ and $D\boldsymbol{f}_{B_k}(\boldsymbol{z})$ are both full column rank, then there exist a matrix $\boldsymbol{A}_k \in \mathbb{R}^{|B_k| \times |B_k|}$ such that:

$$\boldsymbol{M} := D\boldsymbol{f}_{B_k}(\boldsymbol{z})\boldsymbol{A}_k \tag{E.6}$$

Now define the function $\bar{\boldsymbol{f}}$ as follows:

$$\forall \boldsymbol{z} \in \mathcal{Z}, \qquad \bar{\boldsymbol{f}}\left(\boldsymbol{A}_1^{-1} \boldsymbol{z}_{B_1}, \ldots, \boldsymbol{A}_K^{-1} \boldsymbol{z}_{B_K}\right) = \boldsymbol{f}(\boldsymbol{z}_{B_1}, \ldots, \boldsymbol{z}_{B_K}). \tag{E.7}$$

such that $\boldsymbol{A}_i^{-1}$ is defined as above when $i = k$, and otherwise it is the identity matrix.

Writing the derivative of $D\bar{\boldsymbol{f}}_{B_k}(\bar{\boldsymbol{z}})$ in terms of $\boldsymbol{f}$ we get $D\boldsymbol{f}_{B_k}(\boldsymbol{z})\boldsymbol{A}_k = \boldsymbol{M}$. Because $\boldsymbol{M}$ has a block structure we conclude that there exist a partition of $B_k$ such that these latents have no interaction within $\bar{\boldsymbol{f}}$ at $\bar{\boldsymbol{z}}$. Because $\bar{\boldsymbol{f}}$ is equivalent to $\boldsymbol{f}$ we conclude that the function does not satisfy interaction asymmetry for $n = 0$. $\qquad\square$

## E.2 At Most $1^{\text{st}}$ Order Interaction Across Slots

We now prove that the assumptions in Lachapelle et al. (2023) are a special case of our assumptions for $n = 1$. To this end, we first restate their assumptions. The first assumption in Lachapelle et al. (2023) is that the generator $\boldsymbol{f}$ is *additive*:

**Definition E.4** (Additive decoder). A $C^2$ diffeomorphism $\boldsymbol{f} : \mathcal{Z} \to \mathcal{X}$ is said to be additive if:

$$\boldsymbol{f}(\boldsymbol{z}) = \sum_{k \in [K]} \boldsymbol{f}^k(\boldsymbol{z}), \quad \text{where } \boldsymbol{f}^k : \mathbb{R}^{|B_k|} \to \mathbb{R}^{d_x} \text{ for any } k \in [K] \text{ and } \boldsymbol{z} \in \mathcal{Z}. \tag{E.8}$$

**Definition E.5** (Sufficient Nonlinearity). Let $\boldsymbol{f} : \mathcal{Z} \to \mathcal{X}$ be a $C^2$ diffeomorphism. For all $k \in [K]$, let $B_{k\leq}^2 := B_k^2 \cap \{(i_1, i_2)|i_2 \leq i_1\}$. $\boldsymbol{f}$ is said to satisfy *sufficiently nonlinearity* if $\forall \boldsymbol{z} \in \mathcal{Z}$ the following matrix has full column-rank:

$$\boldsymbol{W}(\boldsymbol{z}) := \left[\left[D_i\boldsymbol{f}(\boldsymbol{z})\right]_{i \in B_k} \left[D_{i,i'}^2\boldsymbol{f}(\boldsymbol{z})\right]_{(i,i') \in B_{k\leq}^2}\right]_{k \in [K]} \tag{E.9}$$

We now state our result.

**Theorem E.6.** *Let $\boldsymbol{f} : \mathcal{Z} \to \mathcal{X}$ be a $C^2$ diffeomorphism. If $\boldsymbol{f}$ satisfies additivity (Def. E.4) and sufficient nonlinearity (Def. E.5) then $\boldsymbol{f}$ has at most $1^{st}$ order interactions across slots (Defn. 3.3), satisfies sufficient independence (Defn. A.9), and satisfies interaction asymmetry (Asm. 3.5) for all equivalent generators (Defn. 4.1) for $n = 1$.*

*Proof.* We note that $\boldsymbol{f}$ having at most first order interactions across slots is equivalent to having a block-diagonal Hessian for every observed component. Such functions were proven to be equivalent to additive functions in Lachapelle et al. (2023). Furthermore, sufficient independence is clearly implied by sufficient nonlinearity as if all columns of the matrix $\boldsymbol{W}(\boldsymbol{z})$ are linearly independent, then blocks $[D_i \boldsymbol{f}(\boldsymbol{z})]_{i \in B_k} \left[ D_{i,i'}^2 \boldsymbol{f}(\boldsymbol{z}) \right]_{(i,i') \in B_{k \le}^2}$ will have non-intersecting columns spaces for all $k \in [K]$ and will thus satisfy sufficient independence (Def. A.9. Consequently, the only thing we need to show is that sufficient nonlinearity (Def. E.5) implies interaction asymmetry (Assm. 3.5) for all equivalent generators (4.1).

Assume for a contradiction that sufficient nonlinearity (Def. E.5) did not imply interaction asymmetry (Assm. 3.5) for all equivalent generators (4.1) with $n = 1$. This would imply that there exists an equivalent generator to $\boldsymbol{f}$ denoted $\bar{\boldsymbol{f}}$ defined in terms of a slot-wise linear function $\boldsymbol{h}$:

$$\bar{\boldsymbol{f}} = \boldsymbol{f} \circ \boldsymbol{h} \tag{E.10}$$

such that $\bar{\boldsymbol{f}}$ has at most first order interaction within some slot $B_k$. In other words, leveraging Lemma A.21, there exist a $(j, j') \in B_k^2$ and a $\boldsymbol{z} \in \mathcal{Z}$ s.t.

$$0 = D_{j,j'}^2 \bar{\boldsymbol{f}}(\boldsymbol{z}) = \boldsymbol{W}^{\boldsymbol{f}}(\boldsymbol{h}(\boldsymbol{z})) \boldsymbol{m}^{\boldsymbol{h}}(\boldsymbol{z}, (j, j')), \tag{E.11}$$

Because $\boldsymbol{W}^{\boldsymbol{f}}(\boldsymbol{h}(\boldsymbol{z}))$ is assumed to be full rank by sufficient nonlinearity (Def. E.5), then in order for this equation to hold $\boldsymbol{m}^{\boldsymbol{h}}(\boldsymbol{z}, (j, j'))$ must be zero. Note, however, that by construction $\boldsymbol{h}$ is defined slot-wise such that $z_j, z_j'$ map to the same slot $\boldsymbol{h}_{B_k}$. By construction, if two $z_j, z_j'$ affect the same slot $\boldsymbol{h}_{B_k}$ then $\boldsymbol{m}^{\boldsymbol{h}}(\boldsymbol{z}, (j, j'))$, cannot be zero. Thus, we obtain a contradiction. $\square$

# F   TRANSFORMERS FOR INTERACTION REGULARIZATION

Each layer of a Transformer (Vaswani et al., 2017) consist of two main components: an MLP sub-layer and an attention mechanism. Notably, in the MLP sub-layer, MLPs are applied separately to each slot or pixel query and their outputs are then concatenated. Further, additional layer normalization operations (Ba, 2016) are typically used in Transformers but are also separately applied to each slot or pixel query. Thus, the only opportunity for interaction between slots in a Transformer occurs through the attention mechanism. Our focus in this work is on the cross-attention mechanism, opposed to the alternative self-attention. As noted in § 5, cross-attention takes the form:

$$\boldsymbol{K} = \boldsymbol{W}^K [\hat{\boldsymbol{z}}_{B_k}]_{k \in [K]}, \qquad \boldsymbol{V} = \boldsymbol{W}^V [\hat{\boldsymbol{z}}_{B_k}]_{k \in [K]}, \qquad \boldsymbol{Q} = \boldsymbol{W}^Q [\boldsymbol{o}_d]_{d \in [d_x]}, \tag{F.1}$$

$$\boldsymbol{A}_{d,k} = \frac{\exp\left(\boldsymbol{Q}_{:,d}^\top \boldsymbol{K}_{:,k}\right)}{\sum_{l \in [K]} \exp\left(\boldsymbol{Q}_{:,d}^\top \boldsymbol{K}_{:,l}\right)}, \qquad \bar{\boldsymbol{x}}_d = \boldsymbol{A}_{d,:} \boldsymbol{V}^\top, \qquad \hat{x}_d = \psi(\bar{\boldsymbol{x}}_d). \tag{F.2}$$

where $\boldsymbol{K}_{:,k}, \boldsymbol{V}_{:,k} \in \mathbb{R}^{d_q}$, $\boldsymbol{W}^K, \boldsymbol{W}^V \in \mathbb{R}^{d_q \times |B_k|}$ for query dimension $d_q$. Further, $\boldsymbol{o}_d \in \mathbb{R}^{d_o}$, $\boldsymbol{Q}_{:,l} \in \mathbb{R}^{d_q}$, $\boldsymbol{W}^Q \in \mathbb{R}^{d_q \times d_o}$, where $d_o$ is the dimension of a pixel coordinate vector, and $\psi : \mathbb{R}^{d_q} \to \mathbb{R}$.

**Additional Details.** In Eq. (F.2), we do not include the scaling factor $d_q^{-\frac{1}{2}}$ for $\boldsymbol{A}_{d,k}$, that is typically used as it does not affect our arguments below. We do, however, include it in our experiments. Further, when $\boldsymbol{x}$ is an RGB image, $\hat{x}_d$ will not be a scalar but will instead be a vector in $\mathbb{R}^3$ since each pixel has 3 color channels. Additionally, in our experiments, multi-head attention is used. In this case, slot keys and values and pixel queries are partitioned into $h$ sub-vectors. Eqs. (F.1) and (F.2) are then applied separately to each resulting sub-latent, and the resulting outputs are concatenated. When using multiple layers of cross-attention, as we do in our experiments, $\psi$ is only applied at the last layer and vectors $\boldsymbol{o}_l$ for a subsequent layer are defined as the vectors $\bar{\boldsymbol{x}}_d$ from the prior layer. Eqs. (F.1) and (F.2) is then repeated. We discuss how these additional caveats are dealt with empirically when implementing $\mathcal{L}_{\text{interact}}$ below in Appx. F.2, however, they do not affect our formal argument regarding regularizing interactions in Appx. F.1.

## F.1  JACOBIAN OF CROSS-ATTENTION MECHANISM

Our goal is to show that if $\boldsymbol{A}_{d,k}$ in equation is 0, then partial derivative of Eqs. (F.1) and (F.2) w.r.t slot $\hat{\boldsymbol{z}}_{B_k}$, i.e, $\frac{\partial \hat{x}_d}{\partial \hat{z}_{B_k}}$ will also be zero. This would then imply that if $\boldsymbol{A}_{d,:}$ is non-zero for at most one slot $k$ for every $d \in [d_x]$, and every $\hat{z} \in \hat{\mathcal{Z}}_{\text{supp}}$, then slots do not interact in the sense of Defn. 3.2, since all such derivative products $\frac{\partial \hat{x}_d}{\partial \hat{z}_{B_k}} \frac{\partial \hat{x}_d}{\partial \hat{z}_{B_l}}$ for $l \neq k$ are zero. To this end, we are interested in computing the derivative:

$$\frac{\partial \hat{x}_d}{\partial (\hat{\boldsymbol{z}}_{B_m})_r} = \partial_i \psi(\bar{\boldsymbol{x}}) \frac{\partial (\bar{\boldsymbol{x}}_d)_i}{\partial (\hat{\boldsymbol{z}}_{B_m})_r} \tag{F.3}$$

where we here and from now on use the convention that we sum over every index that appears only on one side. To evaluate this we decompose the terms

$$(\bar{\boldsymbol{x}}_d)_i = \boldsymbol{A}_{d,k} \boldsymbol{V}_{i,k} = \boldsymbol{A}_{d,k} \boldsymbol{W}_{i,j}^V (\hat{\boldsymbol{z}}_{B_k})_j. \tag{F.4}$$

We set $\boldsymbol{M}_{d,:} = \boldsymbol{o}_d^\top (\boldsymbol{W}^Q)^\top \boldsymbol{W}^K$ so that

$$\boldsymbol{Q}_{:,d}^\top \boldsymbol{K}_{:,k} = \boldsymbol{M}_{d,i} (\hat{\boldsymbol{z}}_{B_k})_i. \tag{F.5}$$

This implies that

$$\frac{\partial}{\partial (\hat{\boldsymbol{z}}_{B_m})_i} \exp(\boldsymbol{Q}_{:,d}^\top \boldsymbol{K}_{:,k}) = \boldsymbol{M}_{d,i} \delta_{km} \exp(\boldsymbol{Q}_{:,d}^\top \boldsymbol{K}_{:,k}) \tag{F.6}$$

where $\delta$ is the Kronecker-Delta (and here no summation over $k$ or $d$ is done). This implies using the product rule and the chain rule that

$$\frac{\partial \boldsymbol{A}_{d,k}}{\partial (\hat{\boldsymbol{z}}_{B_m})_i} = \boldsymbol{M}_{d,i} \delta_{k,m} \boldsymbol{A}_{d,k} - \boldsymbol{M}_{d,i} \boldsymbol{A}_{d,k} \boldsymbol{A}_{d,m}. \tag{F.7}$$

Plugging this together we get

$$\begin{aligned}
\frac{\partial \hat{x}_d}{\partial (\hat{\boldsymbol{z}}_{B_m})_r} &= \partial_i \psi(\bar{\boldsymbol{x}}) \frac{\partial (\bar{\boldsymbol{x}}_d)_i}{\partial (\hat{\boldsymbol{z}}_{B_m})_r} \\
&= \boldsymbol{A}_{d,m} \boldsymbol{W}_{i,r}^V \partial_i \psi(\bar{\boldsymbol{x}}) + \partial_i \psi(\bar{\boldsymbol{x}}) \boldsymbol{W}_{i,j}^V (\hat{\boldsymbol{z}}_{B_k})_j \frac{\partial \boldsymbol{A}_{d,k}}{\partial (\hat{\boldsymbol{z}}_{B_m})_r} \\
&= \boldsymbol{A}_{d,m} \boldsymbol{W}_{i,r}^V \partial_i \psi(\bar{\boldsymbol{x}}) + \partial_i \psi(\bar{\boldsymbol{x}}) \boldsymbol{W}_{i,j}^V (\hat{\boldsymbol{z}}_{B_k})_j (\boldsymbol{M}_{d,r} \delta_{k,m} \boldsymbol{A}_{d,k} - \boldsymbol{M}_{d,r} \boldsymbol{A}_{d,k} \boldsymbol{A}_{d,m}) \\
&= \boldsymbol{A}_{d,m} \partial_i \psi(\bar{\boldsymbol{x}}) (\boldsymbol{W}_{i,r}^V + \boldsymbol{W}_{i,j}^V (\hat{\boldsymbol{z}}_{B_m})_j \boldsymbol{M}_{d,r}) - \partial_i \psi(\bar{\boldsymbol{x}}) \boldsymbol{W}_{i,j}^V (\hat{\boldsymbol{z}}_{B_k})_j \boldsymbol{M}_{d,r} \boldsymbol{A}_{d,k} \boldsymbol{A}_{d,m}
\end{aligned} \tag{F.8}$$

From this, we can see that if $\boldsymbol{A}_{d,m} = 0$, then the partial derivative $\frac{\partial \hat{x}_d}{\partial \hat{z}_{B_m}}$, will indeed be zero as $\boldsymbol{A}_{d,m}$ scales both terms in the last line of Eq. (F.8).

## F.2  INTERACTION REGULARIZER

Based on Appx. F.1, we propose to regularize the interaction in a Transformer by minimizing the sum of all pairwise products $\boldsymbol{A}_{l,j} \boldsymbol{A}_{l,k}$, where $j \neq k$. More specifically, we minimize the following loss:

$$\mathcal{L}_{\text{interact}} := \mathbb{E} \sum_{l \in [d_x]} \sum_{j \in [K]} \sum_{k=j+1}^{K} \boldsymbol{A}_{l,j}(\hat{\boldsymbol{z}}) \boldsymbol{A}_{l,k}(\hat{\boldsymbol{z}}) \tag{F.9}$$

where $\boldsymbol{A}_{l,k}(\hat{\boldsymbol{z}})$ is used to indicate the input dependence of attention weights on latents $\hat{\boldsymbol{z}}$. $\mathcal{L}_{\text{interact}}$ is a non-negative quantity which will be zero if and only if a matrix has at most one non-zero for each row (Brady et al., 2023).

Code to compute $\mathcal{L}_{\text{interact}}$ for a batch of attention matrices can be seen in Fig. 4. We note that when using multiple attention heads, we first sum the attention matrices over all heads to ensure consistent pixel assignments across different heads.

When using multiple layers, we also sum the attention matrices over each layer, for the same reason. $\mathcal{L}_{\text{interact}}$ is then computed on the resulting attention matrix.

**Regularizing Higher Order Interactions.** We note that while we motivate $\mathcal{L}_{\text{interact}}$ as a regularizer for $1^{\text{st}}$ order interactions, we do not explicitly address regularizing for higher order interactions, i.e., for $n \geq 2$. We conjecture there

```python
def L_interact(attn):
    batch_size, num_slots, num_pixels = attn.shape
    interaction = 0
    for i in range(num_slots):
        for j in range(i, num_slots - 1):
            interaction += attn[:, i] * attn[:, j+1]
    return interaction.mean()
```

Figure 4: PyTorch code to compute $\mathcal{L}_{\text{interact}}$.

is a relationship between regularizing $\mathcal{L}_{\text{interact}}$ and higher order interactions but that it is less direct than the $1^{\text{st}}$ order case. We leave it to future work to explore these connections further, as well as alternative, computationally efficient regularizers which can more directly penalize higher order interactions.

**Computational Efficiency.** We note that regularizing with $\mathcal{L}_{\text{interact}}$ adds minimal additional computational overhead since attention weights are already computed at each forward pass through the model, and, moreover can be easily optimized using gradient descent. This is in contrast to Brady et al. (2023) which required computing the Jacobian of the decoder $\hat{f}$ at each forward pass and then optimizing it using gradient descent. This results in second-order optimization which is computationally intractable for high-dimensional data such as images (Brady et al., 2023).

## G    EXTENDED RELATED WORK

### G.1    THEORY

**Relationship Between Principle 3.1 and Other Principles.** The principle of interaction asymmetry, "parts of the same concept have more complex interactions, than parts of different concepts" (3.1), is intuitively similar to several prior principles explored for learning concepts. For example, the prior works of Baldwin et al. (2008); Reynolds et al. (2007); Schmidhuber (1990); Zacks et al. (2011) on disentangling events/sub-task (e.g., "making coffee", "driving to work"), Greff et al. (2015); Hyvarinen and Perkio (2006) on disentangling objects in an image, and Schmidhuber (1992) are all essentially based on the principle that parts of same concept are *more mutually predictable* than parts of different concepts. Similarly, Hochreiter and Schmidhuber (1999); Jiang et al. (2022) implicitly use the idea that parts of same concept are *more compressible* than different concepts. Research on networks, use the idea that nodes from the same "community" interact more strongly than nodes from different communities (Fortunato and Hric, 2016), which also resembles ideas from clustering that points from the same cluster have higher mutual information than from different clusters (Kraskov et al., 2005). This network-based framework was applied by Schapiro et al. (2013) as a model for grouping temporal events. Lastly, Greff et al. (2020) propose that objects do not interact much with their surroundings but internally have a strong structure. While these different ideas are intuitively similar to interaction asymmetry, they take on different formalizations. Moreover, these principles are generally used as high-level heuristics for designing a learning algorithm, and their theoretical implications for disentanglement and compositional generalization are not explored.

**Connection with Information Bottleneck Principle.** Another notable principle for learning representations is the Information Bottleneck principle (Alemi et al., 2016; Tishby et al., 2000) which has also been applied in the context of learning disentangled representations (Meo et al., 2024). In the context of disentanglement, this principle suggest learning a representation which tries to balance a trade-off between minimizing the mutual information between a latent vector $z$ and an observation $x$, and ensuring that $z$ contains sufficient information to predict, i.e., reconstruct $x$. From a theoretical standpoint, the Information Bottleneck principle differs from the principle of interaction asymmetry as defined in Asm. 3.5. Specifically, Asm. 3.5 is an assumption on the *generator* $f$ and does not place assumptions on the latent distribution $p_z$. Consequently, our theory describes a setting in which disentanglement can be achieved without explicitly enforcing any additional properties on $p_z$. We note, however, that despite this key difference, our theory does yield insights which resemble the Information Bottleneck principle. Specifically, as noted in § 5, our theory suggest that if a model uses an inferred latent dimension $d_{\hat{z}}$ greater than the ground-truth dimension $d_z$, then it should aim to encode $x$ using the minimal necessary latent dimension, i.e., the mutual information

between $x$ and unnecessary latent dimensions should be minimized, while ensuring that $x$ can be reconstructed from $z$.

**On the Relationship Between Disentanglement and Compositional Generalization.** A key premise motivating our theoretical study of compositional generalization is that, from a theoretical perspective, disentanglement does not directly imply compositional generalization. Specifically, this would require that equality between $f$ and $\hat{f}$ on $\mathcal{Z}_{\text{supp}}$ (disentanglement Defn. 2.1) implies that these functions were also equal on all of $\mathcal{Z}$ (compositional generalization Defn. 2.2). As noted by Lachapelle et al. (2023); Wiedemer et al. (2024a), this will not be true for arbitrary functions and necessitates restrictions on the form of $f$, $\hat{f}$ on all of $\mathcal{Z}$. While several works have provided empirical corroboration of this theoretical statement for concepts of objects (Wiedemer et al., 2024a) and object attributes (Montero et al., 2021; 2022b; Schott et al., 2022), prior works in disentanglement have suggested that disentanglement can in some cases enable compositional generalization (Esmaeili et al., 2018; Higgins et al., 2017; Mahon et al., 2023). We hypothesize that the compositional generalization abilities observed in the latter works are a consequence of only leaving a small number of novel combinations out of the training set, such that compositional generalization becomes much easier compared to the more restricted training domains explored in (Montero et al., 2021; Wiedemer et al., 2024a). With this being said, it is possible that through hidden inductive biases in a model, disentanglement can directly lead to compositional generalization, which would not be at odds with our theoretical observation.

**Polynomial Decoders.** As noted in § 4.2, Asm. 3.5 implies that the cross-partial derivatives of the generator $f$ consisting of components from different slots will be finite-degree polynomials. This partially resembles the polynomial constraints on $f$ in Ahuja et al. (2023) for disentanglement. Importantly, however, Ahuja et al. (2023) assume that *all* cross-partial derivatives of $f$ are polynomial such that the entire function $f$ is a finite-degree polynomial. In contrast, Asm. 3.5 constrains the form of cross-partial derivatives *across* slots to be polynomial, but *does not* constrain the form of cross-partial derivatives *within* the same slot. In other words, Asm. 3.5 only constrains the interactions across slots, while Ahuja et al. (2023) constrains all possible interactions. This is an important distinction since the former gives rise to much more flexible generators than the latter (see Eq. (4.2)).

## G.2 METHOD AND EXPERIMENTS

**VAE Losses in Object-Centric Models.** Prior work in Wang et al. (2023) also apply a VAE loss to an unsupervised object-centric learning setting. However, while we minimize $\mathcal{L}_{\text{KL}}$ directly on inferred slots in $\hat{z}$ given by our Transformer encoder, Wang et al. (2023) minimize $\mathcal{L}_{\text{KL}}$ on an intermediate representation which is then further processed to yield $\hat{z}$. Furthermore, the focus of Wang et al. (2023) is on scene generation an not penalizing the capacity of $\hat{z}$. Additionally, Kori et al. (2024) explore a loss for object-centric learning resembling a VAE loss, though their aim is to enforce a certain probabilistic structure on $\hat{z}$ implied by their theoretical disentanglement result, opposed to penalize latent capacity.

**Inductive Bias Through Explicit Supervision.** Recently, many works have shown remarkable empirical success in disentangling (Kirillov et al., 2023; Ravi et al., 2024) and composing (Brooks et al., 2023; Ramesh et al., 2021; 2022; Ruiz et al., 2023; Saharia et al., 2022) visual concepts in images on web-scale data. These works achieve this through explicit supervision via segmentation masks or natural language descriptions of each concept, opposed to constraints on the generative process in Eq. (2.1). Notably, however, many species in human's evolutionary lineage disentangle and compose concepts in sensory data *without* using explicit supervision like natural language (Behrens et al., 2018; LeCun, 2022; Summerfield, 2022; Tolman, 1948). This suggest the existence of a self-supervised coding mechanism for disentanglement and compositional generalization that is still lacking in current machine learning models. The present work aims to make theoretical and empirical progress towards such a mechanism.

**Relation Between a Transformer Regularized with $\mathcal{L}_{\text{interact}}$ and Prior Works.** Goyal et al. (2021) proposed RIMs which is a Transformer-style architecture aimed at enforcing a "modular" structure. Contrary to our work, Goyal et al. (2021) do not regularize for modularity, but posit that it may emerge from "competition" induced by an attention mechanism. Similarly, Lamb et al. (2021) propose an alternative Transformer architecture aimed at enforcing modularity, which also tries to enforce competition using a mechanism similar to Goyal et al. (2021). More recently, Vani et al. (2024) propose a novel Transformer component which is aimed at yielding disentanglement by processing a

Transformer embedding into different slots using separate attention heads for each slot. While these works are similar to ours in that they aim to learn disentangled representations of concepts using a Transformer-style architecture, they are based on architectural changes to a Transformer, whereas we use a standard cross-attention Transformer decoder and regularize it explicitly towards having a "modular" structure using $\mathcal{L}_{\text{interact}}$.

## H  EXTENDED DISCUSSION

### H.1  THEORETICAL ASSUMPTIONS

**Non-Homogeneous Interactions.** One potential limitation of our formulation of interaction asymmetry (Asm. 3.5) is that the order of interaction, $n$, across slots, must be the same for all latent vectors $z \in \mathcal{Z}$ and for any two slots $z_{B_i}, z_{B_j}$. This assumption will potentially be violated in practice. For example, for concepts of visual objects, it is likely that within each image, only a few objects, i.e., slots, interact at a time (e.g., see Fig. 1), such that different slots will have different orders of interaction within $z$. We conjecture that our theory can be extended to handle such non-homogeneous interactions, however, we leave this for future work. Furthermore, we note that despite this potential mismatch between theory and practice, our method still achieved robust object-disentanglement on data in which the order of interaction appears to be non-homogeneous, e.g., CLEVR6.

**Requirements on the Observed Dimension.** We note that an implication of sufficient independence for $n > 0$ (where $n$ is the order of interaction across slots) is that the observed dimension $d_x$ must be greater than the latent dimension $d_z$. Moreover, the required $d_x$ will scale as a function of the number of latent slots $K$, the slot dimensions $|B_k|$, and the order of interaction across slots $(n)$. For example, for functions with at most 1$^{\text{st}}$ order interactions across slots, ensuring that the rank condition in sufficient independence (Defn. A.9) is met requires that $d_x \geq \sum_{k \in [K]} \frac{|B_k|(|B_k|+1)}{2} + d_z$. Furthermore, for functions with at most 2$^{\text{nd}}$ order interactions, satisfying this condition (Defn. 4.2) requires $d_x \geq \sum_{k \in [K]} \frac{|B_k|(|B_k|+1)(|B_k|+2)}{6} + \frac{d_z(d_z+1)}{2} + d_z$. We note that we are interested in modelling high-dimensional sensory data, such as images, in which the observed dimension $d_x$ will be much greater than the latent dimension $d_z$. Thus, for practical cases of interest, we expect these requirements on $d_x$ to be met.

**Concepts Potentially not Captured by Interaction Asymmetry.** For certain concepts, it is not obvious if interaction asymmetry will always hold. For example, consider object attributes such as the $x$-$y$-position of an object, which can be modelled by one-dimensional slots. For such concepts, the interaction within a slot, i.e., the interaction of each latent component w.r.t. itself, should, intuitively, be a simple function. It is thus not obvious if the interaction within each slot will necessarily be more complex than interactions across slots, such that $f$ may not satisfy interaction asymmetry (Asm. 3.5). Additionally, it is not immediately clear how interaction asymmetry can be applied to more abstract concepts which are not directly grounded in sensory data such as the concept of "democracy" or the concept of a "function" in mathematics.

**Restrictiveness of the Aligned-Connected Assumption.** Our theoretical results in § 4 leverage the assumption that the latent space $\mathcal{Z}_{\text{supp}}$ is aligned-connected Defn. A.16. To assess whether the aligned-connectedness assumption is realistic, we believe it is helpful to look at concrete mathematical examples of supports that satisfy it. For example, the whole space $\mathbb{R}^{d_z}$ is aligned-connected. More generally, any convex set is aligned-connected. This include the hypercube $[0,1]^{d_z}$, any closed ball, and much more. Some aligned-connected sets are not convex. For example, the "L-shaped" set $[0,2]^2 \setminus [1,2]^2$ is aligned-connected but not convex. This last example is useful to model concrete settings where some combinations of latent factors are not observed at training time. This corresponds to the running example of Lachapelle et al. (2023) consisting of two balls moving up and down where the configurations where both balls appear in the top half of the image are never observed.

### H.2  METHOD AND EXPERIMENTS

**Self-Attention in Transformer Decoders.** Our Transformer decoder in § 5 resembles the models from Jaegle et al. (2022); Sajjadi et al. (2022a;b) which only rely on a cross-attention mechanism. However, other works in object-centric learning leverage Transformer decoders which also include a self-attention mechanism between queries at each layer (Seitzer et al., 2023; Singh et al., 2022a).

When reconstructing individual pixels, e.g., on Sprites and CLEVR6 in § 6, applying self-attention between queries will not scale to high-dimensional images since it requires computing $n \times n$ attention weights, where $n$ is the number of pixels. However, when reconstructing *image patches*, as was done in our experiments on CLEVRTex, using self-attention is scalable since we have a significantly smaller number of queries, e.g., $16 \times 16$ on CLEVRTex. While we found that we could achieve strong disentanglement using only a cross-attention mechanism on CLEVRTex, it is possible that using self-attention could be advantageous when reconstructing image patches in even more complex settings. For such models, however, it is not immediately obvious how to regularize the decoder to match our theory since adding self-attention between pixels will introduce additional interactions between slots. We leave it for future work to investigate if our current training objective will still yield robust object-disentanglement for such a model and, furthermore, if such a model can be regularized to be in line with our theory.

**Trade-Offs with Slot Attention.** On Sprites and CLEVR6 (§ 6) as well as CLEVRTex , we found that our regularized Transformer autoencoder achieved superior disentanglement, based on our metrics, to an unregularized variant with a Slot Attention encoder. Despite this, we emphasize that our goal is not to propose our method as superior to Slot Attention-based methods. Instead, we highlight that both methods offer different trade-offs. For example, training with our proposed loss (Eq. (5.3)) enables using a general Transformer encoder, thus potentially allowing our model to be applied more generally at scale compared to encoders with more explicit object-centric priors such as Slot Attention. This, however, comes at the cost of training with regularizers which require hyperparameter selection. While our experiments did not require extensive hyperparameter tuning, it is possible that certain datasets will exhibit increased sensitivity to these hyperparameters. Additionally, our interaction regularizer is based on decoders which only use a cross-attention mechanism. While this architecture yielded strong disentanglement in our experiments, Slot Attention encoders have been shown to enable disentanglement using more expressive decoders which also use self-attention (Seitzer et al., 2023; Singh et al., 2022a).

**Latent Prediction-Based Disentanglement Metrics** One potential issue with our Jacobian-based disentanglement metrics is that they may fail to measure whether multiple slots actually encode the same object. Specifically, if two slots affect the same object in pixel space, this could be due to both slots encoding the object in latent space, or it could be due to slot interactions modelled by the decoder. Definitively resolving this potential ambiguity would require measuring the information encoded in each slot directly in latent space. Along this line, prior works have considered latent prediction metrics in which the R2 score is computed from the predictions of a model fit between each inferred slot and the best matching ground-truth slot (Dittadi et al., 2022; Jiang et al., 2023; Locatello et al., 2020b). While these metrics indicate if an inferred slot contains all information for a given object, they are insufficient for resolving the possible ambiguity of our current metrics. This is because these metrics do not indicate if an inferred slot contains information about *more than one* object. This issue with latent prediction metrics was pointed out by Brady et al. (2023) who aimed to address it by measuring the R2 score from an additional predictor fit to the second-best matching ground-truth slot. We found this metric to yield inconsistent results on CLEVR6, which we hypothesize was due to issues when determining the second-best matching ground-truth slot. This lead us to focus on decoder-based metrics which are more straightforward to compute. We leave it for future work to formulate a latent prediction metric which overcomes the aforementioned issues of prior works.

**On Hyperparameter Selection.** One potential limitation of using our regularized loss (Eq. (5.3)) in terms of scalability is that hyperparameter selection is required. In our experiments on Sprites and CLEVR6 (§ 6), extensive hyperparameter tuning was not required. Furthermore, we found the values of $\alpha = 0.05$, $\beta = 0.05$ to work robustly across both datasets, though the reconstruction loss was weighted by a factor of 5 on Sprites and 1 on CLEVR6. This indicates some level of robustness of these hyperparameters across datasets which contain varying complexity of interaction. On CLEVRTex (Appx. I), we found that these exact hyperparameter values did not transfer directly and a small amount of tuning was required to arrive at our values (.1 for all terms in the loss). We hypothesize that this is because, in our current implementation of our loss, the magnitude of the reconstruction loss scales with the dimension of the data. To this end, because the data dimension increased significantly on CLEVRTex (256 encoded image patches, each with 768 dimensions), the contribution of the reconstruction term to the loss needed to be slightly diminished. With all this being said, it is possible that more complex dataset could require more extensive hyperparameter tuning, however, we leave this for future work to investigate.

**Applying our Method to Other Types of Concepts.** One important direction for future work is to apply our method to data consisting of different types of concepts such as object-attributes or temporal events. For object-attributes, our same empirical framework can be applied, but with the additional caveat that the Transformer is permutation invariant, while object-attributes do not posses the same permutation invariance as objects. To this end, methods such as adding a positional encoding to each slot, must be used to address this. Additionally, as noted by Gopalakrishnan et al. (2023); Kipf et al. (2019), the problem of disentangling temporal events in image sequences can also be modelled naturally using a slot-based framework. In this case, the "tokens" that a slot encoder, e.g., a Transformer or Slot Attention, operates on are not pixels processed by a CNN, as in our current model. Instead, they would correspond to individual images in the sequence which are each mapped into representation "tokens". These tokens can then be mapped into slots by, e.g., a Transformer, and then decoded back to output space, where the queries for the Transformer decoder also would not correspond to individual pixels but instead to images in the temporal sequence.

**Limitations of $\mathcal{L}_{\text{disent}}$.** One potential issue with $\mathcal{L}_{\text{rec}}$ is that for real-world data, reconstructing every pixel in an image exactly, may not be necessary and could lead to overly prioritizing tasks irrelevant information in $\hat{z}$ such as the background (Seitzer et al., 2023). It would thus be interesting to see if our theory and method could be extended to a self-supervised setting, as in Seitzer et al. (2023), in which exact invertibility is not strictly necessary. Regarding $\mathcal{L}_{\text{KL}}$, we first note that in addition to a model having inferred latent dimensionality $d_{\hat{z}}$ equal to ground-truth dimension $d_z$, our theory also requires that the inferred slot dimensions equals the ground-truth slot dimensions. While $\mathcal{L}_{\text{KL}}$ explicitly regularizes for the former, it does not directly regularize for the latter. More specifically, $\mathcal{L}_{\text{KL}}$ could, in principle, penalize latent capacity by putting information from all, e.g., objects, in one slot (assuming the slot size is large enough), opposed to distributing this information over components from different slots. Despite this, we found that this failure mode did not occur in our experiments. Another potential issue with $\mathcal{L}_{\text{KL}}$ is that it aims to enforce statistically independent latents which could lead to sub-optimal solutions if the ground-truth latents exhibit strong statistical dependencies. Lastly, regarding $\mathcal{L}_{\text{interact}}$, a shortcoming of this regularizer is that, while it directly regularizes $1^{\text{st}}$ order interactions (Appx. F.1), its connection to regularizing higher order interactions is not as direct. Future work should thus aim to investigate this point further both theoretically and empirically.

### H.3 ENFORCING THEORETICAL CRITERIA OUT-OF-DOMAIN.

As noted in § 5, enforcing (i) invertibility and (ii) at most $n^{\text{th}}$ order interactions across slots on $\hat{f}$, out-of-domain, i.e., globally on all of $\mathcal{Z}$, poses distinct practical challenges. We now discuss this in detail. To this end, we first discuss enforcing (ii) globally on $\mathcal{Z}$.

**Restricting Interactions Globally.** The easiest way to enforce that $\hat{f}$ has at most $n^{\text{th}}$ order interactions across slots on $\mathcal{Z}$ is to directly parameterize $\hat{f}$ to match the form of such functions for some $n$ (Eq. (4.2)). This is, for example, how at most $1^{\text{st}}$ order interactions were enforced in Lachapelle et al. (2023), i.e., by defining $\hat{f}$ to be an additive function (Defn. E.4) on all of $\mathcal{Z}$. We found for higher order interactions, parameterizing $\hat{f}$ directly to match the form of Eq. (4.2) leads to training difficulties on toy data. Moreover, even if we could easily train such a model, this explicit form would pose an overly restrictive inductive bias when scaling to more realistic data. This motivated us to consider how to regularize for (ii) opposed to enforce it explicitly. The issue with this approach is that we only regularize the derivatives of $\hat{f}$ in-domain on $\hat{\mathcal{Z}}_{\text{supp}}$. Yet, enforcing structure on the derivatives of $\hat{f}$ on $\hat{\mathcal{Z}}_{\text{supp}}$ does not imply that same structure will be enforced on all of $\mathcal{Z}$. As noted in § 4.2, however, by knowing the behavior of the derivatives of $\hat{f}$ on $\hat{\mathcal{Z}}_{\text{supp}}$, we can infer their behavior everywhere on $\mathcal{Z}$. Thus, in principle, it should be possible to propagate the correct derivative structure learned by $\hat{f}$ locally on $\hat{\mathcal{Z}}_{\text{supp}}$, to all of $\mathcal{Z}$. Practically, however, it is not obvious how this can be done in an effective manner. Thus, properly addressing this challenge would require further methodological and empirical contributions, which are not within the scope of the present work.

**Enforcing Invertibility Globally.** Additionally, even if $\hat{f}$ satisfies (ii) globally, we still must enforce (i) invertibility, globally. As noted in § 5, it is not feasible to define $\hat{f}$ such that it is an invertible function from $\mathcal{Z}$ to $\mathcal{X}$ by construction. This necessitated parameterizing the inverse of $\hat{f}$ with *an encoder $\hat{g}$* which was trained to invert the *decoder $\hat{f}$* via a reconstruction loss. Assuming that a decoder $\hat{f}$ satisfies (ii) globally, and is invertible on $\hat{\mathcal{Z}}_{\text{supp}}$, it is possible to show that $\hat{f}$ will be

invertible on all of $\mathcal{Z}$ and thus generalize compositionally. The issue, however, is that our encoder $\hat{g}$ is only trained to invert $\hat{f}$ on $\hat{\mathcal{Z}}_{\text{supp}}$ but not on unseen data from the rest of of $\mathcal{Z}$. Consequently, even if $\hat{f}$ generalizes compositionally, an encoder $\hat{g}$ will not invert $\hat{f}$ out-of-domain, and can thus yield an arbitrary representations $\hat{z}$ on such data. This "encoder-decoder inconsistency" was pointed out by Wiedemer et al. (2024a), which studied compositional generalization for decoders with at most $0^{\text{th}}$ and $1^{\text{st}}$ order interactions. They proposed a loss which addresses this problem by first generating out-of-domain samples using $\hat{f}$, and then training the encoder $\hat{g}$ to invert $\hat{f}$ on this "imagined" data. The implementation of this loss in Wiedemer et al. (2024a), deemed *compositional consistency*, was shown to be ineffective for images consisting of more than 2 objects, however (Wiedemer et al., 2024a). Consequently, scaling this loss, or exploring alternative losses for encoder-decoder consistency, remain open research question that require a deeper investigation to properly address.

For these reasons, the empirical aspects of this work focus on enforcing (i) and (ii) in-domain to achieve disentanglement on $\mathcal{Z}_{\text{supp}}$ (Thm. 4.3). As highlighted above, however, our theory elucidates the core problems that need to be solved empirically to also achieve compositional generalization, thus giving a clear direction for future work.

# I    EXPERIMENTS ON CLEVRTEX

In this section, we conduct additional experiments on the CLEVRTex dataset (Karazija et al., 2021).This dataset constitutes a significant step up in complexity from CLEVR6 and has been shown to be highly challenging for existing object-centric models (Biza et al., 2023; Karazija et al., 2021). We outline our experimental setup and results below.

## I.1    EXPERIMENTAL SETUP

**Data.** Each image in CLEVRTex consist of between 3 and 10 objects with rich textures, set against complex backgrounds (see Fig. 7 for example images). The dataset consists of 50,000 images. We use 40,000 images for training and 5,000 for validation and testing, respectively.

**Models.** We train 4 models on this data. The first model is our regularized Transformer autoencoder from § 5, for which we weight each term in the loss in Eq. (5.3) by a hyperparameter value of .1. The second model is an unregularized Transformer autoencoder, and the third model is an unregularized autoencoder which uses a Slot Attention encoder with both a Transformer and slot-wise MLP decoder. We train all models using the same setup as in § 6, however, instead of reconstructing the original images, we reconstruct a representation of each image given by a Vision Transformer (ViT) (Dosovitskiy et al., 2021), which is pretrained using the DINO method (Caron et al., 2021). This approach, deemed DINOSAUR (Seitzer et al., 2023), was shown to help object-centric models scale to datasets with increased visual complexity. We thus replace the CNN backbone used in our experiments on Sprites and CLEVR6 with a pretrained ViT which operates on $8 \times 8$ patches of the original images. These patches are mapped to features which are then processed by either a Transformer or Slot Attention encoder. For all models, we use 11 slots with a slot dimension of 64.

**Training and Evaluation Details** We train all models across 3 random seeds using batches of 32. In all cases, we use the Adam optimizer (Kingma and Ba, 2015) with a learning rate of $5 \times 10^{-4}$ which we warm-up for the first 10,000 training iterations and then decay by a factor of 10 throughout training. We also warm-up the value of $\alpha$ for the first 25,000 training iterations. We report the J-ARI and JIS for each model after training for 300,000 iterations. To compute these scores, we bilinearly interpolate our normalized Jacobian maps to match the shape of the original image, since we are reconstructing image patches. When computing J-ARI and JIS for the slot-wise MLP decoder, we rely on the alpha-mask of the decoder opposed to its Jacobian due to computational issues when computing the Jacobian for this model.

## I.2    RESULTS

We report our results in Tab. 2. As we can see, similar to on Sprites and CLEVR6, our regularized Transformer achieves strong object-disentanglement, outperforming both unregularized baseline methods in terms of J-ARI. Our model also a achieve superior JIS compared to all baseline models with the exception being the slot-wise MLP decoder. This is not unexpected, however, as this decoder explicitly constrains interactions in the same way as the Spatial Broadcast Decdoer used

Table 2: **Empirical Results.** We show the mean $\pm$ std. dev. for J-ARI and JIS (in %) over 3 seeds for different choices of encoders and weights of the loss terms in Eq. (5.3) on CLEVRTex.

| Encoder | Decoder | Loss | J-ARI ($\uparrow$) | JIS ($\uparrow$) |
|---------|---------|------|--------|-------|
| Transformer | Transformer | $\alpha = 0, \beta = 0$ | $81.4 \pm 3.7$ | $50.9 \pm 2.8$ |
| Slot Attention | Transformer | $\alpha = 0, \beta = 0$ | $94.2 \pm 0.2$ | $54.4 \pm 0.3$ |
| Slot Attention | Slot-wise MLP | $\alpha = 0, \beta = 0$ | $92.8 \pm 0.2$ | $\mathbf{84.3 \pm 0.4}$ |
| Transformer | Transformer | $\alpha = 0.1, \beta = 0.1$ **(Ours)** | $\mathbf{95.9 \pm 0.06}$ | $65.4 \pm 0.6$ |

in other experiments. We also visually corroborate these results by plotting normalized slot-wise Jacobians for each model which can be seen in Fig. 7.

## J EXPERIMENTAL DETAILS

### J.1 DATA, MODEL, AND TRAINING DETAILS

**Data.** The Sprites dataset used in § 6 was generated using the Spriteworld renderer (Watters et al., 2019a) and consist of 100,000 images of size $64 \times 64 \times 3$ each with between 2 and 4 objects. The CLEVR6 dataset (Johnson et al., 2017; Locatello et al., 2020b) consist of 53,483 images of size $128 \times 128 \times 3$ each with between 2 and 6 objects. For Sprites, we use 5,000 images for validation, 5,000 for testing, and the rest for training, while for CLEVR6, we use 2,000 images for validation and 2,000 for testing.

**Encoders.** All models use encoders which first process images using the same CNN of Locatello et al. (2020b). When using a Transformer encoder, these CNN features are fed to a 5 layer Transformer which uses both self- and cross-attention with 4 attention heads. When using a Slot Attention encoder, we use 3 Slot Attention iterations, and use the improved implicit differentiation proposed in Chang et al. (2022). Both the Transformer and Slot Attention encoders use learned query vectors opposed to randomly sample queries, which was shown by Biza et al. (2023) to yield improved performance for Slot Attention. On Sprites, all models use 5 slots, each with 32 dimensions, while on CLEVR6, all models use 7 slots, each with 64 dimensions. When using a VAE loss, this slot dimension doubles since we must model the mean and variance of each latent dimension.

**Decoders.** When using a Spatial Broadcast decoder (Watters et al., 2019b), we use the same architecture as (Locatello et al., 2020b) across all experiments, using a channel dimension of 32 for both datasets. When using a Transformer decoder, we first upscale slots to 516 dimensions by processing them separately using a 2 layer MLP, with a hidden dimension of 2064. We then apply a 2 layer cross-attention Transformer to these features which uses 12 attention heads. To obtain the vectors $o_l$ in Eq. (5.1), we apply a 2D positional encoding to each pixel coordinate. This vector is then mapped by a 2 layer MLP with a hidden dimension of 360 to yield $o_l$, which has dimension 180. The function $\psi$ in Eq. (5.2) is implemented by a 3 layer MLP with a hidden dimension of 180, which outputs a 3 dimensional pixel $\hat{x}_l$ for each pixel $l$. We additionally note that this architecture does not rely on auto-regressive masking as in Singh et al. (2022a).

**Training Details.** We train all models on Spriteworld across 3 random seeds using batches of 64 for 500,000 iterations. For CLEVR6, we use batches of 32 and train for 400,000 iterations. In all cases, we use the Adam optimizer (Kingma and Ba, 2015) with a learning rate of $5 \times 10^{-4}$ which we warm-up for the first 30,000 training iterations and then decay by a factor of 10 throughout training. When training with $\beta \mathcal{L}_{KL}$ and $\alpha \mathcal{L}_{interact}$, we use hyperparameter weights of 0.05, which we found to work well across both datasets. We found much larger values could lead to more training instability and, in some cases, insufficient optimization of $\mathcal{L}_{rec}$, while smaller values often did not lead to sufficient optimization of the regularizers. We warm-up the value of $\alpha$ for the first 30,000 training iterations. Additionally, when training with $\alpha$ or $\beta$, we drop the value of the learning rate after 30,000 training iterations to $1 \times 10^{-4}$, which improved training stability. Lastly, on Sprites, we weight $\mathcal{L}_{rec}$ by a factor of 5, when training with $\alpha$ or $\beta$.

### J.2 METRICS AND EVALUATION

**Computing ARI with Attention Scores.** To compute the Adjusted Rand Index (ARI), each pixel must first be assigned to a unique model slot. To this end, prior works typically choose the slot with the largest attention score from either Slot Attention or the alpha mask of a Spatial Broadcast decoder (Locatello et al., 2020b; Seitzer et al., 2023). This approach can be problematic since the attention scores used are model-dependent, making a direct comparison of ARI across models challenging. Further, the relationship between attention scores and the pixels encoded in a model slot is somewhat indirect. As noted in § 6, we consider an alternative and compute the ARI using the Jacobian of a decoder (J-ARI). Specifically, we assign a pixel $l$ to the slot with the largest $L_1$ norm for the slot-wise Jacobian $D_{B_k}\hat{f}_l(\hat{z})$. This can be done for any autoencoder and provides a more principled metric for object disentanglement since a decoder's Jacobian directly describes the pixels each slot encodes (assuming $\hat{f}, \hat{g}$ invert each other).

**Evaluation.** We select models for testing which had the highest average values for J-ARI and JIS (each of which take values from 0 to 1) on the validation set. These models were then evaluated on the test set yielding the scores reported in Tab. 1.

### J.3 ADDITIONAL FIGURES

In this subsection, we include 3 additional experimental figures. In Fig. 5, we compare the value of $\mathcal{L}_{\text{interact}}$ throughout training for a model with a Transformer encoder and decoder, trained using a our regularized loss Eq. (5.3), the VAE loss and a standard autoencoder loss on both Sprites and CLEVR6. We plot values over 3 random seeds; the shaded regions in the plots indicate one standard deviation from the mean. We find on Sprites (**A**) and CLEVR6 (**B**) that the VAE loss achieves much lower $\mathcal{L}_{\text{interact}}$ than the unregularized model. This provides a possible explanation for the solid object disentanglement often achieved by the VAE loss in Tab. 1. We also observe, however, that using $\alpha > 0$ leads to much lower values for $\mathcal{L}_{\text{interact}}$ compared to the implicit regularization from the VAE loss.

In Fig. 6, we compare slot-wise Jacobians for our model versus baseline models across both Sprites (**A**) and CLEVR6 (**B**). To create these plots, we normalize the partial derivatives across slots such that they only take values between 0 and 1. The colors associated with partial derivative values can be interpreted using the color bar at the bottom of (**A**). We only compute partial derivatives on the foreground pixels and set the derivatives of background pixels w.r.t each slot to 0. We see that when regularizing interactions via our model, slots rarely affect the same pixels (i.e., interact) unnecessarily, while for unregularized models, multiple slots often affect the same pixels even when no interactions should occur, e.g., for images in Sprites (**A**).

In Fig. 7, we compare slot-wise Jacobians on CLEVRTex as was done in Fig. 6 and also observe here that the regularized Transformer achieves cleaner object decompositions compared to baseline models.

In Fig. 8, we compare decoder attention maps w.r.t. each slot for our model versus baseline models from § 6, which also use a Transformer decoder. These maps, which indicate the slots each pixel attends to, are plotted for both Sprites (**A**) and CLEVR6 (**B**). We compute these values by taking the mean attention weight over decoder layers. Similar to Fig. 6, we see that, in our model, pixels rarely unnecessarily attend to multiple slots. On the other hand, for unregularized models, pixels often attend to multiple slots in cases where no interactions between slots should occur.

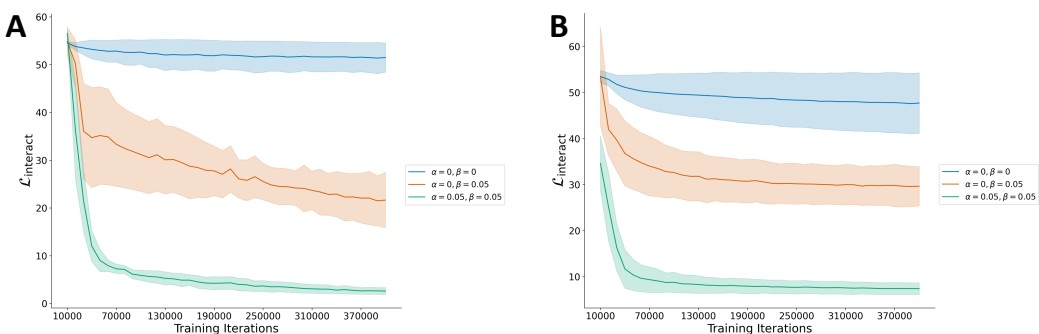

Figure 5: **Analysis of $\mathcal{L}_{\text{interact}}$ when using a VAE loss.** We plot $\mathcal{L}_{\text{interact}}$ for the first 400,000 training iterations for a Transformer autoencoder trained without regularization ($\alpha = 0, \beta = 0$), with a VAE loss which does not explicitly optimize $\mathcal{L}_{\text{interact}}$ ($\alpha = 0, \beta = 0.05$), and with the loss in Eq. (5.3) which regularizes both losses ($\alpha = 0.05, \beta = 0.05$).

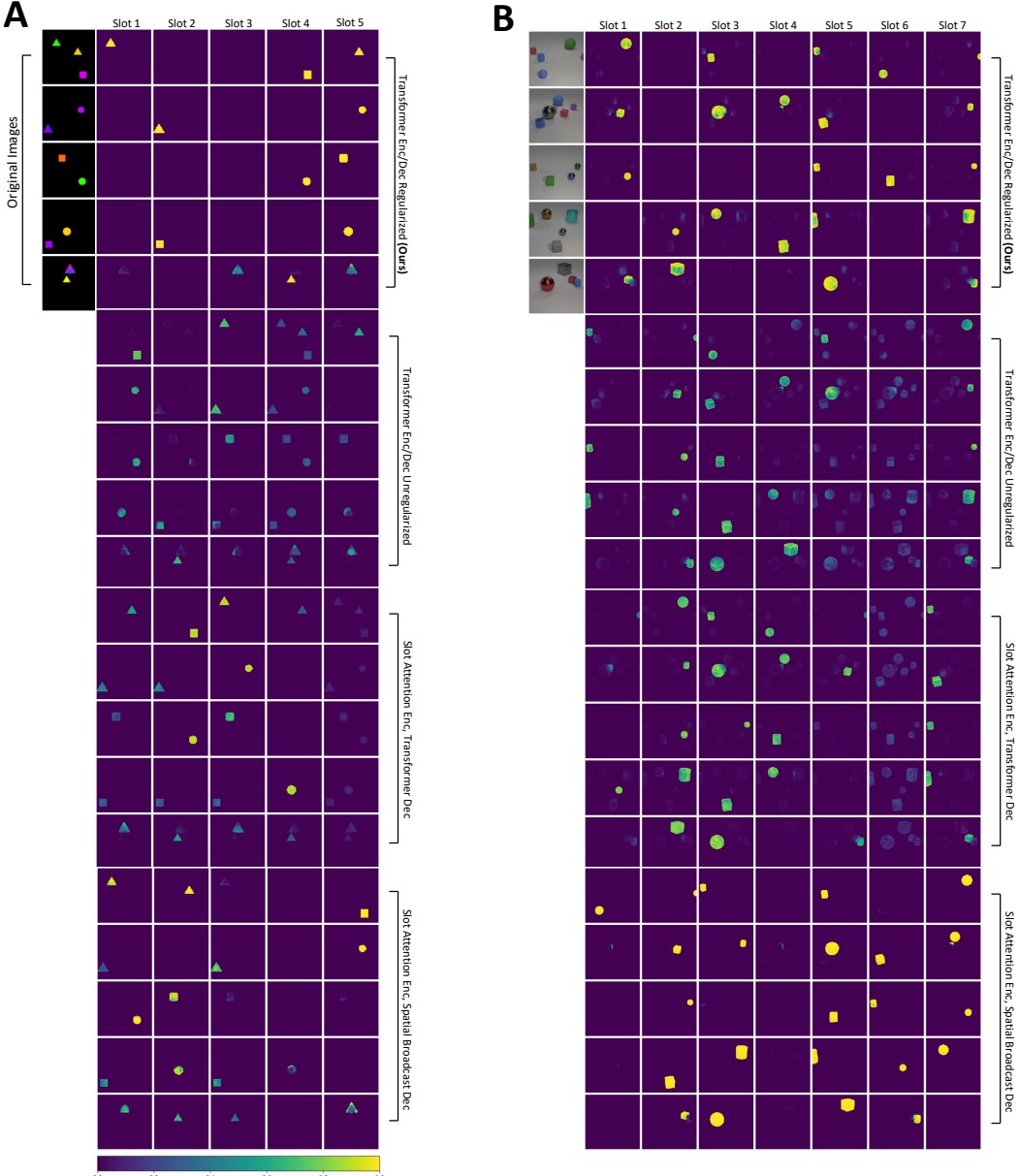

Figure 6: **Normalized Slot-wise Jacobians.** We plot the Jacobians w.r.t. each slot (columns) for 5 random test images (rows) from (**A**) Sprites and (**B**) CLEVR6 for our regularized Transformer model and the baseline models used in our experiments in § 6.

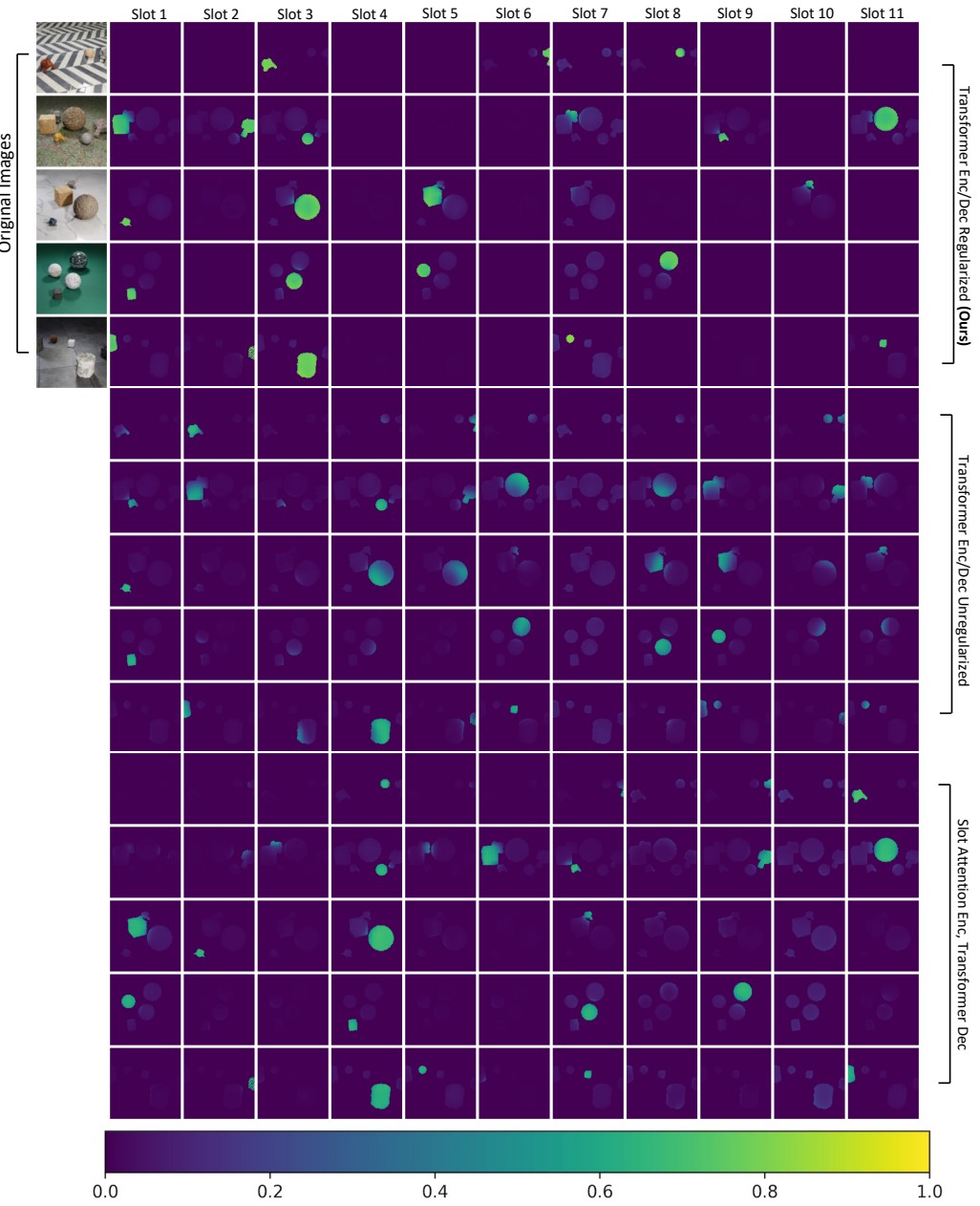

Figure 7: **Normalized Slot-wise Jacobians (CLEVRTex).** We plot the Jacobians w.r.t. each slot (columns) for 5 random test images (rows) from CLEVRTex for our regularized Transformer model and the baseline models used in our experiments in Appx. I.

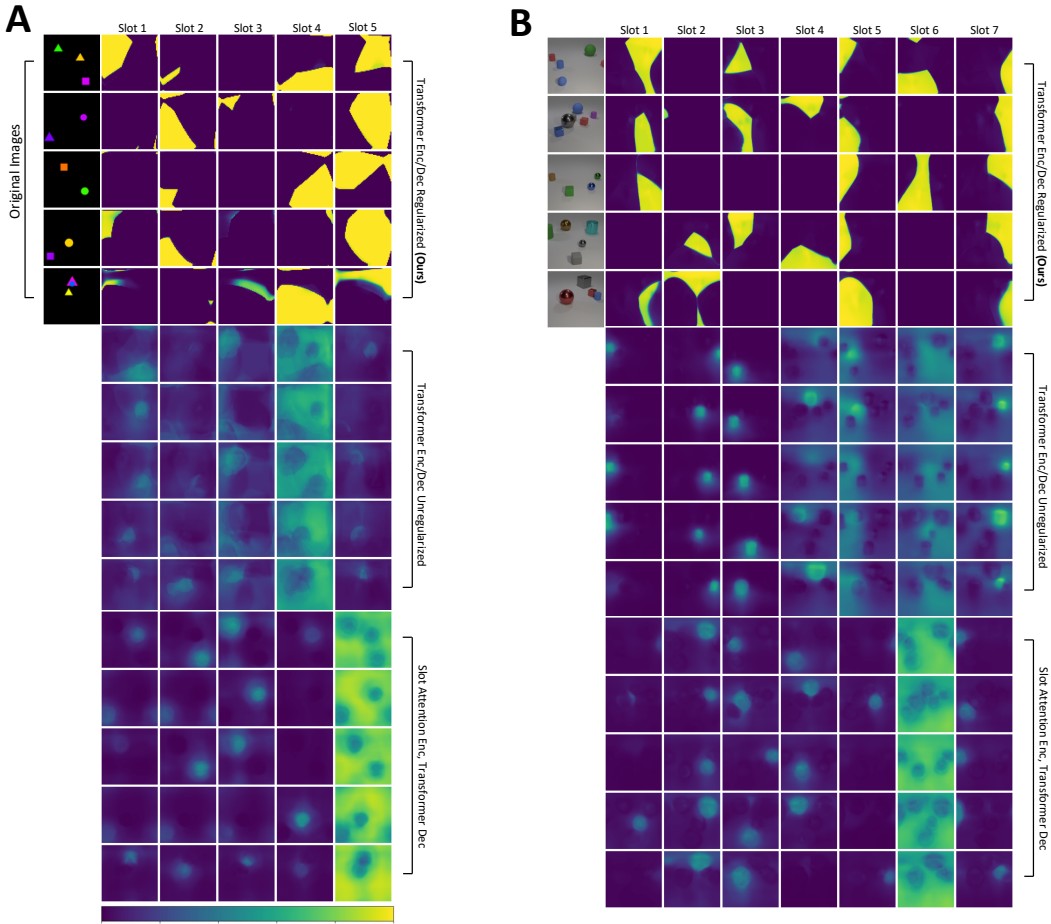

Figure 8: **Slot-wise Transformer Decoder Attention Maps.** We plot decoder attention maps w.r.t. each slot (columns) for 5 random test images (rows) from (**A**) Sprites and (**B**) CLEVR6 for our regularized Transformer decoder and the baseline models in § 6 which also use a Transformer decoder.

