# OpenReview forum: "Interaction Asymmetry: A General Principle for Learning Composable Abstractions"
_ICLR.cc/2025/Conference — ICLR 2025 Poster_

### Official Review · Reviewer_A7GP · 2024-10-16

**Soundness:** 4
**Presentation:** 3
**Contribution:** 3
**Rating:** 8
**Confidence:** 4

**Summary:**

This paper introduces the interaction asymmetry principle as a way to measure disentanglement and compositional generalisation, based on decomposing the latent space into blocks, and requiring non-zero partial derivatives between elements of the same block, but not between those of different blocks. The authors prove, in the appendix, that functions meeting this condition are disentangled, and discuss how this framework generalises some existing characterisations of disentanglement. Then they describe how this idea can be implemented as an additional loss term in a vision transformer VAE, and present results quantifying the degree of disentanglement using two metrics they devise, and compared to a slot attention decoder and a spatial broadcast decoder.

**Strengths:**

The problem of measuring disentanglement and compositional generalisation is interesting.

The intro and high-level background is generally well-motivated.

The general principle of interaction asymmetry is, to my knowledge, creative in that it is unlike previous approaches to this problem. It is convincing and put to good use.

Eq 4.2 is a nice expression of how a function can, in some sense be decomposed into a disentangled and entangled portion.

**Weaknesses:**

The discussion on the relationship between disentanglement and compositional generalisation could be improved by engaging with further existing work. While Montero et al (2020, 2022) claim to find no relationship between disentanglement and compositional generalisation, Higgins et al. (2016) and Esmaili et al. (2019) claim that their disentangled methods also enabled compositional generalisation, and Mahon (2024) claims to find an empirical correlation between disentanglement and compositional generalisation across multiple models and datasets. In fact, the position of Montero seems somewhat incongruent with your argument that they both share an underlying mechanism of interaction asymmetry.

I am slightly confused by some terminology: does ‘latent’ mean latent dimension or latent vector? e.g. on 197 "group of latents from the same or different slots" sounds like the former, but on line 409, it's in bold, suggesting a vector; does 'block' mean the same thing as 'slot'?

In Section 4.2, I don’t follow why $f$ and $\hat{f} \circ h$ being ‘sufficiently predictable’, in having all partial derivatives be fixed-degree polynomials, means that they are both equal on $Z_{cpe}$? It seems like there are three notions here, the interaction asymmetry condition, 'sufficiently predictable' and compositional generalisation.

The description of ARI on lines 478-9 is not the standard one as used e.g. in clustering (Vinh et al; 2009). Please state what the measure is exactly.

Comparing the different ablation settings in Table 1, it looks like having $\beta > 0$ makes a bigger difference than $\alpha > 0$, did you try a range of values for $\beta$? it could be that if you have the correct value of $\beta$, then $alpha>0$ makes no difference. It’s a bit odd to just report the value of these scaling parameters as > 0, it would be more insightful to report the exact values used.

Some information from Appendix I should be in the main paper, at least the number of dimensions in each slot.

**Refs**

Montero et al. (2020) "On the Role of Disentanglement in Generalisation".

Montero et al. (2022) "Lost in Latent Space: Examining failures of disentangled models at combinatorial generalisation"

Higgins et al. (2016) "beta-vae: Learning basic visual concepts with a constrained variational framework."

Esmaeli et al. (2019) "Structured Disentangled Representations"

Mahon et al. (2024) "Correcting Flaws in Common Disentanglement Metrics"

Vinh et al (2009). "Information Theoretic Measures for Clustering Comparison: Is a Correction for Chance Necessary?"

**Questions:**

What does your work imply about the debate in the literature as to whether disentanglement and compositional generalisation are related?

How does  ‘sufficiently predictable’ imply compositional generalisation?

What are the exact values of $\alpha$ and $\beta$ in Table 1?

Also, a general point: This formulation of the interaction principle is binary, either the required derivatives are zero everywhere or they are not. Disentanglement, on the other hand, is normally quantified to a matter of degree, giving a particular model on a particular dataset a score in [0,1]. How easy would it be to extend your formulation to provide a score in [0,1], rather than {0,1}?

---

> ### Author Response · Authors · 2024-11-22
>
> We thank the reviewer for their time and valuable feedback. We address each of your comments below:
>
> **Comment**:  “The discussion on the relationship between disentanglement and compositional generalisation could be improved by engaging with further existing work.”
>
> **Response**:
>
> Thank you for your comment. We have included a new paragraph, "On the Relationship Between Disentanglement and Compositional Generalization" discussing this point further in Appendix G.1. To summarize, our observation that disentanglement does not imply compositional generalization is theoretical in nature and is the same point made in [1, 2]. It amounts to saying that if a model inverts the ground-truth generator on the training set, i.e., achieves disentanglement, this does not imply mathematically that it also inverts the generator out-of-domain, i.e., achieves compositional generalization. Compositional generalization is only theoretically possible if a model’s decoder is enforced to have a certain structure on the full latent space. From an empirical perspective, however, it is possible that hidden inductive biases in a model could lead to a decoder having such a structure on the full latent space, which could potentially explain the results referenced by the reviewer, and would not be at odds with our theoretical observation
>
> **Comment**:  “The position of Montero seems somewhat incongruent with your argument that they both share an underlying mechanism of interaction asymmetry..”
>
> Thank you for raising this nuanced point! The reviewer is correct in pointing out that our work states that the same properties on a model which enable disentanglement are also sufficient for enabling compositional generalization. We emphasize, however, that this is different from stating that disentanglement implies compositional generalization. Specifically, in Theorem 4.3, we show that if certain properties on a models decoder are satisfied **in-domain** on $\mathcal{Z}_{supp}$, then a model will be disentangled, while Theorem 4.4 states that if these same properties are enforced **out-of-domain** on all of $\mathcal{Z}$, then a model will generalize compositionally. Importantly, however, enforcing these properties in-domain does not imply that they will also be enforced out-of-domain. To enforce the properties out-of-domain requires additional methodological contributions, which we discuss in detail in Section H.3.
>
>
> **Comment**:  “I am slightly confused by some terminology: does ‘latent’ mean latent dimension or latent vector?”
>
> **Response**:
>
> Thank you for raising this point! The reviewer is correct in noting that we overloaded the word latent in these context. We have updated the manuscript such that line 197 now reads “groups of latent components” while line 409 has been changed to “latent vector”.
>
> **Comment**: “It seems like there are three notions here, the interaction asymmetry condition, 'sufficiently predictable' and compositional generalisation.”
>
> **Response**:
>
> Thank you for your comment. We apologize for the confusion and hope we can clarify this point. We first note that our use of “sufficiently predictable” is not meant to introduce a new mathematical notion. In Section 2, line 152, we discuss that for a model to generalize compositionally, one must be able to determine how a decoder $\boldsymbol{f}$ should behave out-of-domain on $\mathcal{Z}$, by only knowing its behavior in-domain on $\mathcal{Z}_{supp}$. In this sense, the behavior of the function must be “predictable” from its behavior in-domain. Hence, "predictable" is used in an intuitive fashion to describe this problem. To this end, Figure 2, aims to communicate that by restricting interactions via interaction asymmetry, the behavior of $\boldsymbol{f}$ out-of-domain on $\mathcal{Z}$ can be determined its behavior in-domain, because the cross partial derivatives are simple functions, i.e. polynomials. Thus the function is “predictable” in this sense. We hope this provides some clarity on this point, and based on your comment we have removed the phrase “sufficiently” to avoid potentially viewing this phrase as a new mathematical notion.
>
> **Comment**: “The description of ARI on lines 478-9 is not the standard one as used”
>
> **Response**:
>
> In our experiments, we use the adjusted rand index from [3], which is a standard metric used in object-centric learning. We use the implementation from scikit-learn which can be found [here](https://scikit-learn.org/dev/modules/generated/sklearn.metrics.adjusted_rand_score.html). We hope this is sufficient to resolve the reviewers comment. Otherwise, we can add a more detailed description of the metric to our appendix.

---

> ### Author Response · Authors · 2024-11-22
>
> **Comment**: “ did you try a range of values for $\beta$” ,..  it would be more insightful to report the exact values used.”
>
> **Response**:
>
> Thank you for this question! In our experiments on both Sprites and CLEVR6, we used values of 0.05 for both $\alpha$ and $\beta$. This was stated in line 2515 of our initial manuscript, however, we agree with the reviewer that adding these values to Table 1 would enhance clarity, and have thus made this addition to Table 1. Regarding, how these values were chosen, we did indeed explore different values for both parameters and found the stated values to work robustly. We did not perform extensive hyperparameter sweeps over $\beta$, however, thus we cannot rule out if certain values can be chosen such that using $\alpha > 0$ makes no difference. We believe our results suggest, however, that there is benefit to training with $\alpha > 0$. Specifically, as discussed on line 528 of our initial manuscript, the success of just training with $\beta > 0$ seems to be due to this loss implicitly minimizing $\mathcal{L}_{\text{interact}}$. Despite this implicit regularization, however, one does still achieve much lower values for this loss when training with $\alpha > 0$. We have updated Fig 5 in Appendix J.3, to reflect this point. To this end, we do still expect to see improvements in disentanglement when training with $\alpha > 0$.
>
>
>
> **Comment**: Some information from Appendix I should be in the main paper, at least the number of dimensions in each slot.
>
> **Response**:
> Thank you for this feedback! Accordingly, we will include information such as the slot dimension for the camera ready version of this work.
>
>
> **Comment**: This formulation of the interaction principle is binary, … How easy would it be to extend your formulation to provide a score in [0,1], rather than {0,1}?
>
> **Response**:
>
> Thank you for your question! We think there may be some misunderstanding, which we hope to clarify. The reviewer is correct that whether an inference model achieves disentanglement can be quantified via a continuous value in [0,1]. However, this is measured as the disentanglement of a model **w.r.t. a given a ground-truth model**. This is different than asking whether or not a given ground-truth model satisfies the assumptions necessary for disentanglement. It is possible that a continuous score could be formulated to measure this, however, this will only be possible if one has access to the ground-truth generative process, which is generally not feasible. We note that this point is not unique to our work but will be true for any theoretical disentanglement result which formulates assumptions on the ground-truth generative process.
>
> [1] Lachapelle et al., 2024, Additive Decoders for Latent Variables Identification and Cartesian Product Extrapolation
>
> [2] Wiedmer et al., 2024, Provable Compositional Generalization for Object-Centric Learning
>
> [3] Hubert et al., 1985, Comparing partitions.

---

### Official Review · Reviewer_c6zh · 2024-10-31

**Soundness:** 3
**Presentation:** 4
**Contribution:** 3
**Rating:** 6
**Confidence:** 3

**Summary:**

This paper introduces a novel principle for learning composable representations, termed “interaction asymmetry.”, stating that interactions between distinct concepts are less complex than those within sub-parts of each concept. By formalizing these interactions as high-order derivatives of each concept to output images, the authors theoretically demonstrate that interaction asymmetry facilitates disentanglement and compositional generalization (though the latter is not empirically validated). Notably, their theoretical findings encompass previous approaches for achieving compositional generalization as specific cases (e.g., assumptions of at most 0th or 1st order interactions) and extend to accommodate more complex scenarios with higher-order interactions.To implement these theoretical insights, the authors propose regularizing cross-attention layers in standard transformer architectures. Empirical evaluations on two synthetic datasets show that the proposed method effectively captures distinct objects. The main contributions of the paper are as follows:
- The introduction of interaction asymmetry as a general principle for compositional representation learning, supported by a theoretical framework that links interaction asymmetry to disentanglement and compositional generalization.
- A practical method to enforce interaction asymmetry within transformers by regularizing cross-attention layers, with empirical results on synthetic datasets that demonstrate the method’s ability to capture distinct objects.

**Strengths:**

- The paper is clearly written and easy to follow, with a well-motivated presentation that effectively derives theorems and implementations.
- The theoretical foundation of the paper is solid and sound, offering meaningful insights that contribute to our understanding of compositional representation learning.
- It is interesting and valuable that the proposed approach unifies existing work while also extending to more complex scenarios, e.g., including high-order interactions between concepts, highlighting its versatility and potential impact.

**Weaknesses:**

- Although the theoretical foundation is solid, the experimental support is relatively limited. The empirical results are based solely on the simple Sprite and CLEVR6 datasets, raising questions about the generalizability of the findings to more complex scenarios.
- Theorem 4.3 states that achieving disentanglement requires ensuring at most $n$-th order interaction between slots.
While theoretically sound, implementing this in practice as minimizing interaction without knowing the ground truth value of $n$, could be a limitation. For instance, each dataset (such as Sprite or CLEVR6) may have interactions of different orders, implying that the regularizer's weight to control interaction strength should be carefully adjusted for each case. If the Sprite dataset exhibits 0th order interaction, a stronger regularizer may be necessary compared to that in CLEVR6. Additional interpretation or obervations from the authors on this aspect would make the empirical support for their theory more comprehensive.
- To implement Theorem 4.3 in a feasible way, the authors propose to minimize interactions between distinct slots through an attention regularizer in the transformer. However, it is unclear if this regularizer provides any advantage over the existing spatial exclusiveness inductive bias in slot attention encoders, which already minimizes interaction through slot-wise competition. Clarifying the unique benefits of the proposed regularizer compared to slot attention’s inherent mechanism would enhance the paper's contributions.
- Additionally, the authors claim that their generalized transformer structure with regularization is more flexible than slot attention-based methods (e.g., slot attention encoder + transformer decoder in L460–463). However, this claim is less convincing due to the following reasons: (1) To enforce invertibility, an additional KL term is required in the objective function, as shown in Table 1 (the value suggests that this term is crucial).
Therefore, the proposed approach requires two additional terms (KL term and the regularizer term) and it may introduces additional burden on finding proper hyper-parameters. It is hard to tell if these inductive biases are favorable than architectural bias in slot attention methods.
(2) If my understanding is correct, the transformer decoder lacks a self-attention layer, using only cross-attention to avoid slot interactions. This specific design differs from the conventional transformer architecture, potentially limiting scalability. In contrast, slot attention-based methods, they can resume off-the-shelf models such as off-the-shelf encoder + slot attention encoder + off-the-shelf decoder (as in DINOSAUR [1], SLATE [2], LSD [3]), which might be more flexible in real-world applications.
- Finally, the proposed J-ARI metric may be misleading. The Jacobian of the decoder identifies regions most affected by each slot, but due to slot interactions, it may fail to segment objects accurately. For example, in Figure 3(B), the metallic sphere reflects the blue cuboid, causing the normalized slot-wise Jacobian to activate for the blue cuboid in the reflection area. However, this does not imply that the slot for the blue cuboid genuinely contains information about the metallic sphere. To assess whether composable abstractions are effectively captured, a downstream task like property prediction, as in LSD, may be necessary.

[1] Seitzer et al., bridging the gap to real-world object- centric learning, in ICLR 23.

[2] Singh et al., Illiterate dall-e learns to compose, in ICLR 22.

[3] Jiang et al., Object-centric slot diffusion, in NeurIPS 23.

**Questions:**

- In L519–521, can the trade-off between interaction and model expressivity be effectively controlled through the parameter $\alpha$? For example, when increasing $\alpha$, does it lead to increased J-ARI but prevent reflections in mirrored objects in output images?
- As the Sprite dataset inherently exhibits 0th order interaction between objects, wouldn’t a spatial broadcast decoder (though the final alpha-blending layer does permit some interaction) serve as an architectural bias enforcing 0th order interaction? If so, should this setup represent the upper bound performance? Why does the proposed method outperform this approach?
- Although not a critical question, why did the authors choose to use learned queries instead of randomly sampled queries, which are typically used in Slot Attention?

---

> ### Author Response · Authors · 2024-11-22
>
> We thank the reviewer for their time, and appreciate their comments highlighting the value of our theoretical contribution. We are also grateful for the insightful feedback on our empirical contribution. These comments lead us to address several nuances, which we believe has improved the quality and clarity of our work.
>
> Before addressing your comments, we would first like to clarify our view on the contributions of this work. We view our theory as our core contribution. Such a theoretical foundation for both disentanglement and compositional generalization has thus far been lacking, and we view our main contribution as providing this foundation through a general principle, which also unifies and extends prior work.
>
> Our main objective empirically was to showcase the practical utility of this theory in informing the design of novel learning methods. To this end, our aim is not to present our method as necessarily superior to existing methods. Instead, we aim to highlight that insights from our theory lead to a new object-centric learning method which yields strong object disentanglement while potentially offering different advantages in terms of scalability compared to existing methods.
>
> We now address each of your comments below:
>
>
> **Comment**:  “the experimental support is relatively limited, …raising questions about the generalizability of the findings to more complex scenarios.”
>
> **Response**:
>
> We agree with the reviewer that adding additional experiments is important for understanding the generalizability of our method to more complex scenarios. We have thus conducted additional experiments on the CLEVRTex dataset which is a significant step up in complexity compared to CLEVR6. We found that our method is also able to achieve strong object disentanglement on this data, outperforming the baseline methods we tested against in terms of our metrics. We discuss these results in more detail in the general reply to all reviewers and have added a new section in the paper (Appendix I) presenting our experimental setup and results.
>
> **Comment**:  “minimizing interaction without knowing the ground truth value of “. “If the Sprite dataset exhibits 0th order interaction, a stronger regularizer may be necessary compared to that in CLEVR6.”
>
> **Response**:
>
> We agree that minimizing interactions without knowing n could potentially pose challenges empirically. In our experiments on Sprites and CLEVR6, however, we found the same hyperparamter values ($\alpha=0.05, \beta=0.05$) to work well across both datasets, though the weight on the reconstruction loss was changed from 5. to 1. on CLEVR6. We have now included a new discussion paragraph on hyperparameter selection in Section H.2.
>
>
> **Comment**: “Clarifying the unique benefits of the proposed regularizer compared to slot attention’s inherent mechanism would enhance the paper's contributions.”
>
> **Response**:
>
> We thank the reviewer for their comment. We would like to emphasize that **our aim is not to present our method as inherently superior to Slot Attention**, but instead we highlight that both methods offer different tradeoffs. For example, training with our loss enables using a general Transformer encoder, which potentially yields better scalability than encoders with more explicit object-centric priors such as Slot Attention. This comes at the cost of training with regularizers, however, which require hyperparameter selection. While our experiments did not require extensive hyperparameter tuning, it is possible that certain datasets will exhibit increased hyperparameter sensitivity. We have added a new discussion paragraph *“Trade-offs with Slot Attention”* in Section H.2, discussing these points along with additional details.
>
> We also note, however, that, while the mechanism in Slot Attention provides a useful inductive bias for minimizing interactions between slots, it does not explicitly regularize for interactions, as in our method. Furthermore, we find that on all datasets we consider, our method yields slightly stronger object disentanglement compared to a Slot Attention-based Transformer autoencoder. This can also be seen visually by inspecting Jacobian plots for each dataset, which we have now added in Figures 6 and 7 in Appendix J.3.

---

> ### Author Response · Authors · 2024-11-22
>
> **Comment**:  “the proposed approach requires two additional terms (KL term and the regularizer term) and it may introduces additional burden on finding proper hyper-parameters. It is hard to tell if these inductive biases are favorable than architectural bias in slot attention methods"
>
> **Response**:
>
> We agree with the reviewer that our method requires hyperparameter selection which could present computational overhead for certain datasets. We note, however, that in our experiments, we did not find extensive hyperparameter tuning to be necessary. We have added a discussion paragraph on this point in Section H.2 . Similar to in our previous response, however, we would like to emphasize that we do not view our method as being in competition to Slot Attention but instead again highlight that both methods offer different tradeoffs.
>
> **Comment**: If my understanding is correct, the transformer decoder lacks a self-attention layer, …, potentially limiting scalability.
>
> **Response**:
>
> Thank you for this insightful comment! The reviewer is correct that our Transformer decoder only uses cross-attention layers. To this end, we would first like to note that using self-attention in a decoder is not scalable if the task is to reconstruct individual pixels, since self-attention must be computed on all pixel queries. Consequently, methods which leverage self-attention in the decoder, rely on reconstructing *image patches* opposed to pixels. In our experiments reconstructing image patches on CLEVRTEx, we found that only using cross-attention layers was sufficient for strong object-disentanglement. However, it is possible that leveraging self-attention could be advantageous in even more complex settings. In this case, it is not immediately obvious how to adapt our attention regularizer since self-attention will introduce further interactions between slots. We have added a new discussion paragraph, *"Self-Attention in Transformer Decoders."*, discussing these points in more detail in Section H.2.
>
> **Comment**: “The proposed J-ARI metric may be misleading. To assess whether composable abstractions are effectively captured, a downstream task like property prediction, may be necessary.”
>
> **Response**:
>
> Thank you for this nuanced comment! The reviewer is correct in pointing out this potential issue with our metrics which we did not previously discuss. We have added a new discussion paragraph *“Latent Prediction-Based Disentanglement Metrics“* to Section H.2, discussing this point in detail. To briefly summarize, we agree with the reviewer that latent prediction metrics may be necessary to resolve the ambiguity of whether a given slot actually encodes multiple objects. With this being said, we argue that the metrics used in the works referenced by the reviewer, are insufficient for determining this. Specifically, these metrics indicate if an inferred slot contains all information about a given object, however, they do not indicate if an inferred slot contains information about **more than one** object. This issue was pointed out by [1] who aimed to address it by fitting an additional predictor between the second most informative ground-truth slot for a given inferred slot. In early experiments, we found the metric from [1] to yield inconsistent results on CLEVR6, which lead us to focus on decoder-based metrics. We leave it for future work to formulate a latent prediction metric which overcomes the aforementioned issues of prior works.
>
> **Comment**: “can the trade-off between interaction and model expressivity be effectively controlled through the parameter “
>
> **Response**:
>
> We conjecture that it is possible to have more fine grained control of this trade-off through $\alpha$. While we did not observe a case in our runs on CLEVR6 in which our method did not model interactions, we were also not explicitly targeting this behavior. We note that inducing this behaviour would likely require carefully warming up the value of $\alpha$ throughout training since if the parameter is chosen to be too large at the beginning of training, we observed that models are often pushed towards sub-optimal solutions.
>
>
> **Comment**: “wouldn’t a spatial broadcast decoder … represent the upper bound performance? Why does the proposed method outperform this approach?”
>
> **Response**:
>
> We hypothesize that the worse performance of Spatial Broadcast Decoders (SBDs) relative to our model on Sprites is due to the following reasons: Firstly, the alpha-blending layer in SBDs can indeed introduce interactions, such that these models do not give an upper bound on performance. Additionally, while occlusions are relatively rare on Sprites, they can occur, and are often more pronounced than on CLEVR6. Consequently, it is possible that the restrictions on interactions in SBDs can prevent the model from properly modelling this data, while a more flexible Transformer decoder can more easily model it.

---

> ### Author Response · Authors · 2024-11-22
>
> **Comment**: “why did the authors choose to use learned queries instead of randomly sampled queries,”
>
> **Response**:
> This decision was based on recent work in object-centric learning from [2], which showed performance gains when using learned queries in Slot Attention. We have now added an explanation for this choice and a citation to this work to the manuscript.
>
> [1] Brady et al., 2023, Provably Learning Object-Centric Representations
>
> [2] Biza et al., 2023, Invariant Slot Attention: Object Discovery with Slot-Centric Reference Frames

---

### Official Review · Reviewer_qqbq · 2024-11-01

**Soundness:** 4
**Presentation:** 3
**Contribution:** 3
**Rating:** 8
**Confidence:** 5

**Summary:**

The paper titled "Interaction Asymmetry: A General Principle for Learning Composable Abstractions" presents the principle of "interaction asymmetry" to explain why some latent structures enable better generalization and disentanglement in representation learning. The core idea is that parts of the same concept interact more complexly compared to parts of different concepts, and the authors formalize this using block diagonality in higher-order derivatives of a generator function.

The main contributions of the paper include:

Theory: A formal proof that the principle of interaction asymmetry enables both disentanglement and compositional generalization. The authors extend previous results to more general generator functions, generalizing beyond earlier specific cases.
Method: An autoencoder-based approach that uses a Transformer architecture with an attention-based regularizer to limit interactions across slots, following the interaction asymmetry principle.
Empirical Evaluation: Demonstration on synthetic datasets that the proposed model can achieve disentanglement comparable to other models while allowing for more flexible interactions among the latent representations.
The paper unifies prior work by presenting a more general theoretical framework and proposes a practical implementation that achieves strong results in object disentanglement tasks.

**Strengths:**

**Originality**:
The paper introduces the principle of **interaction asymmetry** to advance disentanglement and compositional generalization, providing a unified theoretical foundation that applies to broader conditions. The use of block diagonality in higher-order derivatives is creative and extends representation learning. The combination of theoretical insights with a practical Transformer-based VAE implementation is also notable.

**Quality**:
The theoretical contributions are well-developed, extending previous findings to more complex settings. The regularized Transformer-based VAE is supported by empirical evidence, showing comparable performance to state-of-the-art methods. The use of multiple datasets (Sprites and CLEVR6) allows for a thorough assessment of the model's capacity for disentanglement.

**Clarity**:
The paper is well-written and accessible. The authors clearly define theoretical constructs like **interaction asymmetry** and **nth order interaction**, using visual aids to illustrate complex concepts effectively. The logical structure and provided diagrams help convey the ideas intuitively.

**Significance**:
The principle of interaction asymmetry offers a general framework for **disentanglement** and **compositional generalization**, extending the applicability of models to more complex generative processes. The flexible and scalable Transformer-based model positions this work as a significant step toward learning useful representations for real-world tasks.

**Weaknesses:**

1. **Limited Empirical Evaluation**: The empirical evaluation, while thorough for the presented datasets, is somewhat limited in scope. The paper only evaluates on synthetic datasets (Sprites and CLEVR6), which may not fully demonstrate the model's potential for real-world applications. Including experiments on more challenging, real-world datasets would strengthen the evidence for the model's practical utility and scalability.

2. **Theoretical Assumptions**: The theoretical results rely on several assumptions (e.g., sufficient independence, specific conditions on the latent distribution). These assumptions may be restrictive in practical scenarios, and their implications are not fully explored in the context of real-world data. It would be beneficial to discuss how these assumptions could impact the applicability of the model and whether there are potential relaxations that could make the theory more practical.

3. **Connection with Information Bottleneck**: The paper does not explicitly connect its disentanglement framework with the **Information Bottleneck** principle, which has been discussed in several works as an effective approach to disentanglement. Incorporating a discussion on how interaction asymmetry relates to information bottleneck theory could strengthen the theoretical underpinnings and offer insights into how limiting mutual information might aid in disentangling the latent representations. Including references to relevant literature, such as Tishby et al. (2000), Alemi et al. (2017)  and Meo et al. (2024), could provide additional context and depth to the discussion.

https://arxiv.org/abs/physics/0004057
https://arxiv.org/abs/1612.00410
https://openreview.net/pdf?id=ptXo0epLQo

**Questions:**

1. Are there plans to extend the empirical evaluation to include more challenging, real-world datasets? This would help establish the practical utility of the model in diverse settings.

2. Given the theoretical assumptions made in the paper, could you discuss how they might be relaxed to make the model more applicable to real-world scenarios? Are there any potential modifications to the theory that could broaden its applicability?

3. How does the interaction asymmetry principle relate to the Information Bottleneck theory? A discussion on this connection could provide additional insights into the theoretical foundations of the proposed approach and highlight its relevance to existing literature on disentanglement.

---

> ### Author Response · Authors · 2024-11-22
>
> We thank the reviewer for their valuable feedback. We also appreciate the positive assessment of our theoretical and empirical contribution as well the comments on the overall significance of our contribution.
>
> We address each of your comments in detail below:
>
> **Comment**:  “The empirical evaluation, [...], is somewhat limited in scope.”
>
> **Response**:
>
> We agree with the reviewer that adding experiments on more complex data is important for understanding the practical utility and scalability of our model. We have thus conducted additional experiments on the CLEVRTex dataset which is a significant step up in complexity compared to CLEVR6. We found that our method is also able to achieve strong object disentanglement on this data, outperforming the baseline methods we tested against in terms of our metrics. We discuss these results in more detail in the general reply to all reviewers and have added a new section in the paper (Appendix I) presenting our experimental setup and results.
>
>
> **Comment**: “It would be beneficial to discuss how these assumptions could impact the applicability of the model and whether there are potential relaxations that could make the theory more practical.”
>
> **Response**:
>
> Thank you for raising this point! We agree with the reviewer that additional discussion of the practical implications of our theoretical assumptions and whether relaxations are possible are important for the paper. We have thus included a new section (Section H.1), which includes several new discussion paragraphs to this end. To summarize, we have added discussion paragraphs on:
>
> - A limitation of interaction asymmetry in that it requires the order of interaction to be the same for all slots. We discuss situations where this will likely not hold in practice and whether this assumption could potentially be relaxed.
>
> - The implications of sufficient independence on the observed dimensionality of the data and whether this will hold in practice.
> - Real-word concepts which might not be captured by interaction asymmetry.
>
> - The restrictiveness of the aligned connectedness condition on the latent support from a mathematical standpoint.
>
>
> **Comment**: “How does the interaction asymmetry principle relate to the Information Bottleneck theory?”
>
> **Response**:
>
> Thank you for this comment as well as the references. Based on your suggestion, we have included a discussion paragraph on the relationship between interaction asymmetry and the Information Bottleneck (IB) principle which can be found in Section H.1. To briefly summarize its contents, the principles have a key difference in that the IB principle aims to enforce conditions on the latent distribution, while interaction asymmetry is formalized only on the generator function, and thus does not impose any conditions on the latent distribution. With this being said, our theoretical results do yield insights which resemble the IB principle. Specifically, our results suggest that a model should encode observations $\boldsymbol{x}$ such that unnecessary latent dimensions contain no information about $\boldsymbol{x}$. This resembles the idea in the IB principle of minimizing mutual information between observations and latents.

---

### Official Review · Reviewer_K8sM · 2024-11-04

**Soundness:** 3
**Presentation:** 3
**Contribution:** 3
**Rating:** 6
**Confidence:** 3

**Summary:**

This paper investigates the challenge of learning disentangled and composable representations, crucial for robust generalization in machine learning. It introduces the principle of "interaction asymmetry," positing that parts of the same concept interact more complexly than parts of different concepts. This principle is formalized through block-diagonality conditions on higher-order derivatives of the generator function mapping concepts to observed data, with the order of the derivative reflecting the "complexity" of interaction. The authors theoretically demonstrate that interaction asymmetry enables both disentanglement and compositional generalization, unifying and extending prior work focused specifically on visual objects. They show that existing results for object-centric learning emerge as special cases of their framework, corresponding to lower-order interactions (0th and 1st order). The theory suggests practical criteria for disentanglement: enforcing invertibility with minimal latent dimensions and penalizing interactions across concept slots during decoding. The authors propose an attention-regularized Transformer-based VAE, leveraging the VAE framework for capacity control and incorporating a novel regularizer on the decoder's attention weights to minimize cross-slot interactions. Experiments on synthetic image datasets with objects demonstrate the effectiveness of this approach, showing that the proposed model learns disentangled object representations, achieving performance comparable to existing methods that rely on more explicit object-centric priors.

**Strengths:**

- the principle of interaction asymmetry, offers a novel and insightful perspective on disentanglement and compositionality. While previous works have explored related ideas, the formalization through higher-order derivatives and the connection to the complexity of interaction is original and provides a unifying framework for prior results in object-centric learning. The proposed regularizer on the Transformer's attention weights is a creative application to a scalable architecture.
- The theoretical results are rigorously presented and appear sound. The connection to existing works is carefully established, demonstrating the generality of the proposed framework. The experiments, while conducted on synthetic datasets, offer convincing evidence for the effectiveness of the proposed method and regularizer.
- The paper is generally well-written and easy to follow. The authors clearly motivate their work, provide intuitive explanations of the core ideas, and present the theoretical results in a structured and accessible manner. The figures effectively illustrate the key concepts and experimental results.

**Weaknesses:**

- the aligned-connectedness condition on $Z_{supp}$ seems to be restrictive and its practical implications are not fully explored.
- The proposed regularizer, while efficient, only approximately enforces minimal interactions. Analyzing the gap between this approximation and the theoretical ideal, and exploring alternative regularization strategies that more directly address higher-order interactions, could further enhance the method's effectiveness.
- The experiments primarily focus on synthetic datasets and object-centric learning. While this provides a controlled setting for evaluating disentanglement, exploring the effectiveness of the proposed method on more complex, real-world datasets and diverse concept types (e.g., attributes, events) is crucial for demonstrating its broader applicability.

**Questions:**

Please see above comments.

---

> ### Author Response · Authors · 2024-11-22
>
> We thank the reviewer for their valuable feedback and their positive assessment of our work. We address each of your comments below:
>
> **Comment**: “Exploring the effectiveness of the proposed method on more complex [...] datasets […] is crucial for demonstrating its broader applicability."
>
> **Response**:
>
> We agree with the reviewer that adding experiments on more complex data is important to demonstrate our method’s broader applicability. We have thus conducted additional experiments on the CLEVRTex dataset which is a significant step up in complexity compared to CLEVR6. We found that our method is also able to achieve strong object disentanglement on this data, outperforming the baseline methods we tested against in terms of our metrics. We discuss these results in more detail in the general reply to all reviewers and have added a new section in the paper (Appendix I) presenting our experimental setup and results.
>
> **Comment**: “Aligned connectedness seems restrictive and its practical implications not fully explored”.
>
> **Response**:
>
> Thank you for raising this point. We have now included a discussion paragraph in Section H.1 discussing the restrictiveness of the aligned connectedness condition. To briefly summarize, we discuss the restrictiveness of this assumption from a mathematical point of view, highlighting that it is not an overly restrictive or unrealistic condition. For example, any set that is convex is also aligned connected. We agree with the reviewer, however, that, in practice, understanding whether aligned-connectedness will generally hold, and to what extent it can be violated are important questions which we leave for future work.
>
> **Comment**: . “Analyzing the gap between approximation and theoretical ideal, and exploring alternative regularization strategies that more directly address higher-order interactions, could enhance the method's effectiveness”
>
> **Response**:
>
> We agree with the reviewer that understanding the gap between our regularizer and the theoretical ideal is an interesting question. To this end, we computed the Pearson correlation between the value of our regularizer throughout training and the Jacobian Interaction Score, which can be seen as an exact measure of $0^{\text{th}}$ order interaction. On Sprites, we found an absolute correlation of .93, and .77 on CLEVR6. This indicates that while there is some gap between our regularizer and the theoretical ideal, the two scores are strongly correlated. Regarding directly regularizing higher-order interactions, as noted in Section F.2 of our paper, we agree with the reviewer that this is an important question, which we leave for future work to explore further.

---

### Author Response · Authors · 2024-11-22
**General Reply to All Reviewers**

We thank reviewers for their valuable feedback, and their assessment of our work as “a significant step toward learning useful representations for real-world tasks“ (`qqbq`) and “offering meaningful insights that contribute to our understanding of compositional representation learning” (`c6zh`). We appreciate that the reviewers found our theory “interesting and valuable” (`c6zh`) and “creative” (`A7GP`, `qqbq`), and our experiments as “convincing” (`K8sM`) and “thorough” (`qqbq`). We were also pleased that all reviewers rated the soundness, contribution, and presentation of our work as good or excellent.

We are grateful for the constructive feedback given by all reviewers. Based on this feedback, we have made several additions to the manuscript which are highlighted in yellow in the revised version. We briefly summarize the most significant additions below:

**Experiments on More Complex Data**

Reviewers `K8sM`, `qqbq`, and `c6zh` each highlighted that our contribution would be further strengthened by experiments on more complex data. To this end, we would first like to note that we believe that our methodological and empirical contributions are already more extensive than other theoretical studies of disentanglement and compositional generalization (e.g, [1, 2, 3, 4] ). With this being said, we agree with the reviewers that experiments on more complex data are important for better understanding the utility of our proposed method.

We have thus run additional experiments on the CLEVRTex dataset [5]. This dataset is a significant step up in complexity from CLEVR6, and has been shown to be very challenging for popular object-centric models [5, 6]. We evaluate three models on this dataset:

- Our regularized Transformer autoencoder, ($\alpha=.1, \beta=.1$)
- An unregularized variant of this model ($\alpha=0, \beta=0$)
- An unregularized variant in which the Transformer encoder is replaced with Slot Attention.

We train all models using the same protocol as on Sprites and CLEVR6, with the one key difference in that we do not reconstruct the original image but instead reconstruct a representation of the image given by a pre-trained Vision Transformer. This training method, referred to as DINOSAUR [7], has been shown to be helpful for scaling object-centric models to complex data. We note that by testing our method with DINOSAUR, we are also better able to understand its applicability to more scalable training pipelines.

We report results in the table below after training for 300,000 iterations across 3 seeds for each model:

| Encoder | Loss | J-ARI | JIS |
|------------------|------------------|------------------|------------------|
| Transformer | $\alpha=0, \beta=0$ | $81.4 \pm 3.7$ | $50.9 \pm 2.8$ |
| Slot Attention | $\alpha=0, \beta=0$ | $94.2 \pm 0.2$ | $54.4 \pm 0.3$ |
| Transformer | $\alpha=0.1, \beta=0.1$ **(Ours)** | $\mathbf{95.9} \pm \mathbf{0.06}$ | $\mathbf{65.4} \pm \mathbf{0.6}$

Similar to Sprites and CLEVR6, we find that **the regularized Transformer achieves superior object disentanglement** in terms of J-ARI and JIS compared to these baseline models. We also visually corroborate these results by including plots of the Jacobian of the decoder for each model in Fig 7. in App J.3. We have added a dedicated section (Section I), discussing these results in more detail. We will conduct experiments for additional baselines and ablations, in the camera-ready version of this work, and then will add these results to the main text.

**Additional Discussion of Theoretical Assumptions**

Reviewers `K8sM` and `qqbq` suggested that our work would benefit from additional discussions on the implications of our theoretical assumptions and whether relaxations are possible. To address this, we have added a new section, App. H.1, that includes several such discussions. We elaborate further on the content of this section in our individual replies.

**Discussion on Scalability of Method and Relationship to Slot Attention**

Reviewer `c6zh` raised several points related to the scalability of our proposed method as well as its relationship to the object-centric method, Slot Attention. We have included several additional discussion paragraphs in App H.2, addressing these points in detail. This is elaborated upon further in our individual reply.

[1] Brady et al., 2023, Provably Learning Object-Centric Representations

[2] Lachapelle et al., 2024, Additive Decoders for Latent Variables Identification and Cartesian Product Extrapolation

[3] Wiedemer et al., 2023, Compositional Generalization from First Principles

[4] Wiedemer et al., 2024, Provable Compositional Generalization for Object-Centric Learning

[5] Karazija et al., 2021, ClevrTex: A Texture-Rich Benchmark for Unsupervised Multi-Object Segmentation

[6] Biza et al., 2023, Invariant Slot Attention: Object Discovery with Slot-Centric Reference Frames

[7] Seitzer et al, 2023 Bridging the Gap to Real-World Object-Centric Learning

---

### Meta-Review · Area_Chair_Se5k · 2024-12-14

**Metareview:**

# Summary
This paper investigates the conditions for disentangling concept attributes. It
proposes a theory of interaction asymmetry principle for learning disentangled representation, and introduces metrics to evaluate disentanglement. Implementation of theory and experiments are conducted to support the claims.

# strengths
+ The principle of interaction asymmetry, offers a novel and insightful perspective on disentanglement and compositionally.
+ It presents a solid theoretical foundation.
+ It unifies existing work while also extending to more complex scenarios.
+ The paper is well-written and easy to follow.

# weaknesses
- experimental support is limited: need to conduct experiments on more complex data.
- need additional discussion on theoretical assumptions.
- need to discuss the relationship between disentanglement and compositional generalization further.
- need to discuss scalability of method and relationship to Slot Attention.

# reasons for decision
The paper provides insights for better understanding compositional representation learning, and it makes a significant step toward learning useful representations for real-world tasks.
The combination of theoretical insights with a practical Transformer-based VAE implementation is also notable. I recommend accept.

**Additional Comments On Reviewer Discussion:**

The authors had a successful rebuttal. Although the reviewers had concerns on theoretical assumptions, scalability of the proposed method and its connection to compositional generalization, the authors rebuttal resolved the above concerns. All reviewers raised their scores and are positive about the paper.

---

### Decision · Program_Chairs · 2025-01-22

Accept (Poster)